# Differentially Private Algorithms for Learning Mixtures of Separated Gaussians

**Gautam Kamath**
David R. Cheriton School of Computer Science
University of Waterloo
Waterloo, ON, Canada N2L 3G1
g@csail.mit.edu

**Or Sheffet**
Department of Computer Science, Faculty of Exact Sciences
Bar-Ilan University
Ramat-Gan, 5290002 Israel
or.sheffet@biu.ac.il

**Vikrant Singhal**
Khoury College of Computer Sciences
Northeastern University
360 Huntington Ave., Boston, MA 02115
singhal.vi@husky.neu.edu

**Jonathan Ullman**
Khoury College of Computer Sciences
Northeastern University
360 Huntington Ave., Boston, MA 02115
jullman@ccs.neu.edu

## Abstract

Learning the parameters of Gaussian mixture models is a fundamental and widely studied problem with numerous applications. In this work, we give new algorithms for learning the parameters of a high-dimensional, well separated, Gaussian mixture model subject to the strong constraint of differential privacy. In particular, we give a differentially private analogue of the algorithm of Achlioptas and McSherry (COLT 2005). Our algorithm has two key properties not achieved by prior work: (1) The algorithm's sample complexity matches that of the corresponding non-private algorithm up to lower order terms in a wide range of parameters. (2) The algorithm requires very weak *a priori* bounds on the parameters of the mixture.

## 1   Introduction

The *Gaussian mixture model* is one of the most important and widely studied models in Statistics—with roots going back over a century [58]—and has numerous applications in the physical, life, and social sciences. In a Gaussian mixture model, we suppose that each sample is drawn by randomly selecting from one of $k$ distinct Gaussian distributions $G_1, \ldots, G_k$ in $\mathbb{R}^d$ and then drawing a sample from that distribution. The problem of *learning a Gaussian mixture model* asks us to take samples from this distribution and approximately recover the parameters (mean and covariance) of each of the underlying Gaussians. The past decades have seen tremendous progress towards understanding both the sample complexity and computational complexity of learning Gaussian mixtures [20, 21, 5, 64, 3, 17, 16, 52, 7, 59, 41, 27, 51, 46, 54, 10, 42, 4, 11, 40, 38, 66, 23, 6, 35, 36, 22, 63, 26, 53, 15].

---

Due to significant space restrictions, a full version of the paper, with additional details and all proofs, appears in the supplementary material [48].

In many of the applications of Gaussian mixtures models, especially those in the social sciences, the sample consists of sensitive data belonging to individuals. In these cases, it is crucial that we not only learn the parameters of the mixture model, but do so while preserving these individuals' *privacy*. In this work, we study algorithms for learning Gaussian mixtures subject to *differential privacy* [32], which has become the *de facto* standard for individual privacy in statistical analysis of sensitive data. Intuitively, differential privacy guarantees that the output of the algorithm does not depend significantly on any one individual's data, which in this case means any one sample. Differential privacy is used as a measure of privacy for data analysis systems at Google [34], Apple [28], and the U.S. Census Bureau [19]. Differential privacy and related notions of *algorithmic stability* are also crucial for statistical validity even when individual privacy is not a primary concern, as they provide generalization guarantees in an adaptive setting [31, 9].

The first differentially private algorithm for learning Gaussian mixtures comes from the work of Nissim, Raskhodnikova, and Smith [55] as an application of their influential *subsample-and-aggregate* framework. However, their algorithm is a reduction from private estimation to non-private estimation that blows up the sample complexity by at least a $\text{poly}(d)$ factor.

The contribution of this work is new differentially private algorithms for recovering the parameters of an unknown Gaussian mixture provided that the components are sufficiently well separated. In particular we give differentially private analogues of the algorithm of Achlioptas and McSherry [3], which requires that the means are separated by a factor proportional to $\sqrt{k}$, but independent of the dimension $d$. Our algorithms have two main features not shared by previous methods:

- The sample complexity of the algorithm matches that of the corresponding non-private algorithm up to lower order additive terms for a wide range of parameters.

- The algorithm requires very weak *a priori* bounds on the parameters of the mixture components. That is, like many algorithms, we require that the algorithm is seeded with some information about the range of the parameters, but the algorithm's sample complexity depends only mildly (polylogarithmically) on the size of this range.

## 1.1 Problem Formulation

There are a plethora of algorithms for (non-privately) learning Gaussian mixtures, each with different learning guarantees under different assumptions on the parameters of the underlying distribution.[1] In this section we describe the version of the problem that our work studies and give some justification for these choices.

We assume that the underlying distribution $\mathcal{D}$ is a mixture of $k$ Gaussians in high dimension $d$. The mixture is specified by $k$ *components*, where each component $G_i$ is selected with probability $w_i \in [0, 1]$ and is distributed as a Gaussian with mean $\mu_i \in \mathbb{R}^d$ and covariance $\Sigma_i \in \mathbb{R}^{d \times d}$. Thus the mixture is specified by the tuple $\{(w_i, \mu_i, \Sigma_i)\}_{i \in [k]}$.

Our goal is to accurately $\alpha$-recover this tuple of parameters. Intuitively, with probability $1 - \beta$, we would like to recover a tuple $\{(\widehat{w}_i, \widehat{\mu}_i, \widehat{\Sigma}_i)\}_{i \in [k]}$, specifying a mixture $\tilde{\mathcal{G}}$ such that $\|\widehat{w} - w\|_1$ is small and $\|\widehat{\mu}_i - \mu_i\|_{\Sigma_i}$ and $\|\widehat{\Sigma}_i - \Sigma_i\|_{\Sigma_i}$ are small for every $i \in [k]$ (specifically, we require them all to be $O(\alpha)$). Here, $\| \cdot \|_\Sigma$ is the appropriate vector/matrix norm that ensures $\mathcal{N}(\mu_i, \Sigma_i)$ and $\mathcal{N}(\widehat{\mu}_i, \widehat{\Sigma}_i)$ are close in total variation distance and we also compare $\widehat{w}$ and $w$ in $\| \cdot \|_1$ to ensure that the mixtures are close in total variation distance. Of course, since the labeling of the components is arbitrary, we can actually only hope to recover the parameters up to some unknown permutation $\pi : [k] \to [k]$ on the components. We say that an algorithm *learns a mixture of Gaussian using $n$ samples* if it takes $n$ i.i.d. samples from an unknown mixture $\mathcal{D}$ and outputs the parameters of a mixture $\widehat{\mathcal{D}}$ satisfies these conditions, which will imply the resulting mixtures are $\alpha$-close in total variation distance.[2]

In this work, we consider mixtures that satisfy the *separation* condition

$$\forall i \neq j \ \ \|\mu_i - \mu_j\|_2 \geq \widetilde{\Omega}\left(\sqrt{k + 1/w_i + 1/w_j}\right) \cdot \max\left\{\|\Sigma_i^{1/2}\|, \|\Sigma_j^{1/2}\|\right\} \tag{1}$$

Note that the separation condition does not depend on the dimension $d$, only on the number of mixture components. However, (1) is *not* the weakest possible condition under which one can learn a mixture of Gaussians. We focus on (1) because in this regime we can learn the mixture components using *statistical properties of the data*, such as the large principal components of the data and the centers of a good clustering, which are amenable to privacy. In contrast, algorithms that learn with separation proportional to $k^{1/4}$ [64], $k^\varepsilon$ [41, 51, 27], or even $\sqrt{\log k}$ [59] use algorithmic machinery such as the sum-of-squares algorithm that has not been studied from the perspective of privacy, or are not computationally efficient. In particular, a barrier to learning with separation $k^{1/4}$ is that the non-private algorithms are based on single-linkage clustering, which is not amenable to privacy due to its crucial dependence on the distance between individual points. We remark that one can also learn without any separation conditions, but only with $\exp(k)$ many samples from the distributions [54].

In this work, our goal is to design learning algorithms for mixtures of Gaussians that are also differentially private. An $(\varepsilon, \delta)$-*differentially private* [32] randomized algorithm $A$ for learning mixtures of Gaussians is an algorithm that takes a dataset $X$ of samples and:

- $A$ is $(\varepsilon, \delta)$-differentially private in the *worst-case*. That is, for *every* pair of samples $X, X' \in \mathcal{X}^n$ differing on one sample, $A(X)$ and $A(X')$ are $(\varepsilon, \delta)$-*close* in the following sense: for $b \in \{0, 1\}$, $\Pr[A(X) = b] \leq e^\varepsilon \Pr[A(X') = b] + \delta$
- If $n$ is sufficiently large and $X_1, \ldots, X_n \sim \mathcal{D}$ for a mixture $\mathcal{D}$ satisfying our assumptions, then $A(X)$ outputs an approximation to the parameters of $\mathcal{G}$.

Note that, while the learning guarantees necessarily rely on the assumption that the data is drawn i.i.d. from some mixture of Gaussians, the privacy guarantee is worst-case. It is important for privacy not to rely on distributional assumptions because we have no way of verifying that the data was truly drawn from a mixture of Gaussians, and if our assumption turns out to be wrong we cannot recover privacy once it is lost.

Furthermore, our algorithms assume certain *boundedness* of the mixture components. Specifically, we assume that there are known quantities $R, \sigma_{\max}, \sigma_{\min}, w_{\min}$ such that

$$\forall i \in [k] \ \ \|\mu_i\|_2 \leq R \ \text{ and } \ \sigma_{\min}^2 \leq \|\Sigma_i\|_2 \leq \sigma_{\max}^2 \ \text{ and } \ w_{\min} \leq w_i. \tag{2}$$

For brevity, we define $\mathcal{G}(\ell, k, R, \sigma_{\min}, \sigma_{\max}, w_{\min}, s)$ to be the family of mixtures of $k$ Gaussians in $\ell$ dimensions satisfying 2 and a separation condition defined by $s$ (in our case, $s$ conforms to 1). These assumptions are to some extent necessary, as even the state-of-the-art algorithms for learning a single multivariate normal [47] require boundedness.[3] However, since $R$ and $\sigma_{\max}/\sigma_{\min}$ can be quite large—and even if they are not we cannot expect the user of the algorithm to know these parameter *a priori*—the algorithm's sample complexity should depend only mildly on these parameters so that they can be taken to be quite large.

## 1.2 Our Contributions

The main contribution of our work is an algorithm with improved sample complexity for learning mixtures of Gaussians that are separated and bounded.

**Theorem 1.1** (Main, Informal). *There is an $(\varepsilon, \delta)$-differentially private algorithm that takes*

$$n = \left(\frac{d^2}{\alpha^2 w_{\min}} + \frac{d^2}{\alpha w_{\min}\varepsilon} + \frac{\text{poly}(k)d^{3/2}}{w_{\min}\varepsilon}\right) \cdot \text{polylog}\left(\frac{dkR(\sigma_{\max}/\sigma_{\min})}{\alpha\beta\varepsilon\delta}\right)$$

*samples from an unknown mixture of $k$ Gaussians $\mathcal{D}$ in $\mathbb{R}^d$ satisfying (1) and (2), where $w_{\min} = \min_i w_i$, and, with probability at least $1 - \beta$, learns the parameters of $\mathcal{D}$ up to error $\alpha$.*

We remark that the sample complexity in Theorem 1.1 compares favorably to the sample complexity of methods based on subsample-and-aggregate. In particular, when $\varepsilon \geq \alpha$ and $k$ is a small polynomial in $d$, the sample complexity is dominated by $d^2/\alpha^2 w_{\min}$, which is optimal even for non-private algorithms. In Section 5 we give an optimized version of the subsample-and-aggregate-based reduction from [55] and show that we can learn mixtures of Gaussians with sample complexity roughly $\tilde{O}(\sqrt{d}/\varepsilon)$ times the sample complexity of the corresponding non-private algorithm. In contrast the sample complexity of our algorithm does not grow by dimension-dependent factors compared to the non-private algorithm on which it is based.

At a high level, our algorithm mimics the approach of Achlioptas and McSherry [3], which is to use PCA to project the data into a low-dimensional space, which has the effect of projecting out much of the noise, and then recursively clustering the data points in that low-dimensional space. However, where their algorithm uses a Kruskal-based clustering algorithm, we have to use alternative clustering methods that are more amenable to privacy. We develop our algorithm in two distinct phases addressing different aspects of the problem:

In Section 3 we consider an "easy case" of Theorem 1.1, where we assume that: all components are spherical Gaussians, such that variances of each component lie in a small, known range (such that their ratio is bounded by a constant factor) and that the means of the Gaussians lie in a small ball around the origin. Under these assumptions, it is fairly straight-forward to make the PCA-projection step [64, 3] private. The key piece of the algorithm that needs to be private is computing the principal components of the data's covariance matrix. We can make this step private by adding appropriate noise to this covariance, and the key piece of the analysis is to analyze the effect of this noise on the principal components, extending the work of Dwork *et al.* [33] on private PCA. Using the assumptions we make in this easy case, we can show that the projection shifts each component's mean by $O(\sqrt{k}\sigma_{\max})$, which preserves the separation of the data because all variances are within constant factor of one another. Then, we iteratively cluster the data using the 1-cluster technique of [57, 56]. Lastly, we apply a simplified version of [47] to learn each component.

We then consider the general case where the Gaussians can be non-spherical and wildly different from each other. In this case, if we directly add noise to the covariance matrix to achieve privacy, then the noise will scale polynomially with $\sigma_{\max}/\sigma_{\min}$, which is undesirable. For the general case, we use a recursive algorithm, which repeatedly identifies a *secluded* cluster in the data, and then recurses on this isolated cluster and the points outside of the cluster. To that end we develop in Section 4.1 a variant of the private clustering algorithm of [57, 56] that finds a secluded ball—a set of points that lie inside of some ball $B_r(p)$ such that the annulus $B_{10r}(p) \setminus B_r(p)$ is (essentially) empty.[4]

We can obtain a recursive algorithm in the following way. First we try to find a secluded ball in the unprojected data. If we find one then we can split and recurse on the inside and outside of the ball. If we cannot find a ball, then we can argue that the diameter of the dataset is $\mathrm{poly}(d, k, \sigma_{\max})$. In the latter case, we can ensure that with $\mathrm{poly}(d, k)$ samples, the PCA-projection of the data preserves the mean of each component up to $O(\sqrt{k}\sigma_{\max})$, which guarantees that the cluster with the largest variance is secluded, in which case we can find the secluded ball and recurse.

## 1.3 Related Work

There has also been a great deal of work on learning mixtures of distribution classes, particularly mixtures of Gaussians. There are many ways the objective can be defined, including clustering [20, 21, 5, 64, 3, 17, 16, 52, 7, 59, 41, 27, 51], parameter estimation [46, 54, 10, 42, 4, 11, 40, 38, 66, 23, 6], proper learning [35, 36, 22, 63, 26, 53], and improper learning [15].

There has recently been a great deal of interest in differentially private distribution learning, the most relevant being [50, 47], which focus on learning a single Gaussian. There are also algorithms for learning structured univariate distributions in TV-distance [25], and learning arbitrary univariate distributions in Kolomogorov distance [13]. Upper and lower bounds for learning the mean of a product distribution over the hypercube in $\ell_\infty$-distance include [12, 14, 32, 61]. [1] focuses on estimating properties of a distribution, rather than the distribution itself. [60] gives an algorithm which allows one to estimate asymptotically normal statistics with minimal increase in the sample

complexity. There has also been a great deal of work on distribution learning in the local model of differential privacy [29, 65, 45, 2, 30, 44, 67, 37].

Within differential privacy, there are many algorithms for tasks that are related to learning mixtures of Gaussians, notably PCA [12, 49, 18, 33] and clustering [55, 39, 57, 56, 8, 62, 43]. Applying these algorithms naïvely to the problem of learning Gaussian mixtures would necessarily introduce a polynomial dependence on the range of the data, which we seek to avoid. Nonetheless, private algorithms for PCA and clustering feature in our solution, which builds directly on these works.

## 2    Robustness of PCA-Projection to Noise

One of the main tools used in learning mixtures of Gaussians under separation is principal component analysis (PCA). Specifically, we project onto the subspace spanned by the top $k$ principal components, which has the effect of preserving the means of the components while eliminating the directions that are purely noise. In this section, we show that PCA achieves a similar effect even when adding additional noise for privacy.

Before showing the result for perturbed PCA, we reiterate the very simple analysis of Achlioptas and McSherry [3]. Let $X \in \mathbb{R}^{n \times d}$ be a matrix of samples and $A \in \mathbb{R}^{n \times d}$ be the rank-$k$ matrix of the corresponding cluster centers. Fixing a cluster $i$, denoting its empirical mean of as $\bar{\mu}_i$, the mean of the resulting projection as $\hat{\mu}_i$, $\Pi$ as the $k$-PCA projection matrix and $u_i \in \{0,1\}^n$ as the vector indicating which datapoint was sampled from cluster $i$, we have

$$\|\bar{\mu}_i - \hat{\mu}_i\|_2 = \|\tfrac{1}{n_i}\left(X^T - (X\Pi)^T\right)u_i\|_2 \leq \|X - X\Pi\|_2 \tfrac{\|u_i\|_2}{n_i} \leq \tfrac{1}{\sqrt{n_i}}\|X - A\|_2,$$

where the last inequality follows from the $X\Pi$ being the best $k$-rank approximation of $X$ whereas $A$ is any rank-$k$ matrix. We extend this analysis to a perturbed $k$-PCA projection as given by the following lemma.

**Lemma 2.1.** *Let $X \in \mathbb{R}^{n \times d}$ be a collection of $n$ datapoints from $k$ clusters each centered at $\mu_1, \mu_2, ..., \mu_k$. Let $A \in \mathbb{R}^{n \times d}$ be the corresponding matrix of (unknown) centers (for each $j$ we place the center $\mu_{c(j)}$ with $c(j)$ denoting the clustering point $X_j$ belongs to). Let $\Pi_{V_k} \in \mathbb{R}^{d \times d}$ denote the $k$-PCA projection of $X$'s rows. Let $\Pi_U \in \mathbb{R}^{d \times d}$ be a projection such that for some bound $B \geq 0$ it holds that $\|X^T X - (X\Pi_U)^T(X\Pi_U)\|_2 \leq \|X^T X - (X\Pi_{V_k})^T(X\Pi_{V_k})\|_2 + B$. Denote $\bar{\mu}_i$ as the empirical mean of all points in cluster $i$ and denote $\hat{\mu}_i$ as the projection of the empirical mean $\hat{\mu}_i = \Pi_U \bar{\mu}_i$. Then $\|\bar{\mu}_i - \hat{\mu}_i\|_2 \leq \tfrac{1}{\sqrt{n_i}}\|X - A\|_2 + \sqrt{B/n_i}$.*

We can instantiate in the following lemma for mixtures of Gaussians.

**Lemma 2.2.** *Let $X \in \mathbb{R}^{n \times d}$ be a sample from a mixture of Gaussians $\mathcal{D}$, and let $A \in \mathbb{R}^{n \times d}$ be the matrix where each row $i$ is the (unknown) mean of the Gaussian from which $X_i$ was sampled. For each $i$, let $\sigma_i^2$ denote the maximum directional variance of component $i$, and $w_i$ denote its mixing weight. Define $\sigma^2 = \max_i\{\sigma_i^2\}$ and $w_{\min} = \min_i\{w_i\}$. If $n \geq \frac{1}{w_{\min}}(\xi_1 d + \xi_2 \log(2k/\beta))$, where $\xi_1, \xi_2$ are universal constants, then with probability at least $1 - \beta$, $\frac{\sqrt{n w_{\min}}\sigma}{4} \leq \|X - A\|_2 \leq 4\sqrt{n \sum_{i=1}^{k} w_i \sigma_i^2}$.*

## 3    A Warm Up: Strongly Bounded Spherical Gaussian Mixtures

We first give an algorithm to learn mixtures of spherical Gaussians, whose variances are within a constant factor of one another, and means lie in a ball of radius $k\sqrt{d}\sigma$, where $\sigma$ is the largest variance, and all mixing weights are uniform. We denote such a family of spherical Gaussians by $\mathcal{S}(d, k, \kappa, s)$, where $\kappa$ is an upper bound on the ratio of maxiumum and minimum variances (a constant), and $s$ defines the separation condition. Now, we present the main theorem of this section.

**Theorem 3.1.** *There exists an $(\varepsilon, \delta)$-differentially private algorithm, which if given $n$ independent samples from $\mathcal{D} \in \mathcal{S}(d, k, \kappa, C\sqrt{\ell})$, such that $C = \xi + 16\sqrt{\kappa}$, where $\xi, \kappa \in \Theta(1)$ and $\xi$ is a universal constant, $\ell = \max\{512\ln(nk/\beta), k\}$, and*

$$n \geq \left(\frac{dk}{\alpha^2} + \frac{d^{\frac{3}{2}}k^3}{\varepsilon} + \frac{dk}{\alpha\varepsilon} + \frac{\sqrt{d}k\ln(k/\beta)}{\alpha\varepsilon} + \frac{k^2}{\alpha^2} + \frac{\ell^{\frac{5}{9}}k^{\frac{5}{3}}}{\varepsilon^{\frac{10}{9}}}\right) \cdot \operatorname{polylog}\left(\ell, k, \frac{1}{\varepsilon}, \frac{1}{\delta}, \frac{1}{\beta}\right)$$

*then it $(\alpha, \beta)$-learns $\mathcal{D}$.*

We assume that our algorithm is given samples $X \in \mathbb{R}^{n \times d}$, a parameter $\sigma_{\min}$ that is the exact smallest variance in the mixture and the parameter $\kappa$. The algorithm proceeds as follows:

(1) We first truncate the dataset so that all points lie within a ball of radius $2k\sqrt{d}\kappa\sigma_{\min}$ around the origin. So with high probability, no points in $X$ are lost in this step.

(2) Next, we do privacy-preserving $\ell$-PCA. By the guarantees in the previous section and from Theorem 9 of [33], since all variances are almost identical, the distances between projected means are changed by at most $O(\sqrt{\ell}\sigma_{\min})$, which maintains the separation.

(3) Now, that the Gaussians have shrunk down, and are still far apart, we can find individual components using an extension of the 1-cluster algorithm of [56] given below.

**Theorem 3.2** (Extension of [56]). *There is an $(\varepsilon, \delta)$-DP algorithm $\mathrm{PGLoc}(X, t; \varepsilon, \delta, R, \sigma_{\min}, \sigma_{\max})$ with the following guarantee. Let $X = (X_1, \ldots, X_n) \in \mathbb{R}^{n \times d}$ be a set of $n$ points drawn from a mixture of Gaussians $\mathcal{D} \in \mathcal{G}(\ell, k, R, \sigma_{\min}, \sigma_{\max}, w_{\min}, s)$. Let $S \subseteq X$ such that $|S| \geq t$, and let $0 < a < 1$ be any small absolute constant (say, one can take $a = 0.1$). If $t = \gamma n$, where $0 < \gamma \leq 1$, and*

$$ n \geq \left(\frac{\sqrt{\ell}}{\gamma \varepsilon}\right)^{\frac{1}{1-a}} \cdot 9^{\log^*\left(\sqrt{\ell}\left(\frac{R\sigma_{\max}}{\sigma_{\min}}\right)^\ell\right)} \cdot \mathrm{polylog}\left(\ell, \frac{1}{\varepsilon}, \frac{1}{\delta}, \frac{1}{\beta}, \frac{1}{\gamma}\right) + O\left(\frac{\ell + \log(k/\beta)}{w_{\min}}\right) $$

*then for some absolute constant $c > 4$ that depends on $a$, with probability at least $1 - \beta$, the algorithm outputs $(r, \vec{c})$ such that the following hold: (1) $B_r(\vec{c})$ contains at least $\frac{t}{2}$ points in $S$, that is, $|B_r(\vec{c}) \cap S| \geq \frac{t}{2}$. (2) If $r_{\mathrm{opt}}$ is the radius of the smallest ball containing at least $t$ points in $S$, then $r \leq c(r_{\mathrm{opt}} + \frac{1}{4}\sqrt{\ell}\sigma_{\min})$.*

For simplicity, we set the constant $a = 0.1$. Since we have balls that isolate individual components in the lower-dimensional space and that give us a constant factor approximation of each variance, we can partition the original dataset to find a set of nearly optimal balls containing the points from each cluster.

(4) We can finally learn the parameters of each individual Gaussian using a simplified version of the Gaussian learner of [47] that is tailored specifically for spherical Gaussians.

**Theorem 3.3.** *There exists an $(\varepsilon, \delta)$-differentially private algorithm $\mathrm{PSGE}(X; \vec{c}, r, \alpha_\mu, \alpha_\sigma, \beta, \varepsilon, \delta)$ with the following guarantee. If $B_r(\vec{c}) \subseteq \mathbb{R}^\ell$ is a ball, $X_1, \ldots, X_n \sim \mathcal{N}(\mu, \sigma^2 \mathbb{I}_{\ell \times \ell})$, and $n \geq \frac{6 \ln(5/\beta)}{\varepsilon} + n_\mu + n_\sigma$, where*

$$ n_\mu = O\left(\frac{\ell}{\alpha_\mu^2} + \frac{\ln(\frac{1}{\beta})}{\alpha_\mu^2} + \frac{r \ln(\frac{1}{\beta})}{\alpha_\mu \varepsilon \sigma} + \frac{r \sqrt{\ell \ln(\frac{1}{\delta})}}{\alpha_\mu \varepsilon \sigma}\right), n_\sigma = O\left(\frac{\ln(\frac{1}{\beta})}{\alpha_\sigma^2 \ell} + \frac{\ln(\frac{1}{\beta})}{\alpha_\sigma \varepsilon} + \frac{r^2 \ln(\frac{1}{\beta})}{\alpha_\sigma \varepsilon \sigma^2 \ell}\right), $$

*then with probability at least $1 - \beta$, the algorithm returns $\widehat{\mu}, \widehat{\sigma}$ such that if $X$ is contained in $B_r(\vec{c})$ (that is, $X_i \in B_r(\vec{c})$ for every $i$) and $\ell \geq 8 \ln(10/\beta)$, then $\|\mu - \widehat{\mu}\|_2 \leq \alpha_\mu \sigma$ and $(1 - \alpha_\sigma) \leq \frac{\widehat{\sigma}^2}{\sigma^2} \leq (1 + \alpha_\sigma)$.*

# 4 An Algorithm for Privately Learning Mixtures of Gaussians

In this section, we present our main algorithm for privately learning mixtures of Gaussians and prove Theorem 1.1 from the introduction.

## 4.1 Finding a Terrific Ball

In this section we mention a key building block in our algorithm for learning Gaussian mixtures. This particular subroutine is an adaptation of the work of Nissim and Stemmer [56] (who in turn built on [57]) that finds a ball that contains many datapoints. In this section we show how to tweak their algorithm so that it now produces a ball with a few more additional properties. More specifically, our goal in this section is to privately locate a ball $B_r(p)$ that (i) contains many datapoints, (ii) leaves out many datapoints and (iii) is secluded in the sense that $B_{2r}(p) \setminus B_r(p)$ holds very few (and ideally zero) datapoints. More specifically, we are using the following definition.

**Definition 4.1.** Given a dataset $X$ and an integer $t > 0$, we say a ball $B_r(p)$ is $(c, \Gamma)$-terrific for a constant $c > 1$ and parameter $\Gamma \geq 0$ if all of the following three properties hold: (i) The number of datapoints in $B_r(p)$ is at least $t - \Gamma$; (ii) The number of datapoints outside the ball $B_{cr}(p)$ is least $t - \Gamma$; and (iii) The number of datapoints in the annulus $B_{cr}(p) \setminus B_r(p)$ is at most $\Gamma$.

We say a ball is $c$-terrific if it is $(c, 0)$-terrific, and when $c$ is clear from context we call a ball terrific. In this section, we provide a differentially private algorithm that locates a terrific ball.

The algorithm of [56] is composed of two subroutines. The first, `GoodRadius` privately computes some radius $\tilde{r}$ such that $\tilde{r} \leq 4r_{\text{opt}}$ with $r_{\text{opt}}$ denoting the radius of the smallest ball that contains $t$ datapoints. Their next subroutine, `GoodCenter`, takes $\tilde{r}$ as an input and produces a ball of radius $\gamma\tilde{r}$ that holds (roughly) $t$ datapoints with $\gamma$ denoting some constant $> 2$. The `GoodCenter` procedure works by first cleverly combining locality-sensitive hashing (LSH) and randomly-chosen axes-aligned boxed to retrieve a poly-length list of candidate centers, then applying ABOVETHRESHOLD to find a center point $p$ such that the ball $B_{\tilde{r}}(p)$ satisfies the required (holding enough datapoints).

In our modification, the latter procedure is essentially unchanged, with the single condition before replaced by the three conditions which we require. The previous procedure also requires the same modification, but since our new scoring function (which takes into account all three conditions) is no longer monotone or quasi-concave, we perform a capping of the scores to facilitate our search. Our modification to `GoodRadius` is called `TerrificRadius`, to reflect its stronger guarantees.

The resulting combination of these tools gives the following lemma.

**Lemma 4.2.** *The* `TerrificBall` *procedure is a* $(2\varepsilon, \delta)$-*DP algorithm which, if run using size-parameter* $t \geq \frac{1000c^2}{\varepsilon} n^a \sqrt{d} \log(nd/\beta) \log(1/\delta) \log\log(U/L)$ *for some arbitrary small constant* $a > 0$ *(say* $a = 0.1$*), and is set to find a $c$-terrific ball with $c > \gamma$ ($\gamma$ being the parameter fed into the LSH in the* `GoodCenter` *procedure), then the following holds. With probability $\geq 1 - 2\beta$ if it returns a ball $B_p(r')$, and furthermore this ball is a $(c, 2\Gamma)$-terrific ball of radius $r' \leq (1 + \frac{c}{10})r$.*

## 4.2 The Algorithm

We finally introduce our differentially private version of the Achlioptas-McSherry algorithm. Recall that we assume the separation condition

$$\forall i, j, \quad \|\mu_i - \mu_j\| \geq C(\sigma_i + \sigma_j)\left(\sqrt{k \log(n)} + 1/\sqrt{w_i} + 1/\sqrt{w_j}\right) \tag{3}$$

for some constant $C > 0$, and that $n \geq \text{poly}(d, k)$ (we note that our constant $C$ is larger than that of [3]. We also make one additional technical assumption that the Gaussians are not "too skinny."

$$\forall i, \quad \|\Sigma_i\|_F \sqrt{\log(n/\beta)} \leq \tfrac{1}{8}\text{tr}(\Sigma_i), \quad \text{and} \quad \|\Sigma_i\|_2 \log(n/\beta) \leq \tfrac{1}{8}\text{tr}(\Sigma_i) \tag{4}$$

Note the for a spherical Gaussian (where $\Sigma_i = \sigma_i^2 I_{d \times d}$) we have that $\text{tr}(\Sigma_i) = d\sigma_i^2$, while $\|\Sigma_i\|_F = \sigma_i^2 \sqrt{d}$ and $\|\Sigma_i\|_2 = \sigma_i^2$, thus the above condition translates to requiring a sufficiently large dimension, as assumption made explicit in the work regarding learning spherical Gaussians [64].

We now detail the main components of our algorithm. Algorithm 1 takes the dataset $X$ and returns a $k$-partition of $X$ into subsets corresponding to different mixture components. There are two key points to note about this algorithm. First, the parameter $k$ is an upper bound on the number of mixture components that have points in $X$, so in every recursive call, even though we specify some upper bound $k'$, the number of clusters returned will match the exact number of components. Second, the partition itself cannot be private (since it is a list of points in the dataset). So once this $k$-partition is done, one must apply the existing $(\varepsilon, \delta)$-DP procedure (called, "PGE", which is an adaptation of the private learner for high-dimensional Gaussians from [47] for the case, where a tiny fraction of the points may be lost, as described in the full version of this paper) for estimating the mean and the covariance of each cluster, as well as apply the $\varepsilon$-DP histogram to find the cluster weights. The overall algorithm (PGME) is in Algorithm 2.

The main theorem of our section is as follows.

**Theorem 4.3.** *There is an* $(\varepsilon, \delta)$-*differentially private algorithm that takes*

$$n = \left(\frac{d^2}{\alpha^2 w_{\min}} + \frac{d^2}{\alpha w_{\min}\varepsilon} + \frac{k^{9.06}d^{3/2}}{w_{\min}\varepsilon} + \left(\frac{\sqrt{dk}}{w_{\min}\varepsilon}\right)^{\frac{1}{1-a}} + \frac{k^{\frac{3}{2}}}{\alpha w_{\min}\varepsilon}\right) \cdot \text{polylog}\left(\frac{dkR(\frac{\sigma_{\max}}{\sigma_{\min}})}{\alpha\beta\varepsilon\delta}\right)$$

*(where $a > 0$ is an arbitrarily small constant as in Theorem 3.2) samples from an unknown mixture of $k$ Gaussians $\mathcal{D}$ in $\mathbb{R}^d$ satisfying (1) and (2), where $w_{\min} = \min_i w_i$, and $(\alpha, \beta)$-learns $\mathcal{D}$.*

---

**Algorithm 1:** Private Gaussian Mixture Partitioning $\text{RPGMP}(X; k, R, w_{\min}, \sigma_{\min}, \sigma_{\max}, \varepsilon, \delta)$

---

**Input:** Dataset $X \in \mathbb{R}^{n \times d}$ coming from a mixture of at most $k$ Gaussians, such that each $x_i \in X$
      has length $\leq O(R + \sigma_{\max}\sqrt{d})$. Privacy parameters $\varepsilon, \delta > 0$; failure probability $\beta > 0$.
**Output:** Partition of $X$ into clusters.

1. If $k = 1$ Skip to last step (#8)
2. Find a small ball that contains $X$, and bound the range of points to within that ball:
   Set $n' \leftarrow |X| + \text{Lap}(2/\varepsilon) - \frac{n w_{\min}}{20}$.
   $B_{r''}(p) \leftarrow \text{PGLOC}(X, n'; \frac{\varepsilon}{2}, \delta, R, \sigma_{\min}, \sigma_{\max})$.
   Set $r \leftarrow 12 r''$.
   Set $X \leftarrow X \cap B_r(p)$.
3. Find 5-`TerrificBall` in $X$ with $t = \frac{n w_{\min}}{2}$:
   $B_{r'}(p') \leftarrow \text{PTERRIFICBALL}(X, \frac{n w_{\min}}{2}, c = 5, largest = \text{FALSE}; \varepsilon, \delta, r, \frac{\sqrt{d}\sigma_{\min}}{2})$.
4. If the data is separable already, we recurse on each part.
   If $B_{r'}(p') \neq \perp$ then partition $X$ into $A = X \cap B_{r'}(p')$ and $B = X \setminus B_{5r'}(p')$ and return

   $\text{RPGMP}(A; k-1, r, w_{\min}, \sigma_{\min}, \sigma_{\max}, \varepsilon, \delta) \cup \text{RPGMP}(B; k-1, r, w_{\min}, \sigma_{\min}, \sigma_{\max}, \varepsilon, \delta)$

5. Find a private $k$-PCA of $X$: $\Pi \leftarrow k$-PCA of $X^T X + N$ where $N$ is a symmetric matrix
   whose entries are taken from $\mathcal{N}(0, \frac{4r^4 \ln(2/\delta)}{\varepsilon^2})$.
6. Find 5-`TerrificBall` in $X\Pi$ with $t = \frac{n w_{\min}}{2}$:
   $B_{r'}(p') \leftarrow \text{PTERRIFICBALL}(X\Pi, \frac{w_{\min}}{2}, c = 5, largest = \text{TRUE}; \varepsilon, \delta, r, \frac{\sqrt{k}\sigma_{\min}}{2})$.
7. If the projected data is separable, we recurse on each part.
   If $B_{r'}(p') \neq \perp$ then partition $X$ into $A = \{x_i \in X : \Pi x_i \in B_{r'}(p')\}$ and
   $B = \{x_i \in X : \Pi x_i \notin B_{5r'}(p')\}$ and return

   $\text{RPGMP}(A; k-1, r, w_{\min}, \sigma_{\min}, \sigma_{\max}, \varepsilon, \delta) \cup \text{RPGMP}(B; k-1, r, w_{\min}, \sigma_{\min}, \sigma_{\max}, \varepsilon, \delta)$

8. Since the data isn't separable, we treat it as a single Gaussian.
   Set a single cluster: $C \leftarrow \{i : x_i \in X\}$ and return the singleton $\{C\}$.

---

**Algorithm 2:** Privately Learn Gaussian Mixture $\text{PGME}(X; k, R, w_{\min}, \sigma_{\min}, \sigma_{\max}, \varepsilon, \delta, \beta)$

---

**Input:** Dataset $X \in \mathbb{R}^{n \times d}$ coming from a $k$-Gaussian mixture model. Privacy parameters $\varepsilon, \delta > 0$;
      failure probability $\beta > 0$.
**Output:** Model Parameters Estimation

1. Truncate the dataset so that for all points, $\|X_i\|_2 \leq O(R + \sigma_{\max}\sqrt{d})$
2. $\{C_1, .., C_k\} \leftarrow \text{RPGMP}(X; k, R, w_{\min}, \sigma_{\min}, \sigma_{\max}, \varepsilon, \delta)$
3. For $j$ from 1 to $k$: let $(\mu_j, \Sigma_j) \leftarrow \text{PGE}(\{x_i : i \in C_j\}; R, \sigma_{\min}, \sigma_{\max}, \varepsilon, \delta)$ and
   $\tilde{n}_j \leftarrow |C_j| + \text{Lap}(1/\varepsilon)$.
4. Set weights such that for all $j$, $w_j \leftarrow \tilde{n}_j / (\sum_j \tilde{n}_j)$.
5. Return $\langle \mu_j, \Sigma_j, w_j \rangle_{j=1}^k$

---

## 5  Sample and Aggregate

In this section, we detail methods based on sample and aggregate, and derive their sample complexity. This will serve as a baseline for comparison with our methods. In short, the method repeatedly runs a non-private learning algorithm, and aggregates the results using the 1-cluster algorithms of [57, 56]. A similar sample and aggregate method was considered in [55], but they focused on a restricted case

(when all mixing weights are equal, and all components are spherical with a known variance), and did not explore certain considerations (i.e., how to minimize the impact of a large domain). We provide a more in-depth exploration and attempt to optimize the sample complexity.

The main advantage of the sample and aggregate method we describe here is that it is extremely flexible: given any non-private algorithm for learning mixtures of Gaussians, it can immediately be converted to a private method. However, there are a few drawbacks, which our main algorithm avoids. First, by the nature of the approach, it will increase the sample complexity multiplicatively by $\Omega(\sqrt{d}/\varepsilon)$, thus losing any chance of the non-private sample complexity being the dominating term in any parameter regime. Second, it is not clear on how to adapt this method to non-spherical Gaussians, since the methods of [57, 56] can only find $\ell_2$-balls containing many points, rather than the ellipsoids as required by non-spherical Gaussians. We consider aggregation methods which can handle settings where the required metric is unknown to be a very interesting direction for further study. Our main sample-and-aggregate meta-theorem is the following.

**Theorem 5.1** (Informal)**.** *Let* $m = \tilde{\Theta}\big(\frac{\sqrt{kd}+k^{1.5}}{\varepsilon}\log^2(1/\delta) \cdot 2^{O(\log^*(\frac{dR\sigma_{\max}}{\alpha\sigma_{\min}}))}\big)$. *Given a non-private algorithm which learns a mixture of separated spherical Gaussians with $n$ samples, there exists an $(\varepsilon, \delta)$-differentially private algorithm which learns the same mixture with $O(mn)$ samples.*

Combining with results from [64], this implies the following private learning algorithm.

**Theorem 5.2** (Informal)**.** *There exists a $(\varepsilon, \delta)$-differentially private algorithm learns a mixture of spherical Gaussians with the separation condition that $\|\mu_i - \mu_j\|_2 \geq (\sigma_i + \sigma_j) \cdot \tilde{\Omega}(k^{1/4} \cdot \text{poly} \log(k, d, 1/\varepsilon, \log(1/\delta), \log^*(\frac{R\sigma_{\max}}{\alpha\sigma_{\min}})))$. The number of samples it requires is*

$$n = \tilde{O}\left(\frac{\sqrt{kd}+k^{1.5}}{\varepsilon}\log^2(1/\delta) \cdot 2^{O(\log^*(\frac{dR\sigma_{\max}}{\alpha\sigma_{\min}}))}\left(\frac{d^3}{w_{\min}^2} + \frac{d}{w_{\min}\alpha^2} + \frac{k^2}{\alpha^2}\right)\right).$$

## Acknowledgments

Part of this work was done while the authors were visiting the Simons Institute for Theoretical Computer Science. Parts of this work were done while GK was supported as a Microsoft Research Fellow, as part of the Simons-Berkeley Research Fellowship program, while visiting Microsoft Research, Redmond, and while supported by a University of Waterloo startup grant. This work was done while OS was affiliated with the University of Alberta. OS gratefully acknowledges the Natural Sciences and Engineering Research Council of Canada (NSERC) for its support through grant #2017-06701. JU and VS were supported by NSF grants CCF-1718088, CCF-1750640, and CNS-1816028.

## Footnotes

[1]We remark that there are also many popular *heurstics* for learning Gaussian mixtures, notably the EM algorithm [24], but in this work we focus on algorithms with provable guarantees.

[2]To provide context, one might settle for a weaker goal of *proper learning* where the goal is merely to learn *some* Gaussian mixture, possibly with a different number of components, that is close to the true mixture, or *improper learning* where it suffices to learn *any* such distribution.

[3]These boundedness conditions are also provably necessary to learn even a single univariate Gaussian for pure differential privacy, concentrated differential privacy, or Rényi differential privacy, by the argument of [50]. One could only hope to avoid boundedness using the most general formulation of $(\varepsilon, \delta)$-differential privacy.

[4]Since [57, 56] call the ball found by their algorithm a *good* ball, we call ours a *terrific* ball.

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
