[Supplementary Material]

# Differentially Private Algorithms for Learning Mixtures of Separated Gaussians

Gautam Kamath[*]     Or Sheffet[†]     Vikrant Singhal[‡]     Jonathan Ullman[§]

October 15, 2019

## Abstract

Learning the parameters of Gaussian mixture models is a fundamental and widely studied problem with numerous applications. In this work, we give new algorithms for learning the parameters of a high-dimensional, well separated, Gaussian mixture model subject to the strong constraint of differential privacy. In particular, we give a differentially private analogue of the algorithm of Achlioptas and McSherry. Our algorithm has two key properties not achieved by prior work: (1) The algorithm's sample complexity matches that of the corresponding non-private algorithm up to lower order terms in a wide range of parameters. (2) The algorithm does not require strong *a priori* bounds on the parameters of the mixture components.

---

[*]Cheriton School of Computer Science, University of Waterloo. `g@csail.mit.edu`.

[†]Department of Computer Science, Bar Ilan University. `or.sheffet@biu.ac.il`.

[‡]Khoury College of Computer Sciences, Northeastern University. `singhal.vi@husky.neu.edu`.

[§]Khoury College of Computer Sciences, Northeastern University. `jullman@ccs.neu.edu`.

# Contents

# 1 Introduction

The *Gaussian mixture model* is one of the most important and widely studied models in Statistics—with roots going back over a century [Pea94]—and has numerous applications in the physical, life, and social sciences. In a Gaussian mixture model, we suppose that each sample is drawn by randomly selecting from one of $k$ distinct Gaussian distributions $G_1, \ldots, G_k$ in $\mathbb{R}^d$ and then drawing a sample from that distribution. The problem of *learning a Gaussian mixture model* asks us to take samples from this distribution and approximately recover the parameters (mean and covariance) of each of the underlying Gaussians. The past decades have seen tremendous progress towards understanding both the sample complexity and computational complexity of learning Gaussian mixtures [Das99, DS00, AK01, VW02, AM05, CR08b, CR08a, KK10, AS12, RV17, HL18a, DKS18, KSS18, KMV10, MV10, BS10, HK13, ABG⁺14, BCMV14, HP15, GHK15, XHM16, DTZ17, ABDH⁺18, FOS06, FOS08, DK14, SOAJ14, DKK⁺16, LS17, CDSS14].

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

 [NSV16, NS18]. Lastly, we apply a simplified version of [KLSU19] (Appendix C) to learn each component's parameters.

*Phase II.* We then consider the general case where the Gaussians can be non-spherical and wildly different from each other. In this case, if we directly add noise to the covariance matrix to achieve privacy, then the noise will scale polynomially with $\sigma_{\max}/\sigma_{\min}$, which is undesirable. To deal with the general case, we develop a recursive algorithm, which repeatedly identifies an *secluded* cluster in the data, and then recurses on this isolated cluster and the points outside of the cluster. To that end we develop in Section 5.1 a variant of the private clustering algorithm of [NSV16, NS18] that finds a secluded ball—a set of points that lie inside of some ball $B_r(p)$ such that the annulus $B_{10r}(p) \setminus B_r(p)$ is (essentially) empty.[4]

We obtain a recursive algorithm in the following way. First we try to find a secluded ball in the unprojected data. If we find one then we can split and recurse on the inside and outside of the ball. If we cannot find a ball, then we can argue that the diameter of the dataset is $\mathrm{poly}(d, k, \sigma_{\max})$. In the latter case, we can ensure that with $\mathrm{poly}(d, k)$ samples, the PCA-projection of the data preserves the mean of each component up to $O(\sqrt{k}\sigma_{\max})$, which guarantees that the cluster with the largest variance is secluded, so we can find the secluded ball and recurse.

## 1.3 Related Work

There has been a great deal of work on learning mixtures of distribution classes, particularly mixtures of Gaussians. There are many ways the objective can be defined, including clustering [Das99, DS00, AK01, VW02, AM05, CR08b, CR08a, KK10, AS12, RV17, HL18a, DKS18, KSS18], parameter estimation [KMV10, MV10, BS10, HK13, ABG+14, BCMV14, HP15, GHK15, XHM16, DTZ17, ABDH+18], proper learning [FOS06, FOS08, DK14, SOAJ14, DKK+16, LS17], and improper learning [CDSS14].

Some work on privately learning mixtures of Gaussians includes [NRS07] and [BKSW19]. The former introduced the sample-and-aggregate method to convert non-private algorithms into private algorithms, and applied it to learning mixtures of Gaussians. Our sample-and-aggregate-based method can be seen as a modernization of their algorithm, using tools developed over the last decade to handle somewhat more general settings and importantly, reduce the dependence on the range of the parameter space. As discussed above, our main algorithm improves upon this approach by avoiding an increase in the dependence on the dimension, allowing us to match the sample complexity of the non-private algorithm in certain parameter regimes. The latter paper (which is concurrent with the present work) provides a general method to convert from a *cover* for a class of distributions to a private learning algorithm for the same class. The work gets a near-optimal sample complexity of $\tilde{O}\left(kd^2\left(1/\alpha^2 + 1/\alpha\varepsilon\right)\right)$, but the algorithms have exponential running time in both $k$ and $d$ and their learning guarantees are incomparable to ours (the perform proper learning, while we do clustering and parameter estimation).

Other highly relevant works in private distribution learning include [KV18, KLSU19], which focus on learning a single Gaussian. There are also algorithms for learning structured univariate distributions in TV-distance [DHS15], and learning arbitrary univariate distributions in Kolomogorov distance [BNSV15]. Upper and lower bounds for learning the mean of a product distribution over the hypercube in $\ell_\infty$-distance include [BDMN05, BUV14, DMNS06, SU17]. [AKSZ18] focuses on estimating properties of a distribution, rather than the distribution itself. [Smi11] gives an algorithm which allows one to estimate asymptotically normal statistics with minimal increase in the sample complexity. There has also been a great deal of work on distribution learning in the local model of differential privacy [DJW13, WHW+16, KBR16, ASZ19, DR18, JKMW19, YB18, GRS19].

Within differential privacy, there are many algorithms for tasks that are related to learning mixtures of Gaussians, notably PCA [BDMN05, KT13, CSS13, DTTZ14] and clustering [NRS07, GLM+10, NSV16, NS18, BDL+17, SK18, HL18b]. Applying these algorithms naïvely to the problem of learning Gaussian mixtures would necessarily introduce a polynomial dependence on the range of the data, which we seek to avoid. Nonetheless, private algorithms for PCA and clustering feature prominently in our solution, and build directly on these works.

## 2 Preliminaries

### 2.1 General Preliminaries

Let $\mathsf{Sym}_d^+$ denote set of all $d \times d$, symmetric, and positive semidefinite matrices. Let $\mathcal{G}(d) = \{\mathcal{N}(\mu, \Sigma) : \mu \in \mathbb{R}^d, \Sigma \in \mathsf{Sym}_d^+\}$ be the family of $d$-dimensional Gaussians. We can now define the class $\mathcal{G}(d, k)$ of mixtures of Gaussians as follows.

**Definition 2.1** (Gaussian Mixtures). The class of *Gaussian k-mixtures in* $\mathbb{R}^d$ is

$$\mathcal{G}(d,k) := \left\{ \sum_{i=1}^k w_i G_i : G_1, \ldots, G_k \in \mathcal{G}(d), w_1, \ldots, w_k > 0, \sum_{i=1}^k w_i = 1 \right\}.$$

We can specify a Gaussian mixture by a set of $k$ tuples as: $\{(\mu_1, \Sigma_1, w_1), \ldots, (\mu_k, \Sigma_k, w_k)\}$, where each tuple represents the mean, covariance matrix, and mixing weight of one of its compo-

nents. Additionally, for each $i$, we refer to $\sigma_i^2 = \|\Sigma_i\|_2$ as the maximum directional variance of component $i$.

We are given $n$ points in $d$ dimensions in the form of a $(n \times d)$-matrix $X$, and use $A$ to denote the corresponding matrix of centers. $(\alpha, \beta, \varepsilon, \delta)$ are the parameters corresponding to accuracy in estimation (in total variation distance), failure probability, and privacy parameters, respectively. $R$ denotes the radius of a ball (centered at the origin) which contains all means, and $\kappa$ is the ratio of the upper and the lower bound on the variances (for simplicity, we generally assume the lower bound is 1). Also, $\sigma_{\max}^2$ and $\sigma_{\min}^2$ are the maximum and minimum variance of any single component, namely $\sigma_{\max}^2 = \max_i\{\sigma_i^2\}$ and $\sigma_{\min}^2$ is defined symmetrically; similarly $w_{\min}$ denotes a lower bound on the minimum mixing weight. We will use the notational convention that $B_r^d(\vec{c})$ denotes the ball in $\mathbb{R}^d$ of radius $r$ centered at $\vec{c} \in \mathbb{R}^d$. As $d$ will typically be clear from context, we will often suppress $d$ and write $B_r(\vec{c})$.

In order to (privately) learn a Gaussian mixture, we will need to impose two types of conditions on its parameters—boundedness, and separation.

**Definition 2.2** (Separated and Bounded Mixtures). For $s > 0$, a Gaussian mixtures $\mathcal{D} \in \mathcal{G}(d,k)$ is *s-separated* if

$$\forall 1 \le i < j \le k, \quad \left\|\mu_i - \mu_j\right\|_2 \ge s \cdot \max\{\sigma_i, \sigma_j\}.$$

For $R, \sigma_{\max}, \sigma_{\min}, w_{\min} > 0$, a Gaussian mixture $\mathcal{D} \in \mathcal{G}(d,k)$ is $(R, \sigma_{\min}, \sigma_{\max}, w_{\min})$-*bounded* if

$$\forall 1 \le i \le k, \|\mu_i\|_2 \le R, \quad \min_i\{\sigma_i\} \ge \sigma_{\min}, \quad \max_i\{\sigma_i\} \le \sigma_{\max}, \quad \text{and} \quad \min_i\{w_i\} = w_{\min}.$$

We denote the family of separated and bounded Gaussian mixtures by $\mathcal{G}(d, k, \sigma_{\min}, \sigma_{\max}, R, w_{\min}, s)$.

We now have now established the necessary definitions to define what it means to "learn" a Gaussian mixture in our setting.

**Definition 2.3** $((\alpha, \beta)$-Learning). Let $\mathcal{D} \in \mathcal{G}(d,k)$ be parameterized by $\{(\mu_1, \Sigma_1, w_1), \ldots, (\mu_k, \Sigma_k, w_k)\}$. We say that an algorithm $(\alpha, \beta)$-*learns* $\mathcal{D}$, if on being given sample-access to $\mathcal{D}$, it outputs with probablity at least $1 - \beta$ a distribution $\widehat{\mathcal{D}} \in \mathcal{G}(d,k)$ parameterized by $\{(\widehat{\mu}_1, \widehat{\Sigma}_1, \widehat{w}_1), \ldots, (\widehat{\mu}_k, \widehat{\Sigma}_k, \widehat{w}_k)\}$, such that there exists a permutation $\pi : [k] \to [k]$, for which the following conditions hold.

1. For all $1 \le i \le k$, $\mathrm{d}_{\mathrm{TV}}(\mathcal{N}(\mu_i, \Sigma_i), \mathcal{N}(\widehat{\mu}_{\pi(i)}, \widehat{\Sigma}_{\pi(i)})) \le O(\alpha)$.

2. For all $1 \le i \le k$, $\left|w_i - \widehat{w}_{\pi(i)}\right| \le O\left(\frac{\alpha}{k}\right)$.

Note that the above two conditions together imply that $\mathrm{d}_{\mathrm{TV}}(\mathcal{D}, \widehat{\mathcal{D}}) \le \alpha$.

### 2.1.1 Labelled Samples

In our analysis, it will be useful to think of sampling from a Gaussian mixture by the following two-step process: first we select a mixture component $\rho_i$ where $\rho_i = j$ with probability $w_j$, and then we choose a sample $X_i \sim G_{\rho_i}$. We can then imagine that each point $X_i$ in the sample has a *label* $\rho_i$ indicating which mixture component it was sampled from.

**Definition 2.4** (Labelled Sample). For $\mathcal{D} \in \mathcal{G}(d, k, \sigma_{\min}, \sigma_{\max}, R, w_{\min}, s)$, a *labelled sample* is a set of tuples $X^L = ((X_1, \rho_1), \ldots, (X_m, \rho_m))$ sampled from $\mathcal{D}$ according to the process above. We will write $X = (X_1, \ldots, X_m)$ to denote the (unlabelled) sample.

We emphasize again that the algorithm does not have access to the labelled sample $X^L$. In fact given a fixed sample $X$, *any* labelling has non-zero probability of occurring, so from the algorithm's perspective $X^L$ is not even well defined. Nonetheless, the labelled sample is a well defined and useful construct for the analysis of our algorithm, since it allows us to make sense of the statement that the algorithm with high probability correctly labels each point in the sample by which mixture component it came from.

### 2.1.2 Deterministic Regularity Conditions

In order to analyze our algorithm, it will be useful to establish several regularity conditions that are satisfied (with high probability) by samples from a Gaussian mixture. In this section we will state the following regularity conditions.

The first condition asserts that each mixture component is represented approximately the right number of times.

**Condition 2.5.** *Let $X^L = ((X_1, \rho_1), \ldots, (X_n, \rho_n))$ be a labelled sample from a Gaussian k-mixture $\mathcal{D}$. For every label $1 \le u \le k$,*

1. *the number of points from component $u$ (i.e. $|\{X_i : \rho_i = u\}|$) is in $[\frac{nw_u}{2}, \frac{3nw_u}{2}]$, and*

2. *if $w_u \ge 4\alpha/9k$ then the number of points from component $u$ is in $[n(w_u - \frac{\alpha}{9k}), n(w_u + \frac{\alpha}{9k})]$.*

In Appendix A, we prove the following lemma, which states that if the number of samples is sufficiently large, then with high probability each of the above conditions is satisfied.

**Lemma 2.6.** *Let $X^L = ((X_1, \rho_1), \ldots, (X_n, \rho_n))$ be a labelled sample from a Gaussian k-mixture $\mathcal{D} \in \mathcal{G}(d, k)$ with parameters $\{(\mu_1, \Sigma_1, w_1), \ldots, (\mu_k, \Sigma_k, w_k)\}$. If*

$$n \ge \max\left\{\frac{12}{w_{\min}} \ln(2k/\beta), \frac{405k^2}{2\alpha^2} \ln(2k/\beta)\right\},$$

*then with probability at least $1 - \beta$, $X^L$ (alternatively X) satisfies Condition 2.5.*

## 2.2 Privacy Preliminaries

In this section we review the basic definitions of differential privacy, and develop the algorithmic toolkit that we need.

### 2.2.1 Differential Privacy

**Definition 2.7** (Differential Privacy (DP) [DMNS06]). A randomized algorithm $M : \mathcal{X}^n \to \mathcal{Y}$ satisfies $(\varepsilon, \delta)$- differential privacy ($(\varepsilon, \delta)$-DP) if for every pair of neighboring datasets $X, X' \in \mathcal{X}^n$ (i.e., datasets that differ in exactly one entry),

$$\forall Y \subseteq \mathcal{Y} \quad \mathbb{P}[M(X) \in Y] \le e^\varepsilon \cdot \mathbb{P}[M(X') \in Y] + \delta.$$

Two useful properties of differential privacy are closure under post-processing and composition.

**Lemma 2.8** (Post Processing [DMNS06]). *If $M : \mathcal{X}^n \to \mathcal{Y}$ is $(\varepsilon, \delta)$-DP, and $P : \mathcal{Y} \to \mathcal{Z}$ is any randomized function, then the algorithm $P \circ M$ is $(\varepsilon, \delta)$-DP.*

**Lemma 2.9** (Composition of DP [DMNS06, DRV10]). *If $M_1, \ldots, M_T$ are $(\varepsilon_0, \delta_0)$-DP algorithms, and*

$$M(X) = (M_1(X), \ldots, M_T(X))$$

*is the composition of these mechanisms, then $M$ is $(\varepsilon, \delta)$-DP for*

- *(basic composition) $\varepsilon = \varepsilon_0 T$ and $\delta = \delta_0 T$;*

- *(advanced composition) $\varepsilon = \varepsilon_0 \sqrt{2T \log(1/\delta')} + \varepsilon_0 (e^{\varepsilon_0} - 1)T$ and $\delta = \delta_0 T + \delta'$, for any $\delta' > 0$.*

*Moreover, this property holds even if $M_1, \ldots, M_T$ are chosen adaptively.*

### 2.2.2 Basic Differentially Private Mechanisms.

We first state standard results on achieving privacy via noise addition proportional to sensitivity [DMNS06].

**Definition 2.10** (Sensitivity). *Let $f : \mathcal{X}^n \to \mathbb{R}^d$ be a function, its $\ell_1$-sensitivity and $\ell_2$-sensitivity are*

$$\Delta_{f,1} = \max_{X \sim X' \in \mathcal{X}^n} \|f(X) - f(X')\|_1 \quad \text{and} \quad \Delta_{f,2} = \max_{X \sim X' \in \mathcal{X}^n} \|f(X) - f(X')\|_2$$

*respectively. Here, $X \sim X'$ denotes that $X$ and $X'$ are neighboring datasets (i.e., those that differ in exactly one entry).*

For functions with bounded $\ell_1$-sensitivity, we can achieve $(\varepsilon, 0)$-DP by adding noise from a Laplace distribution proportional to $\ell_1$-sensitivity. For functions taking values in $\mathbb{R}^d$ for large $d$ it is more useful to add noise from a Gaussian distribution proportional to the $\ell_2$-sensitivity, achieving $(\varepsilon, \delta)$-DP.

**Lemma 2.11** (Laplace Mechanism). *Let $f : \mathcal{X}^n \to \mathbb{R}^d$ be a function with $\ell_1$-sensitivity $\Delta_{f,1}$. Then the Laplace mechanism*

$$M(X) = f(X) + \text{Lap}\left(\frac{\Delta_{f,1}}{\varepsilon}\right)^{\otimes d}$$

*satisfies $(\varepsilon, 0)$-DP.*

One application of the Laplace mechanism is private counting. The function $\text{PCOUNT}_\varepsilon(X, T)$ is $\varepsilon$-DP, and returns the number of points of $X$ that lie in $T$, i.e., $|X \cap T| + \text{Lap}(1/\varepsilon)$. The following is immediate since the statistic $|X \cap T|$ is 1-sensitive, and by tail bounds on Laplace random variables.

**Lemma 2.12.** *$\text{PCOUNT}$ is $\varepsilon$-DP, and with probability at least $1 - \beta$, outputs an estimate of $|X \cap T|$ which is accurate up to an additive $O\left(\frac{\log(1/\beta)}{\varepsilon}\right)$.*

**Lemma 2.13** (Gaussian Mechanism). *Let $f : \mathcal{X}^n \to \mathbb{R}^d$ be a function with $\ell_2$-sensitivity $\Delta_{f,2}$. Then the Gaussian mechanism*

$$M(X) = f(X) + \mathcal{N}\left(0, \left(\frac{\Delta_{f,2}\sqrt{2\ln(2/\delta)}}{\varepsilon}\right)^2 \cdot \mathbb{I}_{d\times d}\right)$$

*satisfies $(\varepsilon, \delta)$-DP.*

Now consider a setting where there are multiple queries to be performed on a dataset, but we only want to know if there is a query whose answer on the dataset lies above a certain threshold. We introduce a differentially private algorithm from [DR14, DNR$^+$09] which does that.

**Theorem 2.14** (Above Threshold). *Suppose we are given a dataset $X$, a sequence of queries $f_1, \ldots, f_t$, with $\ell_1$-sensitivity $\Delta$, and a threshold $T$. There exists an $\varepsilon$-differentially private algorithm $\textsc{AboveThreshold}_\varepsilon$ that outputs a stream of answers $a_1, \ldots, a_{t'} \in \{\bot, \top\}$, where $t' \leq t$, such that if $a_{t'} = \top$, then $a_1, \ldots, a_{t'-1} = \bot$. Then the following holds with probability at least $1 - \beta$. If $a_{t'} = \top$, then*

$$f_{t'}(X) \geq T - \Gamma,$$

*and for all $1 \leq i \leq t'$, if $a_i = \bot$, then*

$$f_i(X) \leq T + \Gamma,$$

*where*

$$\Gamma = \frac{8\Delta(\log t + \log(2/\beta))}{\varepsilon}.$$

## 2.3   Technical Preliminaries

**Lemma 2.15** (Hanson-Wright inequality [HW71]). *Let $X \sim \mathcal{N}(0, \mathbb{I}_{d\times d})$ and let $A$ be a $d \times d$ matrix. Then for all $t > 0$, the following two bounds hold:*

$$\mathbb{P}\left[X^T A X - \operatorname{tr}(A) \geq 2\|A\|_F \sqrt{t} + 2\|A\|_2 t\right] \leq \exp(-t);$$

$$\mathbb{P}\left[X^T A X - \operatorname{tr}(A) \leq -2\|A\|_F \sqrt{t}\right] \leq \exp(-t).$$

The following are standard concentration results for the empirical mean and covariance of a set of Gaussian vectors (see, e.g., [DKK$^+$16]).

**Lemma 2.16.** *Let $X_1, \ldots, X_n$ be i.i.d. samples from $\mathcal{N}(0, \mathbb{I}_{d\times d})$. Then we have that*

$$\mathbb{P}\left[\left\|\frac{1}{n}\sum_{i\in[n]} X_i\right\|_2 \geq t\right] \leq 4\exp(c_1 d - c_2 n t^2);$$

$$\mathbb{P}\left[\left\|\frac{1}{n}\sum_{i\in[n]} X_i X_i^T - I\right\|_2 \geq t\right] \leq 4\exp(c_3 d - c_4 n \min(t, t^2)),$$

*where $c_1, c_2, c_3, c_4 > 0$ are some absolute constants.*

We finally have a lemma that translates closeness of Gaussian distributions from one metric to another. It is a combination of Corollaries 2.13 and 2.14 of [DKK+16].

**Lemma 2.17.** *Let $\alpha > 0$ be smaller than some absolute constant. Suppose that*

$$\left\|\Sigma^{-1/2}(\mu - \widehat{\mu})\right\|_2 \leq O(\alpha) \quad and \quad \left\|\mathbb{I} - \Sigma^{-1/2}\widehat{\Sigma}\Sigma^{-1/2}\right\|_F \leq O(\alpha),$$

*where $\mathcal{N}(\mu, \Sigma)$ is a Gaussian distribution in $\mathbb{R}^d$, $\mu \in \mathbb{R}^d$, and $\widehat{\Sigma} \in \mathbb{R}^{d \times d}$ is a PSD matrix. Then*

$$d_{\mathrm{TV}}\left(\mathcal{N}(\mu, \Sigma), \mathcal{N}(\widehat{\mu}, \widehat{\Sigma})\right) \leq \alpha.$$

# 3 Robustness of PCA-Projection to Noise

One of the main tools used in learning mixtures of Gaussians under separation is Principal Component Analysis (PCA). In particular, it is common to project onto the top $k$ principal components (a subspace which will contain the means of the components). In some sense, this eliminates directions which do not contain meaningful information while preserving the distance between the means, thus allowing us to cluster with separation based on the "true" dimension of the data, $k$, rather than the ambient dimension $d$. In this section, we show that a similar statement holds, even after perturbations required for privacy.

Before showing the result for perturbed PCA, we reiterate the (very simple) proof of Achlioptas and McSherry [AM05]. Fixing a cluster $i$, denoting its empirical mean as $\bar{\mu}_i$, the mean of the resulting projection as $\hat{\mu}_i$, $\Pi$ as the $k$-PCA projection matrix, $u_i \in \{0,1\}^n$ as the vector indicating which datapoint was sampled from cluster $i$, and $n_i$ as the number of datapoints which were sampled from cluster $i$, we have

$$\|\bar{\mu}_i - \hat{\mu}_i\|_2 = \|\tfrac{1}{n_i}\left(X^T - (X\Pi)^T\right)u_i\|_2 \leq \|X - X\Pi\|_2 \tfrac{\|u_i\|_2}{n_i} \leq \tfrac{1}{\sqrt{n_i}}\|X - A\|_2,$$

where the last inequality follows from the $X\Pi$ being the best $k$-rank approximation of $X$ whereas $A$ is any rank-$k$ matrix. In particular, we could choose $A$ to be the matrix where each row of $X$ is replaced by the (unknown) center of the component which generated it (as we do in Lemma 3.1 below). We now extend this result to a perturbed $k$-PCA projection as given by the following lemma.

**Lemma 3.1.** *Let $X \in \mathbb{R}^{n \times d}$ be a collection of $n$ datapoints from $k$ clusters each centered at $\mu_1, \mu_2, ..., \mu_k$. Let $A \in \mathbb{R}^{n \times d}$ be the corresponding matrix of (unknown) centers (for each $j$ we place the center $\mu_{c(j)}$ with $c(j)$ denoting the clustering point $X_j$ belongs to). Let $\Pi_{V_k} \in \mathbb{R}^{d \times d}$ denote the $k$-PCA projection of $X$'s rows. Let $\Pi_U \in \mathbb{R}^{d \times d}$ be a projection such that for some bound $B \geq 0$ it holds that $\|X^T X - (X\Pi_U)^T(X\Pi_U)\|_2 \leq \|X^T X - (X\Pi_{V_k})^T(X\Pi_{V_k})\|_2 + B$. Denote $\bar{\mu}_i$ as the empirical mean of all points in cluster $i$ and denote $\hat{\mu}_i$ as the projection of the empirical mean $\hat{\mu}_i = \Pi_U \bar{\mu}_i$. Then*

$$\|\bar{\mu}_i - \hat{\mu}_i\|_2 \leq \tfrac{1}{\sqrt{n_i}}\|X - A\|_2 + \sqrt{\tfrac{B}{n_i}}$$

*Proof.* Fix $i$. Denote $u_i \in \mathbb{R}^n$ as the indicating vector of all datapoints in $X$ that belong to cluster $i$. Assuming there are $n_i$ points from cluster $i$ it follows that $\|u_i\|^2 = n_i$, thus $\tfrac{1}{\sqrt{n_i}}u_i$ is a unit-length

vector. Following the inequality $\text{tr}(AB) \le \|A\|_2\|B\|_F$ for PSD matrices, we thus have that

$$\left\|\bar{\mu}_i - \hat{\mu}_i\right\|_2^2 = \left\|\frac{1}{n_i}X^T u_i - \frac{1}{n_i}(X\Pi_U)^T u_i\right\|_2^2 = \frac{1}{n_i^2}\left\|(X(\mathbb{I}-\Pi_U))^T u_i\right\|_2^2$$

$$\le \frac{1}{n_i^2}\left\|(X(\mathbb{I}-\Pi_U))^T\right\|_2^2 \|u_i\|_2^2 \overset{(*)}{=} \frac{1}{n_i^2}\left\|(X(\mathbb{I}-\Pi_U))^T (X(\mathbb{I}-\Pi_U))\right\|_2 \cdot n_i$$

$$\le \frac{1}{n_i}\left(\left\|\left(X(\mathbb{I}-\Pi_{V_k})\right)^T \left(X(\mathbb{I}-\Pi_{V_k})\right)\right\|_2 + B\right) = \frac{1}{n_i}\left(\|X(\mathbb{I}-\Pi_{V_k})\left(X(\mathbb{I}-\Pi_{V_k})\right)^T\|_2 + B\right)$$

$$= \frac{1}{n_i}\left(\left\|XX^T - X\Pi_{V_k}\Pi_{V_k}^T X^T\right\|_2 + B\right) \overset{(**)}{=} \frac{1}{n_i}\left(\left\|X - X\Pi_{V_k}\right\|_2^2 + B\right) \overset{(***)}{\le} \frac{1}{n_i}\left(\|X - A\|_2^2 + B\right)$$

where the equality marked with $(*)$ follows from the fact that for any matrix $M$ we have $\|MM^T\|_2 = \|M^T M\|_2 = \sigma_1(M)^2$ with $\sigma_1(M)$ denoting $M$'s largest singular value; the equality marked with $(**)$ follows from the fact that for any matrix $M$ we have $\|MM^T - M\Pi_{V_k}M\|_2 = \sigma_{k+1}(M)^2$ with $\sigma_k(M)$ denoting $M$'s $(k+1)$th-largest singular value; and the inequality $(***)$ follows from the fact that $A$ is a rank-$k$ matrix. In this proof, we also used the fact that $\Pi$ is a projection matrix, implying that it is symmetric and equal to its square – these facts allow us to cancel various "cross terms." The inequality $\sqrt{a+b} \le \sqrt{a} + \sqrt{b}$ concludes the proof. $\qquad\square$

The above is a general statement for any type of clustering problem, which we instantiate in the following lemma for mixtures of Gaussians.

**Lemma 3.2.** *Let $X \in \mathbb{R}^{n \times d}$ be a sample from $\mathcal{D} \in \mathcal{G}(d,k)$, and let $A \in \mathbb{R}^{n \times d}$ be the matrix where each row $i$ is the (unknown) mean of the Gaussian from which $X_i$ was sampled. For each $i$, let $\sigma_i^2$ denote the maximum directional variance of component $i$, and $w_i$ denote its mixing weight. Define $\sigma^2 = \max_i\{\sigma_i^2\}$ and $w_{\min} = \min_i\{w_i\}$. If*

$$n \ge \frac{1}{w_{\min}}\left(\xi_1 d + \xi_2 \log\left(\frac{2k}{\beta}\right)\right),$$

*where $\xi_1, \xi_2$ are universal constants, then with probability at least $1 - \beta$,*

$$\frac{\sqrt{nw_{\min}}\sigma}{4} \le \|X - A\|_2 \le 4\sqrt{n\sum_{i=1}^k w_i\sigma_i^2}.$$

*Proof.* Let $C_i \in \mathbb{R}^{n_i \times d}$ be the matrix formed by concatenating the rows drawn from Gaussian component $i$, and let $A_i$ be the corresponding operation applied to $A$. Let $\Sigma_i$ denote the covariance matrix of Gaussian component $i$.

We first prove the lower bound on the norm. Let $C^*$ be the matrix in $\{C_1, \ldots, C_k\}$ with the largest direction variance (i.e., $C^* = C_i$, where $i = \arg\max_i \sigma_i^2$), let $A^*$ be the submatrix of $A$ corresponding to the same rows of $C^*$, and let $\Sigma$ be the covariance matrix of the Gaussian component corresponding to these rows. Then each row of $(C^* - A^*)\Sigma^{-\frac{1}{2}}$ is an independent sample from $\mathcal{N}(\vec{0}, \mathbb{I}_{d \times d})$.

We know that the number of rows in $C^*$ is at least $\frac{nw_{\min}}{2}$. Using Cauchy-Schwarz inequality, Theorem 5.39 of [Ver12], and our bound on $n$, we get that with probability at least $1 - \frac{\beta}{k}$.

$$\|C^* - A^*\|_2 \left\|\Sigma^{-\frac{1}{2}}\right\|_2 \geq \left\|(C^* - A^*)\Sigma^{-\frac{1}{2}}\right\|_2$$

$$\geq \sqrt{\frac{nw_{\min}}{2}} - C_1\sqrt{d} - C_2\sqrt{\log\left(\frac{2k}{\beta}\right)}$$

$$\geq \frac{\sqrt{nw_{\min}}}{4},$$

where $C_1$ and $C_2$ are absolute constants. Since $\left\|\Sigma^{\frac{1}{2}}\right\|_2 = \sigma$, we get,

$$\|C^* - A^*\|_2 \geq \frac{\sqrt{nw_{\min}}\sigma}{4}.$$

Since, the spectral norm of $X - A$ has to be at least the spectral norm of $C^* - A^*$, the lower bound holds.

Now, we prove the upper bound. For each $i$, the number of points in the submatrix $C_i$ is at most $\frac{3nw_i}{2}$. Using Cauchy-Schwarz inequality, Theorem 5.39 of [Ver12], and our bound on $n$ again, we get the following with probability at least $1 - \frac{\beta}{k}$.

$$\|C_i - A_i\|_2 = \left\|(C_i - A_i)\Sigma_i^{-\frac{1}{2}}\Sigma_i^{\frac{1}{2}}\right\|_2$$

$$\leq \left\|(C_i - A_i)\Sigma_i^{-\frac{1}{2}}\right\|_2 \left\|\Sigma_i^{\frac{1}{2}}\right\|_2$$

$$\leq \left(\sqrt{\frac{3nw_i}{2}} + C_1\sqrt{d} + C_2\sqrt{\log\left(\frac{2k}{\beta}\right)}\right)\left\|\Sigma_i^{\frac{1}{2}}\right\|_2$$

$$\leq 4\sqrt{nw_i}\sigma_i$$

Now,

$$\|X - A\|_2^2 = \left\|(X - A)^T(X - A)\right\|_2$$

$$= \left\|\sum_{i=1}^{k}(C_i - A_i)^T(C_i - A_i)\right\|_2$$

$$\leq \sum_{i=1}^{k}\left\|(C_i - A_i)^T(C_i - A_i)\right\|_2$$

$$\leq 16n\sum_{i=1}^{k}w_i\sigma_i^2.$$

The second equality can be seen by noting that each entry of $(X - A)^T(X - A)$ is the inner product of two columns of $X - A$ – by grouping terms in this inner product, it can be be seen as the sum of inner products of the corresponding columns of $C_i - A_i$, since the indices form a partition of the rows. The first inequality is the triangle inequality.

We finally apply the union bound over all $i$ to complete the proof. □

# 4   A Warm Up: Strongly Bounded Spherical Gaussian Mixtures

We first give an algorithm to learn mixtures of spherical Gaussians, whose means lie in a small ball, whose variances are within constant factor of one another, and whose mixing weights are identical. Before we get to that, we formally define such a family of mixtures of spherical Gaussians. Let

$$\mathcal{S}(d) = \{\mathcal{N}(\mu, \sigma^2 \mathbb{I}_{d \times d}) : \mu \in \mathbb{R}^d, \sigma^2 > 0\}$$

be the family of $d$-dimensional spherical Gaussians. As before, we can define the class $\mathcal{S}(d,k)$ of mixtures of spherical Gaussians as follows.

**Definition 4.1** (Spherical Gaussian Mixtures). The class of *Spherical Gaussian k-mixtures in $\mathbb{R}^d$* is

$$\mathcal{S}(d,k) := \left\{ \sum_{i=1}^{k} w_i G_i : G_1, \ldots, G_k \in \mathcal{S}(d), \sum_{i=1}^{k} w_i = 1 \right\}$$

Again, we will need to impose two types of conditions on its parameters—boundedness and separation—that are defined in slightly different ways from our initial definitions in Section 2.1. We also introduce another condition that says that all mixing weights are equal.

**Definition 4.2** (Separated, Bounded, and Uniform (Spherical) Mixtures). For $s > 0$, a spherical Gaussian mixture $\mathcal{D} \in \mathcal{S}(d,k)$ is *s-separated* if

$$\forall 1 \le i < j \le k, \quad \|\mu_i - \mu_j\|_2 \ge s \cdot \max\{\sigma_i, \sigma_j\}.$$

For $R, \sigma_{\max}, w_{\min} > 0$, a Gaussian mixture $\mathcal{D} \in \mathcal{S}(d,k)$ is *$(R, \sigma_{\min}, \kappa)$-bounded* if

$$\forall 1 \le i \le k, \|\mu_i\|_2 \le R, \quad \min_i\{\sigma_i\} = \sigma_{\min}, \quad \text{and} \quad \max_{i,j}\left\{ \frac{\sigma_i^2}{\sigma_j^2} \right\} \le \kappa.$$

A Gaussian mixture $\mathcal{D} \in \mathcal{S}(d,k)$ is *uniform* if

$$\forall 1 \le i \le k, \ w_i = \frac{1}{k}.$$

We denote the family of separated, bounded, and uniform spherical Gaussian mixtures by $\mathcal{S}(d, k, \sigma_{\min}, \kappa, s, R)$. We can specify a Gaussian mixture in this family by a set of $k$ tuples as: $\{(\mu_1, \sigma_1), \ldots, (\mu_k, \sigma_k)\}$, where each tuple represents the mean and standard deviation of one of its components.

**Definition 4.3.** We define the following family of separated, bounded, and uniform mixtures of spherical Gaussians that have similar variances and lie in a small ball around the origin.

$$\mathcal{S}(d, k, \kappa, s) \equiv \bigcup_{\sigma_{\min} > 0} \mathcal{S}(d, k, \sigma_{\min}, \kappa, s, k\sqrt{d}\kappa\sigma_{\min})$$

We define the quantity $s$ in the statement of the main theorem of this section. The above definition could be generalized to have the means lie in a small ball that is not centered at the origin, but because it is easy to privately find a small ball that would contain all the points, we can omit that for simplicity.

The following is our main theorem for this sub-class of mixtures, which quantifies the guarantees of Algorithm 1.

**Algorithm 1:** Private Easy Gaussian Mixture Estimator $\text{PEGME}_{\varepsilon,\delta,\alpha,\beta,\kappa,\sigma_{\min}}(X)$

---

**Input:** Samples $X_1,\ldots,X_{2n} \in \mathbb{R}^d$. Ratio of maximum and minimum variances: $\kappa$.
    Minimum variance of a mixture component: $\sigma_{\min}^2$. Parameters $\varepsilon,\delta,\alpha,\beta > 0$.
**Output:** A mixture of Gaussians $\widehat{G}$, such that $\mathrm{d}_{\mathrm{TV}}(G,\widehat{G}) \leq \alpha$.

Set parameters: $\Lambda \leftarrow 2k\sqrt{d\kappa}\sigma_{\min} \quad \Delta_{\varepsilon,\delta} \leftarrow \frac{2\Lambda^2\sqrt{2\ln(1.25/\delta)}}{\varepsilon} \quad \ell \leftarrow \max\left\{k, O\left(\log\left(\frac{n}{\beta}\right)\right)\right\}$

Throw away all $X_i \in X$ such that $\|X_i\|_2 > \Lambda$, and call this new dataset $X$ as well
Let $Y \leftarrow (X_1,\ldots,X_n)$ and $Z \leftarrow (X_{n+1},\ldots,X_{2n})$

```
// Privately run PCA
```
Let $E \in \mathbb{R}^{d\times d}$ be a symmetric matrix, where each entry $E_{i,j}$ for $j \geq i$ is an independent draw
 from from $\mathcal{N}(0,\Delta_{\varepsilon,\delta}^2)$
Let $\widehat{V_\ell}$ be the $\ell$-dimensional principal singular subspace of $Y^T Y + E$
Project points of $Z$ on to $\widehat{V_\ell}$ to get the set $Z'_\ell$
Rotate the space to align with the axes of $\widehat{V_\ell}$ to get the set $Z_\ell$ from $Z'_\ell$

```
// Privately locate individual components
```
Let $S_\ell \leftarrow Z_\ell$ and $i = 1$
**While** $S_\ell \neq \emptyset$ *and* $i \leq k$
    $(c_i,r'_i) \leftarrow \text{PGLOC}(S_\ell, \frac{n}{2k}; \frac{\varepsilon}{O(\sqrt{k\ln(1/\delta)})}, \frac{\delta}{2k}, R + 8\sqrt{\ell\kappa}\sigma_{\min}, \sigma_{\min}, \sqrt{\kappa}\sigma_{\min})$
    Let $r_i \leftarrow 4\sqrt{3}r'_i$ and $S_i \leftarrow S_\ell \cap B_{r_i}(c_i)$
    $S_\ell \leftarrow S_\ell \setminus S_i$
    $i \leftarrow i+1$
**If** $|C| < k$
    **Return** $\perp$
**For** $i \leftarrow 1,\ldots,k$
    Rotate $c_i$ back to get $\widehat{c_i}$
    Set $\widehat{r_i} \leftarrow r_i + 10\sqrt{\ell\kappa}\sigma_{\min} + 2r_i\sqrt{\frac{3d}{\ell}}$
    Let $\widehat{S_i}$ be points in $Z \cap B_{\widehat{r_i}}(\widehat{c_i})$, whose corresponding points lie in $Z_\ell \cap B_{r_i}(c_i)$

```
// Privately estimate each Gaussian
```
**For** $i \leftarrow 1,\ldots,k$
    $(\widehat{\mu_i},\widehat{\sigma_i^2}) \leftarrow \text{PSGE}(\widehat{S_i};\widehat{c_i},\widehat{r_i},\varepsilon,\delta)$
    $\widehat{G} \leftarrow \widehat{G} \cup \{(\widehat{\mu_i},\widehat{\sigma_i^2},\frac{1}{k})\}$
**Return** $\widehat{G}$

---

**Theorem 4.4.** *There exists an $(\varepsilon,\delta)$-differentially private algorithm, which if given $n$ independent samples from $\mathcal{D} \in \mathcal{S}(d,k,\kappa,C\sqrt{\ell})$, such that $C = \xi + 16\sqrt{\kappa}$, where $\xi,\kappa \in \Theta(1)$ and $\xi$ is a universal*

*constant, $\ell = \max\{512\ln(nk/\beta), k\}$, and*

$$n \geq O\left(\frac{dk}{\alpha^2} + \frac{d^{\frac{3}{2}}k^3\sqrt{\ln(1/\delta)}}{\varepsilon} + \frac{dk\sqrt{\ln(1/\delta)}}{\alpha\varepsilon} + \frac{\sqrt{d}k\ln(k/\beta)}{\alpha\varepsilon}\right) + n',$$

*where*

$$n' \geq O\left(\frac{k^2}{\alpha^2}\ln(k/\beta) + \frac{\ell^{\frac{5}{9}}k^{\frac{5}{3}}}{\varepsilon^{\frac{10}{9}}} \cdot \mathrm{polylog}\left(\ell, k, \frac{1}{\varepsilon}, \frac{1}{\delta}, \frac{1}{\beta}\right)\right),$$

*then it $(\alpha, \beta)$-learns $\mathcal{D}$.*

The algorithm itself is fairly simple to describe. First, we run a private version of PCA, and project to the top $k$ PCA directions. By Lemma 3.1, this will reduce the dimension from $d$ to $k$ while (approximately) preserving the separation condition. Next, we repeatedly run an algorithm which (privately) finds a small ball containing many points (essentially the 1-cluster algorithm of [NS18]) in order to cluster the points such that all points generated from a single Gaussian lie in the same cluster. Finally, for each cluster, we privately estimate the mean and variance of the corresponding Gaussian component.

## 4.1   Privacy

We will first analyze individual components of the algorithm, and then use composition (Lemma 2.9) to reason about privacy.

The PCA section of the algorithm is $(\varepsilon, \delta)$-DP with respect to $Y$. This holds because the $\ell_2$ sensitivity of the function $Y^T Y$ is $2\Lambda^2$, because all points are guaranteed to lie in a ball of radius $\Lambda$ around the origin, and by Lemma 2.13, we know that adding Gaussian noise proportional to $\Delta_{\varepsilon,\delta}$ is enough to have $(\varepsilon, \delta)$-DP.

In the second step, we run PGLOC (which is $(\varepsilon, \delta)$-differentially private) on $Z$ repeatedly with parameters $(\varepsilon/O(\sqrt{k\ln(1/\delta)}), \delta/2)$, after which we only perform computation on the output of this private algorithm So, the whole process, by advanced composition and post-processing (Lemmata 2.9 and 2.8), is $(\varepsilon, \delta)$-DP with respect to $Z$.

In the final step, we apply PSGE (which is $(\varepsilon, \delta)$-differentially private) with parameters $\varepsilon, \delta$ on disjoint datasets $\widehat{S_i}$. Therefore, this step is $(\varepsilon, \delta)$ private with respect to $Z$.

Finally, applying the composition lemma again, we have $(2\varepsilon, 2\delta)$-DP for $Z$. Combined with $(\varepsilon, \delta)$-DP for $Y$, and the fact that $X$ is the union of these two disjoint sets, we have $(2\varepsilon, 2\delta)$-DP for $X$. Rescaling the values of $\varepsilon$ and $\delta$ by 2 gives the desired result.

## 4.2   Accuracy

As indicated in our outline above, the algorithm is composed of three blocks: private PCA, isolating individual Gaussians, and learning the isolated Gaussians. We divide the proof of accuracy in a similar way.

### 4.2.1 Deterministic Regularity Conditions for Spherical Gaussians

We first give two regularity conditions for mixtures of spherical Gaussians in the family mentioned above.

The first condition asserts that each mixture component is represented by points that lie in a ball of approximately the right radius.

**Condition 4.5.** *Let $X^L = ((X_1, \rho_1), \ldots, (X_n, \rho_n))$ be a labelled sample from a Gaussian mixture $\mathcal{D} \in \mathcal{S}(\ell, k, \kappa, s)$, where $\ell \geq 512 \ln(nk/\beta)$ and $s > 0$. For every $1 \leq u \leq k$, the radius of the smallest ball containing the set of points with label $u$ (i.e. $\{X_i : \rho_i = u\}$) is in $[\sqrt{\ell}\sigma_u/2, \sqrt{3\ell}\sigma_u]$.*

The second condition says that if the means of the Gaussians are "far enough", then the inter-component distance (between points from different components) would also be large.

**Condition 4.6.** *Let $X^L = ((X_1, \rho_1), \ldots, (X_n, \rho_n))$ be a labelled sample from a Gaussian mixture $\mathcal{D} \in \mathcal{S}(\ell, k, \kappa, C\sqrt{\ell})$, where $\ell \geq \max\{512 \ln(nk/\beta), k\}$ and $C > 1$ is a constant. For every $\rho_i \neq \rho_j$,*

$$\left\| X_i - X_j \right\|_2 \geq \frac{C}{2} \sqrt{\ell} \max\{\sigma_{\rho_i}, \sigma_{\rho_j}\}.$$

The following lemma is immediate from from Lemmata 2.6 and A.3.

**Lemma 4.7.** *Let $Y$ and $Z$ be datasets sampled from $\mathcal{D} \in \mathcal{S}(d, k, \kappa, (\xi + 16\sqrt{\kappa})\sqrt{\ell})$ (with $Y^L$ and $Z^L$ being their respective labelled datasets) as defined within the algorithm, such that $d \geq \ell \geq \max\{512 \ln(nk/\beta), k\}$ where $\xi, \kappa \in \Theta(1)$ and $\xi > 1$ is a universal contant. If*

$$n \geq O\left( \frac{k^2}{\alpha^2} \ln(k/\beta) \right),$$

*then with probability at least $1 - 4\beta$, $Y^L$ and $Z^L$ (alternatively $Y$ and $Z$) satisfy Conditions 2.5 and 4.5.*

### 4.2.2 PCA

The following result stated in [DTTZ14], though used in a setting where $\Delta_{\varepsilon, \delta}$ was fixed, holds for any value of $\Delta_{\varepsilon, \delta}$.

**Lemma 4.8** (Theorem 9 of [DTTZ14]). *Let $\widehat{\Pi}_\ell$ be the top $\ell$ principal subspace obtained using $Y^T Y + E$ in Algorithm 1. Suppose $\Pi_\ell$ is the top $\ell$ subspace obtained from $Y^T Y$. Then with probability at least $1 - \beta$,*

$$\left\| Y^T Y - (Y\widehat{\Pi}_\ell)^T (Y\widehat{\Pi}_\ell) \right\|_2 \leq \left\| Y^T Y - (Y\Pi_\ell)^T (Y\Pi_\ell) \right\|_2 + O\left( \Delta_{\varepsilon, \delta} \sqrt{d} \right).$$

The main result here is that the PCA step shrinks the Gaussians down in a way that their means, after being projected upon the privately computed subspace, are close to their original locations.

**Lemma 4.9.** *Let $Y$ be the dataset, and $\widehat{V}_\ell$ be the subspace as defined in Algorithm 1. Suppose $\mu_1, \ldots, \mu_k$ are the means of the Gaussians, and $\mu'_1, \ldots, \mu'_k$ are their respective projections on to $\widehat{V}_\ell$. If*

$$n \geq O\left( \frac{d^{\frac{3}{2}} k^3 \sqrt{\ln(1/\delta)}}{\varepsilon} + k \ln(k/\beta) \right),$$

*and $Y$ satisfies Condition 2.5, then with probability at least $1 - 4\beta$,*

1. *for every $i$, we have $\left\|\mu_i - \mu_i'\right\|_2 \le 8\sqrt{\ell}\,\sigma_{\max}$, where $\sigma_{\max} = \max_i\{\sigma_i\}$, and*

2. *for every $i \ne j$, $\left\|\mu_i' - \mu_j'\right\|_2 \ge \left\|\mu_i - \mu_j\right\|_2 - 16\sqrt{\ell}\,\sigma_{\max}$.*

*Proof.* Let $\widetilde{\mu}_i$ be the empirical mean of component $i$ using the respective points in $Y$, and let $\dot{\mu}_i = \widehat{\Pi}_\ell \widetilde{\mu}_i$. Using Lemmata 3.1, 3.2, and 4.8, we know that with probability at least $1 - \beta$, for all $1 \le i \le k$,

$$\left\|\dot{\mu}_i - \widetilde{\mu}_i\right\|_2 \le 4\sqrt{2\sum_{j=1}^{k}\sigma_i^2 + O\left(\sqrt{\frac{\Delta_{\varepsilon,\delta}\sqrt{d}k}{n}}\right)}$$

$$\le 4\sqrt{2k}\,\sigma_{\max} + O\left(\sqrt{\frac{d^{\frac{3}{2}}k^3\sigma_{\min}^2\sqrt{\ln(1/\delta)}}{\varepsilon n}}\right). \qquad (\kappa \in \Theta(1))$$

Because $n \ge \frac{d^{\frac{3}{2}}k^3\sqrt{\ln(1/\delta)}}{\varepsilon}$ and $\frac{\sigma_{\max}^2}{\sigma_{\min}^2} \le \kappa \in \Theta(1)$, we have

$$O\left(\sqrt{\frac{d^{\frac{3}{2}}k^3\sigma_{\min}^2\sqrt{\ln(1/\delta)}}{\varepsilon n}}\right) \le O(\sigma_{\max})$$

$$\implies \left\|\dot{\mu}_i - \widetilde{\mu}_i\right\|_2 \le 6\sqrt{k}\,\sigma_{\max}$$

$$\le 6\sqrt{\ell}\,\sigma_{\max}.$$

Since $n \ge O\left(\frac{dk}{\ell} + \frac{k\ln(k/\beta)}{d} + \frac{k\ln(k/\beta)}{\ell}\right)$, using Lemma 2.16, we have that with probability at least $1 - 2\beta$, for all $1 \le i \le k$,

$$\left\|\widetilde{\mu}_i - \mu_i\right\|_2 \le \sqrt{\ell}\,\sigma_i \le \sqrt{\ell}\,\sigma_{\max} \quad \text{and} \quad \left\|\dot{\mu}_i - \mu_i'\right\|_2 \le \sqrt{\ell}\,\sigma_i \le \sqrt{\ell}\,\sigma_{\max}.$$

Finally, we get the required results by using triangle inequality. $\qquad\square$

### 4.2.3 Clustering

After the PCA step, individual Gaussian components shrink (i.e., the radius decreases from $O(\sqrt{d}\sigma)$ to $O(\sqrt{\ell}\sigma)$), but the means do not shift a lot. Given the large initial separation, we can find individual components using the private location algorithm, and learn them separately using a private learner. In this section, we show that our algorithm is able to achieve the first goal.

First, we prove that our data is likely to satisfy some of the conditions have have already defined.

**Lemma 4.10.** *Let $Y, Z, Z_\ell$ be datasets as defined within the algorithm (with $Z_\ell^L$ being the corresponding labeled dataset of $Z_\ell$), where $Y, Z$ are sampled from $\mathcal{D} \in \mathcal{S}(d, k, \kappa, (\xi + 16\sqrt{\kappa})\sqrt{\ell})$, such that $d \ge \ell \ge \max\{512\ln(nk/\beta), k\}$, and $\xi, \kappa \in \Theta(1)$ and $\xi > 1$ is a universal constant. If*

$$n \ge O\left(\frac{d^{\frac{3}{2}}k^3\sqrt{\ln(1/\delta)}}{\varepsilon} + \frac{k^2}{\alpha^2}\ln(k/\beta)\right),$$

*and Y and Z satisfy Condition 2.5, then with probability at least $1 - 7\beta$, $Z_\ell^L$ (alternatively $Z_\ell$) satisfies Conditions 2.5, 4.5, and 4.6.*

*Proof.* Let $\mu_1, \dots, \mu_k$ be means of the Gaussians in $\mathcal{D}$, and $\mu_1', \dots, \mu_k'$ be their respective projections onto $\widehat{V}_\ell$, and let $\sigma_1^2, \dots, \sigma_k^2$ be their respective variances. Because $Y$ satisfies Condition 2.5, we know from Lemma 4.9 that with probability at least $1 - 4\beta$, for each $1 \leq i \neq j \leq k$

$$
\begin{aligned}
\left\| \mu_i' - \mu_j' \right\|_2 &\geq \left\| \mu_i - \mu_j \right\|_2 - 16\sqrt{\ell}\,\sigma_{\max} \\
&\geq (\xi + 16\sqrt{\kappa})\sqrt{\ell}\,\max\{\sigma_i, \sigma_j\} - 16\sqrt{\ell}\,\sigma_{\max} \\
&\geq \xi\sqrt{\ell}\,\max\{\sigma_i, \sigma_j\}.
\end{aligned}
$$

Therefore, points in $Z_\ell$ are essentially points from some $\mathcal{D}' \in \mathcal{S}(\ell, k, \kappa, \xi\sqrt{\ell})$. $Z_\ell$ clearly satisfies Condition 2.5. Using Lemmata A.3 and A.4, we have that $Z_\ell$ satisfies the other two conditions as well. Using the union bound over these three events, we get the required result. $\qquad\square$

The following theorem guarantees the existence of an algorithm that finds approximately smallest balls containing almost the specified number of points. This is based off of the 1-cluster algorithm of Nissim and Stemmer [NS18]. We provide its proof, and state such an algorithm in Section B.

**Theorem 4.11** (Private Location for GMMs, Extension of [NS18])**.** *There is an $(\varepsilon, \delta)$-differentially private algorithm* $\mathrm{PGLOC}(X, t; \varepsilon, \delta, R, \sigma_{\min}, \sigma_{\max})$ *with the following guarantee. Let $X = (X_1, \dots, X_n) \in \mathbb{R}^{n \times d}$ be a set of $n$ points drawn from a mixture of Gaussians $\mathcal{D} \in \mathcal{G}(\ell, k, R, \sigma_{\min}, \sigma_{\max}, w_{\min}, s)$. Let $S \subseteq X$ such that $|S| \geq t$, and let $0 < a < 1$ be any small absolute constant (say, one can take $a = 0.1$). If $t = \gamma n$, where $0 < \gamma \leq 1$, and*

$$
n \geq \left( \frac{\sqrt{\ell}}{\gamma\varepsilon} \right)^{\frac{1}{1-a}} \cdot 9^{\log^* \left( \sqrt{\ell} \left( \frac{R\sigma_{\max}}{\sigma_{\min}} \right)^\ell \right)} \cdot \mathrm{polylog}\left( \ell, \frac{1}{\varepsilon}, \frac{1}{\delta}\frac{1}{\beta}, \frac{1}{\gamma} \right) + O\left( \frac{\ell + \log(k/\beta)}{w_{\min}} \right),
$$

*then for some absolute constant $c > 4$ that depends on $a$, with probability at least $1 - \beta$, the algorithm outputs $(r, \vec{c})$ such that the following hold:*

1. *$B_r(\vec{c})$ contains at least $\frac{t}{2}$ points in $S$, that is, $\left| B_r(\vec{c}) \cap S \right| \geq \frac{t}{2}$.*

2. *If $r_{\mathrm{opt}}$ is the radius of the smallest ball containing at least $t$ points in $S$, then $r \leq c\left( r_{\mathrm{opt}} + \frac{1}{4}\sqrt{\ell}\,\sigma_{\min} \right)$.*

Since the constant $a$ can be arbitrarily small, for simplicity, we fix it to 0.1 for the remainder of this section.

The first lemma we prove says that individual components are located correctly in the lower dimensional subspace, which is to say that we find $k$ disjoint balls, such that each ball completely contains exactly one component.

We first define the following events.

1. $E_{Y,Z}$: $Y$ and $Z$ satisfy Condition 2.5

2. $E_Z$: $Z$ satisfies Conditions 2.5 and 4.5

3. $E_{Z_\ell}$: $Z_\ell$ satisfies Conditions 2.5, 4.5, and 4.6.

**Lemma 4.12.** *Let $Z_\ell$ be the dataset as defined in the algorithm, and let $Z_\ell^L$ be its corresponding labelled dataset. If*

$$n \geq O\left( \frac{d^{\frac{3}{2}} k^3 \sqrt{\ln(1/\delta)}}{\varepsilon} + \frac{k^2}{\alpha^2} \ln(k/\beta) + \frac{\ell^{\frac{5}{9}} k^{\frac{5}{3}}}{\varepsilon^{\frac{10}{9}}} \cdot \text{polylog}\left(\ell, k, \frac{1}{\varepsilon}, \frac{1}{\delta}, \frac{1}{\beta}\right) \right),$$

*and events $E_{Y,Z}$ and $E_{Z_\ell}$ happen, then with probability at least $1 - 5\beta$, at the end of the first loop,*

1. *$i = k + 1$, that is, the algorithm has run for exactly $k$ iterations;*

2. *for all $1 \leq i \leq k$, if $u, v \in B_{r_i}(c_i)$, and $(u, \rho_u), (v, \rho_v) \in Z_\ell^L$, then $\rho_u = \rho_v$;*

3. *for all $1 \leq i \neq j \leq k$, if $u \in B_{r_i}(c_i)$, $v \in B_{r_j}(c_j)$, and $(u, \rho_u), (v, \rho_v) \in Z_\ell^L$, then $\rho_u \neq \rho_v$;*

4. *for all $1 \leq i \neq j \leq k$, $B_{r_i}(c_i) \cap B_{r_j}(c_j) = \emptyset$;*

5. *$S_1 \cup \cdots \cup S_k = Z_\ell$;*

6. *for all $1 \leq i \leq k$, if $u \in B_{r_i}(c_i)$, and $(u, \rho_u) \in Z_\ell^L$, then $r_i \in \Theta(\sqrt{k} \sigma_{\rho_u})$.*

*Proof.* Throughout the proof, we will omit the conditioning on events $E_{Y,Z}$ and $E_{Z_\ell}$ for brevity. To prove this lemma, we first prove a claim that at the end of iteration $i$, the balls we have found so far are disjoint, and that each ball completely contains points from exactly one Gaussian, no two balls contain the same component, and the radius of each ball is not too large compared to the radius of the optimal ball that contains the component within it.

**Claim 4.13.** *Let $E_i$ be the event that at the end of iteration $i$,*

1. *the number of components found so far is $i$;*

2. *for all $1 \leq a \leq i$, if $u, v \in B_{r_a}(c_a)$, and $(u, \rho_u), (v, \rho_v) \in Z_\ell^L$, then $\rho_u = \rho_v$;*

3. *for all $1 \leq a \neq b \leq i$, if $u \in B_{r_a}(c_a)$, $v \in B_{r_b}(c_b)$, and $(u, \rho_u), (v, \rho_v) \in Z_\ell^L$, then $\rho_u \neq \rho_v$;*

4. *for all $1 \leq a \neq b \leq i$, $B_{r_a}(c_a) \cap B_{r_b}(c_b) = \emptyset$;*

5. *if $B_i = B_{r_1}(c_1) \cup \cdots \cup B_{r_i}(c_i)$, then for all $u \in B_i \cap Z_\ell$ and $v \in Z_\ell \setminus B_i$, such that $(u, \rho_u), (v, \rho_v) \in Z_\ell^L$, it holds that $\rho_u \neq \rho_v$.*

6. *for all $1 \leq a \leq i$, if $u \in B_{r_a}(c_a)$, and $(u, \rho_u) \in Z_\ell^L$, then $\frac{\sqrt{\ell} \sigma_{\rho_u}}{2} \leq r_a \leq 48c\sqrt{3\ell}\sigma_{\rho_u}$.*

*If*

$$n \geq O\left( \frac{d^{\frac{3}{2}} k^3 \sqrt{\ln(1/\delta)}}{\varepsilon} + \frac{k^2}{\alpha^2} \ln(k/\beta) + \frac{\ell^{\frac{5}{9}} k^{\frac{5}{3}}}{\varepsilon^{\frac{10}{9}}} \cdot \text{polylog}\left(\ell, k, \frac{1}{\varepsilon}, \frac{1}{\delta}, \frac{1}{\beta}\right) \right),$$

*then*

$$\mathbb{P}[E_i] \geq 1 - \frac{i\beta}{k}$$

*Proof.* We prove this by induction on $i$. Suppose the claim holds for $i - 1$. Then it is sufficient to show the following.

$$\mathbb{P}\left[E_i | E_{i-1}\right] \geq 1 - \frac{\beta}{k}$$

Note that the points are now from $\ell$ dimensional Gaussians owing to being projected upon an $\ell$ dimensional subspace. Conditioning on $E_{i-1}$ entails that at the beginning of iteration $i$, there is no label for points in $S_\ell$ that occurs for points in $B_{i-1} \cap Z_\ell$. Therefore, the number of points having the remaining labels is the same as in the beginning.

For any two labels $\rho_u, \rho_v$, suppose $\mu''_{\rho_u}, \mu''_{\rho_v}$ are the respective means (in the space after projection and rotation), $\sigma^2_{\rho_u}, \sigma^2_{\rho_v}$ are the respective variances, $\mu_{\rho_u}, \mu_{\rho_v}$ are the original means, and $\mu'_{\rho_u}, \mu'_{\rho_v}$ are the projected, unrotated means. Then by conditioning on $E_{Y,Z}$ and $E_{Z_\ell}$, and using Lemma 4.9, we have

$$\left\|\mu_{\rho_u} - \mu_{\rho_v}\right\|_2 \geq (400c + 16\kappa)\sqrt{\ell}\max\{\sigma_{\rho_u}, \sigma_{\rho_v}\}$$
$$\implies \left\|\mu'_{\rho_u} - \mu'_{\rho_v}\right\|_2 \geq 400c\sqrt{\ell}\max\{\sigma_{\rho_u}, \sigma_{\rho_v}\}$$
$$\implies \left\|\mu''_{\rho_u} - \mu''_{\rho_v}\right\|_2 \geq 400c\sqrt{\ell}\max\{\sigma_{\rho_u}, \sigma_{\rho_v}\},$$

where the final inequality holds because the $\ell_2$ norm is rotationally invariant. Therefore by conditioning on event $E_{Z_\ell}$ again, for any two points with labels $\rho_1$ and $\rho_2$, the distance between the two is strictly greater than

$$200c\sqrt{\ell}\max\{\sigma_{\rho_1}, \sigma_{\rho_2}\}.$$

Now, conditioning on $E_{Z_\ell}$, we know that the radii of individual clusters are bounded. Because the components in this subspace are well-separated, the smallest ball containing at least $\frac{n}{2k}$ points cannot have points from two different components, that is, the radius of the smallest ball containing at least $\frac{n}{2k}$ points has to be the radius of the smallest ball that contains at least $\frac{n}{2k}$ points from a single component. Let that component have label $\rho_1$ (without loss of generality), and let its radius be $r_{opt}$. By Theorem 4.11 (by setting $S = S_\ell$), with probability at least $1 - \frac{\beta}{k}$, we obtain a ball of radius at most

$$c\left(r_{opt} + \frac{1}{4}\sqrt{\ell}\sigma_{\min}\right) \leq 2cr_{opt},$$

that contains at least $\frac{n}{4k}$ points from the component. Since $Z_\ell$ satisifes Condition 4.5, multiplying this radius by $4\sqrt{3}$ (to get $r_i$) is enough to cover all points of that component, hence, we have that

$$\frac{\sqrt{\ell}\sigma_{\rho_1}}{2} \leq r_{opt} \leq r_i \leq 8\sqrt{3}cr_{opt} \leq 24c\sqrt{\ell}\sigma_{\rho_1}. \tag{3}$$

We want to show that $B_{r_i}(c_i)$ contains points from exactly one component among all points in $Z_\ell$. There are two cases to consider. First, where the ball contains points that have label $\rho_1$, that is, the ball returned contains points only from the cluster with the smallest radius. Second, where the ball returned contains points from some other component that has a different label $\rho_2$ (without loss of generality), that is, the ball returned does not contain any points from the smallest cluster, but has points from a different cluster. As we will show later, this can only happen if the radius of this other cluster is "similar" to that of the smallest cluster.

In the first case, $B_{r_i}(c_i)$ completely contains all points with label $\rho_1$. But because this radius is less than its distance from every other point in $Z_\ell$ with a different label, it does not contain any points from any other labels in $Z_\ell$. This proves (1), (2), (3), (5), and (6) for this case. Now, consider any index $a \leq i - 1$. Then $B_{r_a}(c_a)$ contains exactly one component, which has label (without loss of generality) $\rho_3$. Let $u \in B_{r_a}(c_a)$ and $v \in B_{r_i}(c_i)$. Then we have the following.

$$
\begin{aligned}
\|c_i - c_a\|_2 &= \|(u - v) - (v - c_i) - (c_a - u)\|_2 \\
&\geq \|u - v\|_2 - \|v - c_i\|_2 - \|c_a - u\|_2 \\
&> 200c\sqrt{\ell}\max\{\sigma_{\rho_i}, \sigma_{\rho_a}\} - 48c\sqrt{3\ell}\sigma_{\rho_a} - 24c\sqrt{\ell}\sigma_{\rho_i} \\
&> r_i + r_a
\end{aligned}
$$

This shows that $B_{r_a}(c_a)$ and $B_{r_i}(c_i)$ do not intersect. This proves (4) for this case.

In the second case, let $r_{\rho_2}$ be the radius of the smallest ball containing points only from the component with label $\rho_2$. Since $r_{opt}$ is the radius of the component with label $\rho_1$, and $Z_\ell$ satisfies Condition 4.5, it must be the case that

$$
\frac{\sqrt{\ell}\sigma_{\rho_1}}{2} \leq r_{opt} \leq r_{\rho_2} \leq \sqrt{3\ell}\sigma_{\rho_2} \implies \sqrt{3\ell}\sigma_{\rho_2} \geq \frac{\sqrt{\ell}\sigma_{\rho_1}}{2} \implies 2\sqrt{3}\sigma_{\rho_2} \geq \sigma_{\rho_1},
$$

otherwise the smallest component would have been the one with label $\rho_2$. Combined with Inequality 3, this implies that $r_i \leq 48c\sqrt{3\ell}\sigma_{\rho_2}$, which proves (6) for this case. The arguments for other parts for this case are identical to those for the first case.

Since the only randomness comes from running PGLoc, using Theorem 4.11, we get,

$$
\mathbb{P}[E_i | E_{i-1}] \geq 1 - \frac{\beta}{k}.
$$

Therefore, by the inductive hypothesis,

$$
\mathbb{P}[E_i] \geq 1 - \frac{i\beta}{k}.
$$

The argument for the base case is the same, so we omit that for brevity. □

We complete the proof by using the above claim by setting $i = k$. □

The next lemma states that given that the algorithm has isolated individual components correctly in the lower dimensional subspace, it can correctly classify the corresponding points in the original space. To elaborate, this means that for each component, the algorithm finds a small ball that contains the component, and is able to correctly label all points in the dataset.

**Lemma 4.14.** *Let $Z$, $Z_\ell$, and $\widehat{S}_i, \ldots, \widehat{S}_k$ be datasets as defined in the the algorithm, and let $Z^L$ be the corresponding labelled dataset of $Z$. If*

$$
n \geq O\left(\frac{d^{\frac{3}{2}}k^3\sqrt{\ln(1/\delta)}}{\varepsilon} + \frac{k^2}{\alpha^2}\ln(k/\beta) + \frac{\ell^{\frac{5}{9}}k^{\frac{5}{3}}}{\varepsilon^{\frac{10}{9}}} \cdot \text{polylog}\left(\ell, k, \frac{1}{\varepsilon}, \frac{1}{\delta}, \frac{1}{\beta}\right)\right),
$$

*and events $E_{Y,Z}$, $E_Z$, and $E_{Z_\ell}$ happen then with probability at least $1 - 11\beta$,*

1. *for all $1 \leq i \leq k$, if $u, v \in \widehat{S}_i$, and $(u, \rho_u), (v, \rho_v) \in Z^L$, then $\rho_u = \rho_v$;*

2. *for all $1 \leq i \neq j \leq k$, if $u \in \widehat{S}_i$, $v \in \widehat{S}_j$, and $(u, \rho_u), (v, \rho_v) \in Z^L$, then $\rho_u \neq \rho_v$;*

3. *$\widehat{S}_1 \cup \cdots \cup \widehat{S}_k = Z$;*

4. *for all $1 \leq i \leq k$, if $u \in \widehat{S}_i$, and $(u, \rho_u) \in Z^L$, then $r_i \in \Theta(\sqrt{k}\sigma_{\rho_u})$.*

*Proof.* Again for brevity, we will omit conditioning on events $E_{Y,Z}$, $E_Z$, and $E_{Z_\ell}$. We will prove the following claim that says that by the end of iteration $i$ of the second loop, each set formed so far completely contains exactly one component, and the ball that contains it is small. But before that, note that: (1) from Lemma 4.9, with probability at least $1 - 4\beta$, for all components, the distance between the original mean and its respective projection onto $\widehat{V}_\ell$ is small; and (2) from Lemma 4.12, with probability at least $1 - 5\beta$, the balls found in the lower dimensional subspace in the previous step isolate individual components in balls, whose radii are small. We implicitly condition the following claim on them.

**Claim 4.15.** *Let $E_i$ be the event that at the end of iteration $i$,*

1. *for all $1 \leq a \leq i$, if $u, v \in \widehat{S}_i$, and $(u, \rho_u), (v, \rho_v) \in Z^L$, then $\rho_u = \rho_v$;*

2. *for all $1 \leq a \neq b \leq i$, if $u \in \widehat{S}_i$, $v \in \widehat{S}_j$, and $(u, \rho_u), (v, \rho_v) \in Z^L$, then $\rho_u \neq \rho_v$;*

3. *if $T_i = \widehat{S}_1 \cup \cdots \cup \widehat{S}_i$, then for all $u \in T_i \cap Z$ and $v \in Z \setminus T_i$, such that $(u, \rho_u), (v, \rho_v) \in Z^L$, it holds that $\rho_u \neq \rho_v$.*

4. *for all $1 \leq a \leq i$, if $u \in \widehat{S}_a$, and $(u, \rho_u) \in Z_\ell^L$, then $r_a \in \Theta(\sqrt{d}\sigma_{\rho_u})$.*

*If*

$$ n \geq O\left( \frac{d^{\frac{3}{2}}k^3\sqrt{\ln(1/\delta)}}{\varepsilon} + \frac{k^2}{\alpha^2}\ln(k/\beta) + \frac{\ell^{\frac{5}{9}}k^{\frac{5}{3}}}{\varepsilon^{\frac{10}{9}}} \cdot \mathrm{polylog}\left(\ell, k, \frac{1}{\varepsilon}, \frac{1}{\delta}, \frac{1}{\beta}\right) \right), $$

*then*

$$ \mathbb{P}[E_i] \geq 1 - \frac{2i\beta}{k}. $$

*Proof.* We prove this by induction on $i$. Suppose the claim holds for $i - 1$. Then it is sufficient to show the following.

$$ \mathbb{P}[E_i | E_{i-1}] \geq 1 - \frac{2\beta}{k} $$

From Lemma 4.12, we know that $B_{r_i}(c_i)$ completely contains exactly one component from $Z_\ell$. This implies that the empirical mean of those points ($\dot{\mu}$) also lies within $Z_\ell$. Suppose the mean of their distribution in $\widehat{V}_\ell$ is $\mu'$ and its variance is $\sigma^2$. We know from Lemma 2.16, and because $n \geq O\left(\frac{\ln(k/\beta)}{\ell}\right)$, that with probability at least $1 - \beta/k$,

$$ \left\| \dot{\mu} - \mu' \right\|_2 \leq \sqrt{\ell}\sigma. $$

Let $\mu$ be the mean of the Gaussian in the original subspace. We know from Lemma 4.9 that,

$$ \left\| \mu - \mu' \right\|_2 \leq 8\sqrt{\ell}\sigma_{\max}. $$

Let the empirical mean of the corresponding points in $Z$ be $\widetilde{\mu}$. Again, from Lemma 2.16, and because $n \geq O\left(\frac{\ln(k/\beta)}{\ell} + \frac{d}{\ell}\right)$, we know that with probability at least $1 - \beta/k$,

$$\left\|\mu - \widetilde{\mu}\right\|_2 \leq \sqrt{\ell}\sigma.$$

Therefore, by triangle inequality, $B_{r_i + 10\sqrt{\ell}\kappa\sigma_{\min}}(\widehat{c_i})$ contains $\widetilde{\mu}$. Now, from the proof of Lemma A.3, we know that all points of the Gaussian in $Z$ will be at most $\sqrt{3d}\sigma$ away from $\widetilde{\mu}$. But since $B_{r_i}(c_i)$ contains all points from the Gaussian in $Z_\ell$, we know from conditioning on $E_{Z_\ell}$ that $r_i \geq \frac{\sqrt{\ell}\sigma}{2}$, which means that the each of the corresponding points in $Z$ is at most $2r_i\sqrt{\frac{3d}{\ell}}$ away from $\widetilde{\mu}$. Therefore, by triangle inequality, the distance between $\widehat{c_i}$ and any of these points in $Z$ can be at most

$$r_i + 10\sqrt{\ell}\kappa\sigma_{\min} + 2r_i\sqrt{\frac{3d}{\ell}}.$$

Because this is exactly how $\widehat{r_i}$ is defined, $B_{\widehat{r_i}}(\widehat{c_i})$ completely contains all points from the component in $Z$. Since $S_i$ contains all the corresponding points from $Z_\ell$, it must be the case that $\widehat{S}_i$ contains all the points from the component.

Finally, we prove that the radius of the ball enclosing the component in $Z$ is small enough.

$$\widehat{r_i} = r_i + 10\sqrt{\ell}\kappa\sigma_{\min} + 2r_i\sqrt{\frac{3d}{\ell}}$$
$$\implies r_i + 10\sqrt{\ell}\kappa\sigma_{\min} + \Omega(\sqrt{d}\sigma) \leq \widehat{r_i} \leq r_i + 10\sqrt{\ell}\kappa\sigma_{\min} + O(\sqrt{d}\sigma) \qquad (r_i \in \Theta(\sqrt{\ell}\sigma))$$
$$\implies \widehat{r_i} \in \Theta(\sqrt{d}\sigma). \qquad (\kappa \in \Theta(1))$$

Therefore,

$$\mathbb{P}\left[E_i | E_{i-1}\right] \geq 1 - \frac{2\beta}{k}.$$

Hence, by the inductive hypothesis,

$$\mathbb{P}\left[E_i\right] \geq 1 - \frac{2i\beta}{k}.$$

The argument for the base case is identical, so we skip it for brevity. $\qquad\square$

We complete the proof by setting $i = k$ in the above claim, and using the union bound. $\qquad\square$

### 4.2.4 Estimation

Once we have identified individual components in the dataset, we can invoke a differentially private learning algorithm on each one of them separately. The next theorem establishes the existence of one such private learner that is tailored specifically for spherical Gaussians, and is accurate even when the number of points in the dataset constitutes sensitive information. We provide its proof, and state such an algorithm in Section C.

**Theorem 4.16.** *There exists an $(\varepsilon,\delta)$-differentially private algorithm $\mathrm{PSGE}(X;\vec{c},r,\alpha_\mu,\alpha_\sigma,\beta,\varepsilon,\delta)$ with the following guarantee. If $B_r(\vec{c}) \subseteq \mathbb{R}^\ell$ is a ball, $X_1,\ldots,X_n \sim \mathcal{N}(\mu,\sigma^2\mathbb{I}_{\ell\times\ell})$, and $n \geq \frac{6\ln(5/\beta)}{\varepsilon} + n_\mu + n_\sigma$, where*

$$n_\mu = O\left( \frac{\ell}{\alpha_\mu^2} + \frac{\ln(1/\beta)}{\alpha_\mu^2} + \frac{r\ln(1/\beta)}{\alpha_\mu\varepsilon\sigma} + \frac{r\sqrt{\ell\ln(1/\delta)}}{\alpha_\mu\varepsilon\sigma} \right),$$

$$n_\sigma = O\left( \frac{\ln(1/\beta)}{\alpha_\sigma^2\ell} + \frac{\ln(1/\beta)}{\alpha_\sigma\varepsilon} + \frac{r^2\ln(1/\beta)}{\alpha_\sigma\varepsilon\sigma^2\ell} \right),$$

*then with probability at least $1-\beta$, the algorithm returns $\widehat{\mu},\widehat{\sigma}$ such that if $X$ is contained in $B_r(\vec{c})$ (that is, $X_i \in B_r(\vec{c})$ for every $i$) and $\ell \geq 8\ln(10/\beta)$, then*

$$\|\mu - \widehat{\mu}\|_2 \leq \alpha_\mu\sigma \qquad and \qquad (1-\alpha_\sigma) \leq \frac{\widehat{\sigma}^2}{\sigma^2} \leq (1+\alpha_\sigma).$$

With the above theorem in our tookit, we can finally show that the each estimated Gaussian is close to its respective actual Gaussian to within $\alpha$ in TV-distance.

**Lemma 4.17.** *Suppose $\mu_1,\ldots,\mu_k$ are the means, and $\sigma_1^2,\ldots,\sigma_k^2$ are the variances of the Gaussians of the target distribution $\mathcal{D} \in \mathcal{S}(d,k,\kappa,(\xi + 16\sqrt{\kappa})\sqrt{\ell})$, where $\xi,\kappa \in \Theta(1)$ and $\xi$ is a universal constant, and $\ell = \max\{512\ln(nk/\beta),k\}$. Let $\widehat{\mu}_1,\ldots,\widehat{\mu}_k$, and $\widehat{\sigma}_1^2,\ldots,\widehat{\sigma}_k^2$ be their respective estimations produced by the algorithm. If*

$$n \geq n_{\mathrm{CLUSTERING}} + O\left( \frac{dk}{\alpha^2} + \frac{dk\sqrt{\ln(1/\delta)}}{\alpha\varepsilon} + \frac{\sqrt{d}k\ln(k/\beta)}{\alpha\varepsilon} + \frac{k\ln(k/\beta)}{\alpha^2} \right),$$

*where*

$$n_{\mathrm{CLUSTERING}} \geq O\left( \frac{d^{\frac{3}{2}}k^3\sqrt{\ln(1/\delta)}}{\varepsilon} + \frac{k^2}{\alpha^2}\ln(k/\beta) + \frac{\ell^{\frac{5}{9}}k^{\frac{5}{3}}}{\varepsilon^{\frac{10}{9}}} \cdot \mathrm{polylog}\left( \ell,k,\frac{1}{\varepsilon},\frac{1}{\delta},\frac{1}{\beta} \right) \right),$$

*and events $E_{Y,Z}$ and $E_Z$ happen, then with probability at least $1-12\beta$,*

$$\forall\, 1 \leq i \leq k, \quad \mathrm{d}_{\mathrm{TV}}\left( \mathcal{N}(\mu_i,\sigma_i^2\mathbb{I}_{d\times d}), \mathcal{N}(\widehat{\mu}_i,\widehat{\sigma}_i^2\mathbb{I}_{d\times d}) \right) \leq \alpha.$$

*Proof.* We know that from Lemma 4.14 that all points in $Z$ get correctly classified as per their respective labels by the algorithm with probability at least $1-11\beta$, and that we have centers and radii of private balls that completely contain one unique component each. In other words, for all $1 \leq i \leq k$, we have a set $\widehat{S}_i$ that contains all points from component $i$, such that $\widehat{S}_i \subset B_{\widehat{r}_i}(\widehat{c}_i)$, where $\widehat{r}_i \in \Theta(\sqrt{d}\sigma_i)$.

Now, from Theorem 4.16 and our bound on $n$, since $Z$ satisfies Condition 2.5, we know that for each $1 \leq i \leq k$, with probability at least $1 - \frac{\beta}{k}$, we output $\widehat{\mu}_i$ and $\sigma_i$, such that

$$\left\|\mu_i - \widehat{\mu}_i\right\|_2 \leq O(\alpha) \quad and \quad 1 - O\left(\frac{\alpha}{\sqrt{d}}\right) \leq \frac{\widehat{\sigma}_i^2}{\sigma_i^2} \leq 1 + O\left(\frac{\alpha}{\sqrt{d}}\right).$$

This implies from Lemma 2.17 that

$$\mathrm{d}_{\mathrm{TV}}\left( \mathcal{N}(\widehat{\mu}_i,\widehat{\sigma}_i^2\mathbb{I}_{d\times d}), \mathcal{N}(\mu_i,\sigma^2\mathbb{I}_{d\times d}) \right) \leq \alpha.$$

By applying the union bound, finally, we get the required result. $\qquad\square$

### 4.2.5 Putting It All Together

Given all the results above, we can finally complete the proof of Theorem 4.4.

*Proof of Theorem 4.4.* Let $G_1, \ldots, G_k$ be the actual Gaussians in $\mathcal{D}$, and let $\widehat{G}_1, \ldots, \widehat{G}_k$ be their respective estimations. Now that the individual estimated Gaussians are close to within $\alpha$ in TV-distance to the actual respective Gaussians, we can say the following about closeness of the two mixtures.

$$
\begin{aligned}
\mathrm{d}_{\mathrm{TV}}(\widehat{G}, \mathcal{D}) &= \max_{S \in \mathbb{R}^d} \left| \widehat{G}(S) - \mathcal{D}(S) \right| \\
&= \frac{1}{k} \max_{S \in \mathbb{R}^d} \left| \sum_{i=1}^{k} \widehat{G}_i(S) - G_i(S) \right| \\
&\leq \frac{1}{k} \sum_{i=1}^{k} \max_{S \in \mathbb{R}^d} \left| \widehat{G}_i(S) - G_i(S) \right| \\
&= \frac{1}{k} \sum_{i=1}^{k} \mathrm{d}_{\mathrm{TV}}(\widehat{G}_i, G_i) \\
&\leq \alpha
\end{aligned}
$$

Here, the last inequality follows with probability at least $1 - 12\beta$ from Lemma 4.17. Note that our algorithms and theorem statements along the way required various regularity conditions. By Lemma 4.7 and 4.10, these events all occur with probability at least $1 - 11\beta$, which gives us the success probability of at least $1 - 23\beta$. This completes our proof of Theorem 4.4. $\qquad \square$

## 5  An Algorithm for Privately Learning Mixtures of Gaussians

In this section, we present our main algorithm for privately learning mixtures of Gaussians and prove the following accuracy result.

**Theorem 5.1.** *For all $\varepsilon, \delta, \alpha, \beta > 0$, there exists an $(\varepsilon, \delta)$-DP algorithm that takes $n$ samples from $\mathcal{D} \in \mathcal{G}(d, k, \sigma_{\min}, \sigma_{\max}, R, w_{\min}, s)$, where $\mathcal{D}$ satisfies (7), and $s$ is defined by the following separation condition*

$$
\forall i, j, \quad \|\mu_i - \mu_j\|_2 \geq 100(\sigma_i + \sigma_j)\left( \sqrt{k \log(n)} + \frac{1}{\sqrt{w_i}} + \frac{1}{\sqrt{w_j}} \right)
$$

*and returns a mixture of $k$ Gaussians $\widehat{\mathcal{D}}$, such that if*

$$
n \geq \max \left\{ \tilde{\Omega}\left( \left( \frac{\sqrt{dk}\log(dk/\beta)\log^{\frac{3}{2}}(1/\delta)\log\log((R + \sqrt{d}\sigma_{\max})/\sigma_{\min})}{w_{\min}\varepsilon} \right)^{\frac{1}{1-a}} \right) \text{ for an arbitrary constant } a > 0, \right.
$$

$$
\Omega\left( \frac{k^{9.06}d^{3/2}\log(1/\delta)\log(k/\beta)}{w_{\min}\varepsilon} \right),
$$

$$
\left. \Omega\left( \frac{k^{\frac{3}{2}}\log(k\log((R + \sqrt{d}\sigma_{\max})/\sigma_{\min})/\beta)\log(1/\alpha)\log(1/\delta)}{\alpha\varepsilon w_{\min}} \right), \quad \frac{1}{w_{\min}}(n_1 + n_2) \right\}
$$

*where,*

$$n_1 \geq \Omega\left(\frac{(d^2 + \log(\frac{k}{\beta}))\log^2(1/\alpha)}{\alpha^2} + \frac{(d^2\text{polylog}(\frac{dk}{\alpha\beta\varepsilon\delta}))}{\alpha\varepsilon} + \frac{d^{\frac{3}{2}}\log^{\frac{1}{2}}(\frac{\sigma_{\max}}{\sigma_{\min}})\text{polylog}(\frac{dk\log(\sigma_{\max}/\sigma_{\min})}{\beta\varepsilon\delta})}{\varepsilon}\right)$$

$$n_2 \geq \Omega\left(\frac{d\log(\frac{dk}{\beta})\log^2(1/\alpha)}{\alpha^2} + \frac{d\log(\frac{dk\log R\log(1/\delta)}{\beta\varepsilon})\log^{\frac{1}{2}}(\frac{1}{\delta})\log^2(1/\alpha)}{\alpha\varepsilon} + \frac{\sqrt{d}\log(\frac{Rdk}{\beta})\log^{\frac{1}{2}}(\frac{1}{\delta})}{\varepsilon}\right)$$

*then it $(\alpha, \beta)$-learns $\mathcal{D}$.*

## 5.1  Finding a Secluded Ball

In this section we detail a key building block in our algorithm for learning Gaussian mixtures. This particular subroutine is an adaptation of the work of Nissim and Stemmer [NS18] (who in turn built on [NSV16]) that finds a ball that contains many datapoints. In this section we show how to tweak their algorithm so that it now produces a ball with a few more additional properties. More specifically, our goal in this section is to privately locate a ball $B_r(p)$ that (i) contains many datapoints, (ii) leaves out many datapoints (i.e., its complement also contains many points) and (iii) is secluded in the sense that $B_{cr}(p) \setminus B_r(p)$ holds very few (and ideally zero) datapoints for some constant $c > 1$. More specifically, we are using the following definition.

**Definition 5.2** (Terrific Ball)**.** Given a dataset $X$ and an integer $t > 0$, we say a ball $B_r(p)$ is $(c, \Gamma)$-terrific for a constant $c > 1$ and parameter $\Gamma \geq 0$ if all of the following three properties hold: (i) The number of datapoints in $B_r(p)$ is at least $t - \Gamma$; (ii) The number of datapoints outside the ball $B_{cr}(p)$ is least $t - \Gamma$; and (iii) The number of datapoints in the annulus $B_{cr}(p) \setminus B_r(p)$ is at most $\Gamma$.

We say a ball is $c$-terrific if it is $(c, 0)$-terrific, and when $c$ is clear from context we call a ball terrific. In this section, we provide a differentially private algorithm that locates a terrific ball.

**The 1-Cluster Algorithm of Nissim-Stemmer.**  First, we give an overview of the Nissim-Stemmer algorithm for locating a ball $B_r(p)$ that satisfy only property (i) above, namely a ball that holds at least $t$ points from our dataset containing $n$ points. Their algorithm is actually composed of two subroutines that are run sequentially. The first, `GoodRadius` privately computes some radius $\tilde{r}$ such that $\tilde{r} \leq 4r_{\text{opt}}$ with $r_{\text{opt}}$ denoting the radius of the smallest ball that contains $t$ datapoints. Their next subroutine, `GoodCenter`, takes $\tilde{r}$ as an input and produces a ball of radius $\gamma\tilde{r}$ that holds (roughly) $t$ datapoints with $\gamma$ denoting some constant $> 2$. The `GoodCenter` procedure works by first cleverly combining locality-sensitive hashing (LSH) and randomly-chosen axes-aligned boxes to retrieve a poly($n$)-length list of candidate centers, then applying ABOVETHRESHOLD to find a center point $p$ such that the ball $B_{\tilde{r}}(p)$ satisfies the required condition (holding enough datapoints).

In this section, we detail how to revise the two above-mentioned subprocedures in order to retrieve a terrific ball, satisfying all three properties (rather than merely holding many points). The revision isn't difficult, and it is particularly straight-forward for the `GoodCenter` procedure. In fact, we keep `GoodCenter` as is, except for the minor modification of testing for all

3 properties together. Namely, we replace the naïve counting query in ABOVETHRESHOLD (i.e., "is $|\{x \in X : x \in B_r(p)\}| \geq t?$") with the query

$$\min\Big\{ |\{x \in X : x \in B_r(p)\}|, \ \ |\{x \in X : \ x \notin B_{cr}(p)\}|, \ \ t - |\{x \in X : x \in B_{cr}(p) \setminus B_r(p)\}|\Big\} \overset{?}{\geq} t \qquad (4)$$

Note that this query is the minimum of 3 separate counting queries, and therefore its global sensitivity is 1. Thus, our modification focuses on altering the first part where we find a good radius, replacing the `GoodRadius` procedure with the `TerrificRadius` procedure detailed below. Once a terrific radius is found, we apply the revised `GoodCenter` procedure and retrieve a center.

**Remark 5.3.** In the work of Nissim-Stemmer [NS18] `GoodCenter`, the resulting ball has radius $\leq \gamma \tilde{r}$ since the last step of the algorithm is to average over all points in a certain set of radius $\leq \gamma \tilde{r}$. In our setting however it holds that $\tilde{r}$ is a radius of a terrific ball, and in particular, in a setting where $\gamma < c$ (which is the application we use, where $\gamma \approx 2.5$ whereas $c \geq 4$), this averaging is such that effectively all points come from $B_p(\tilde{r})$, and so the returned ball is of radius $\approx \tilde{r}$.

**The TerrificRadius procedure.** Much like the work of [NSV16], we too define a set of possible $r$'s, traverse each possible value of $r$ and associate a score function that measures its ability to be the sought-after radius. However, we alter the algorithm is two significant ways. The first is modifying the score function to account for all three properties of a terrific ball, and not just the one about containing many points; the second is that we do not apply the recursive algorithm to traverse the set of possible $r$'s, but rather try each one of these values ourselves using ABOVETHRESHOLD. The reason for the latter modification stems from the fact that our terrific radius is no longer upward closed (i.e. it is not true that if $r$ is a radius of a ball satisfying the above three properties then any $r' > r$ is also a radius of a ball satisfying these properties) and as a result our scoring function is *not* quasi-concave.

The modification to the scoring function is very similar to the one detailed in Equation (4), with the exception that rather than counting exact sizes, we cap the count at $t$. Formally, given a (finite) set $S$ we denote $\#^t S \overset{\text{def}}{=} \min\{|S|, t\}$, and so we define the counting query

$$Q_X(p, r) \overset{\text{def}}{=} \min\Big\{\#^t\{x \in X : \ x \in B_r(p)\}, \ \ \#^t\{x \in X : \ x \notin B_{cr}(p)\},$$
$$t - \#^t\{x \in X : \ x \in B_{cr}(p) \setminus B_r(p)\}\Big\} \qquad (5)$$

It is evident that for any dataset $X$, $p$ and $r > 0$ it holds that $Q_X(p, r)$ is an integer between 0 and $t$. Moreover, for any $p$ and $r$ and any two neighboring datasets $X$ and $X'$ it holds that $|Q_X(p, r) - Q_{X'}(p, r)| \leq 1$. We now have all ingredients for introducing the Terrific Radius subprocedure below.

**Claim 5.4.** *Algorithm 2 satisfies $(\varepsilon, 0)$-Differential Privacy.*

*Proof.* Much like [NSV16], we argue that $L_X(r)$ has $\ell_1$-sensitivity at most 2. With this sensitivity bound, the algorithm is just an instantiation of ABOVETHRESHOLD (Theorem 2.14), so it is $(\varepsilon, 0)$-Differentially Private.

Fix two neighboring datasets $X$ and $X'$. Fix $r$. Let $x_1^1, ..., x_t^1 \in X$ be the $t$ points on which $L_X(r)$ is obtained and let $x_1^2, ..., x_t^2 \in X'$ be the $t$ points on which $L_{X'}(r)$ is obtained. Since $X$ and $X'$ differ

---
**Algorithm 2:** Find Terrific Radius $(X, t, c, largest; \varepsilon, U, L)$

---

**Input:** Dataset $X \in \mathbb{R}^{n \times d}$. Privacy parameters $\varepsilon, \delta > 0$; failure probability $\beta > 0$; candidate size $t > 0$. Upper- and lower-bounds on the radius $L < U$. Boolean flag $largest$.

**Output:** A radius of a terrific ball.

Denote $L_X(r) = \frac{1}{t} \max\limits_{\text{distinct } x_1, x_2, \ldots, x_t \in X} \sum\limits_{j=1}^{t} Q_X(x_j, r)$.

Denote $T = \lceil \log_2(U/L) \rceil + 1$ and $\Gamma \overset{\text{def}}{=} \frac{16}{\varepsilon} (\log(T) + \log(2/\beta))$.

Set $r_0 = L, r_1 = 2L, r_2 = 4L, \ldots, r_i = 2^i L, \ldots, r_T = U$.

**If** $largest =$TRUE

    reverse the order of $r_i$'s.

Run ABOVETHRESHOLD on queries $L_X(r_i)$ for all $0 \le i \le T$, with threshold $t - \Gamma$ and sensitivity 2, and output the $r_i$ for which ABOVETHRESHOLD returns $\top$.

---

on at most 1 datapoint, then at most one point from each $t$-tuple can be a point that doesn't belong to $X \cap X'$. Without loss of generality, if such a point exists then it is $x_1^1$ and $x_1^2$. For all other points it follows that $|Q_X(x, r) - Q_{X'}(x, r)| \le 1$ as mentioned above. Thus we have

$$L_X(r) = \frac{1}{t} \sum_{i=1}^{t} Q_X(x_i^1, r) = \frac{1}{t} Q_X(x_1^1, r) + \frac{1}{t} \sum_{i=2}^{t} Q_X(x_i^1, r)$$

$$\le \frac{t}{t} + \frac{1}{t} \sum_{i=2}^{t} \left( Q_{X'}(x_i^1, r) + 1 \right) \le 1 + \left( \frac{1}{t} Q_{X'}(x_1^2, r) + \frac{1}{t} \sum_{i=2}^{t} Q_{X'}(x_i^1, r) \right) + 1 \le 2 + L_{X'}(r)$$

based on the fact that $0 \le Q(x, r) \le t$ for any $x$ and any $r$ and the definition of $L_X$ as a max-operation. The inequality $L_{X'}(r) \le L_X(r) + 2$ is symmetric. $\qquad \square$

**Claim 5.5.** *Fix $\beta > 0$ and denote $\Gamma$ as in Algorithm 2. With probability $\ge 1 - \beta$, if Algorithm 2 returns a radius r (and not $\perp$), then there exists a ball $B_r(p)$ which is $(c, 2\Gamma)$-terrific.*

*Proof.* Let $T + 1 = (\lceil \log_2(U/L) \rceil + 1)$ be the number of queries posed to ABOVETHRESHOLD. From Theorem 2.14, we know that with probability at least $1 - \beta$, if radius $r$ is returned, then $L_X(r) \ge t - \Gamma - \Gamma = t - 2\Gamma$, where $\Gamma = \frac{16}{\varepsilon} (\log(T) + \log(2/\beta))$. It follows that one of the $t$ distinct datapoints on which $L_X(r)$ is obtained must satisfy $Q(x_i, r) \ge t - 2\Gamma$. Thus, the ball $B_r(x_i)$ is the sought-after ball. $\qquad \square$

Next we argue that if the data is such that it has a terrific ball, our algorithm indeed returns a ball of comparable radius. Note however that the data could have multiple terrific balls. Furthermore, given a $c$-terrific ball, there could be multiple different balls of different radii that yield the same partition. Therefore, given dataset $X$ which has a $c$-terrific ball $B_r(p)$, denote $A = X \cap B_r(p)$ and define $r_A$ as the *minimal terrific radius* of a $c$-terrific ball forming $A$, i.e., for any other $c$-terrific $B_{r'}(p')$ such that $A = X \cap B_{r'}(p')$, we have that $r' \ge r_A$. Let $\mathcal{R}_c$ be the set of all minimal terrific radii of $c$-terrific balls forming subsets of $X$.

**Claim 5.6.** *Fix $c > 1$ and assume we apply Algorithm 2 with this $c$ as a parameter. Given X such that $\mathcal{R}_{4c+1}$ is not empty, then with probability $\geq 1 - \beta$ it holds that Algorithm 2 returns $r$ such that if largest = TRUE then $r \geq \max\{r' \in \mathcal{R}_{4c+1}\}$ and if largest = FALSE then $r \leq 4 \cdot \min\{r' \in \mathcal{R}_{4c+1}\}$.*

*Proof.* Given $T + 1 = (\lceil \log_c(U/L) \rceil + 1)$ queries and threshold $t - \Gamma$ (using the same definition of $\Gamma$ as in Algorithm 2), from Theorem 2.14, it holds with probability $\geq 1 - \beta$ that ABOVETHRESHOLD must halt by the time it considers the very first $r$ for which $L_X(r) \geq t$. We show that for any $r \in \mathcal{R}_{4c+1}$ there exists a query among the queries posed to ABOVETHRESHOLD over some $r'$ such that (a) $r \leq r' \leq 4r$ and (b) $L_X(r') = t$. Since the order in which the ABOVETHRESHOLD mechanism iterates through the queries is determined by the boolean flag *largest* the required follows.

Fix $r \in \mathcal{R}_{4c+1}$. Let $B_r(p)$ be a terrific ball of radius $r$ that holds at least $t$ points, and let $x_1, ..., x_{t'}$ denote the set of $t' \geq t$ distinct datapoints in $B_r(p)$. Denote $D = \max_{i \neq j} \|x_i - x_j\|$. Clearly, $r \leq D \leq 2r$, the lower-bound follows from the minimality of $r$ as a radius that separates these datapoints from the rest of $X$ and the upper-bound is a straight-forward application of the triangle inequality. Next, let $r^*$ denote the radius in the range $[D, 2D] \subset [r, 4r]$ which is posed as a query to ABOVETHRESHOLD. We have that $Q(x_j, r^*) = t$ for each $x_j$. Indeed, the ball $B_{r^*}(x_j)$ holds $t' \geq t$ datapoints. Moreover, $B_{cr^*}(x_j) \subset B_{r+cr^*}(p) \subset B_{(4c+1)r}(p)$ and therefore $(B_{2r^*}(x_j) \setminus B_{r^*}(x_j)) \subset (B_{(4c+1)r}(p) \setminus \{x_1, ..., x_{t'}\})$ and thus it is empty too. Therefore, any datapoint outside of $B_r(p)$ which must also be outside $B_{(4c+1)r}(p)$ is contained in the compliment of $B_{r^*}(x_j)$, and so the compliment also contains $t$ points. As this holds for any $x_j$, it follows that $L_X(r^*) = t$ and thus the required is proven. $\square$

**The entire** `PTerrificBall` **procedure.** The overall algorithm that tries to find a $c$-terrific ball is the result of running `TerrificRadius` followed by the `GoodCenter` modified as discussed above: we conclude by running ABOVETHRESHOLD with the query given by (4) for a ball of radius $(1 + \frac{c}{10})\tilde{r}$. Its guarantee is as follows.

**Lemma 5.7.** *The* `PTerrificBall` *procedure is a $(2\varepsilon, \delta)$-DP algorithm which, if run using size-parameter $t \geq \frac{1000c^2}{\varepsilon} n^a \sqrt{d} \log(nd/\beta) \log(1/\delta) \log\log(U/L)$ for some arbitrary small constant $a > 0$ (say $a = 0.1$), and is set to find a $c$-terrific ball with $c > \gamma$ ($\gamma$ being the parameter fed into the LSH in the* `GoodCenter` *procedure), then the following holds. With probability at least $1 - 2\beta$, if it returns a ball $B_p(r)$, then this ball is a $(c, 2\Gamma)$-terrific ball of radius $r \leq (1 + \frac{c}{10})\tilde{r}$, where $\tilde{r}$ denotes the radius obtained from its call to* `TerrificRadius`, *and $\Gamma$ is $\frac{16}{\varepsilon}(\log(\lceil \log_2(U/L) \rceil + 1) + \log(2/\beta))$.*

*Proof.* By Claim 5.5, it follows that $\tilde{r}$ is a radius for some $(c, 2\Gamma)$-terrific ball. The analysis in [NS18] asserts that with probability at least $1 - \beta$, `GoodCenter` locates a $(n^{-a}/2)$-fraction of the points inside the ball, and uses their average. Note that $t$ is set such that $t \cdot n^{-a}/2 > 10c^2 \cdot 2\Gamma$. Fix $x$ to be any of the $\geq t - 2\Gamma$ points inside the ball. Due to the quality function we use in `TerrificRadius`, at most $2\Gamma$ of the points, which got the same hash value as $x$, are within distance $\gamma\tilde{r} < c\tilde{r}$, and the remaining $t - 2\Gamma$ are within distance $\tilde{r}$ from $x$. It follows that their average is within distance at most $\tilde{r} + \frac{c\tilde{r}}{10c^2} \leq \tilde{r}(1 + \frac{c}{10})$. The rest of the [NS18] proof follows as in the `GoodCenter` case. $\square$

## 5.2   The Algorithm

We now introduce our overall algorithm for GMM clustering which mimics the approach of Achlioptas and McSherry [AM05]. Recall, Achlioptas and McSherry's algorithm correctly learns the model's parameters, provided that

$$\forall i,j, \quad \|\mu_i - \mu_j\|_2 \geq C(\sigma_i + \sigma_j)\left(\sqrt{k \log(nk/\beta)} + \frac{1}{\sqrt{w_i}} + \frac{1}{\sqrt{w_j}}\right) \tag{6}$$

for some constant $C > 0$, and that $n \geq \text{poly}(n,d,k)$. We argue that under the same separation condition (albeit we replace the constant of [AM05] with a much larger constant, say $C = 100$) a $(\varepsilon, \delta)$-differentially private algorithm can also separate the $k$-Gaussians and learn the model parameters. Alas, we are also forced to make some additional assumptions. First, we require a bound on the distribution parameters, that is, the norm of all means, and the eigenvalues of covariances; this is due to standard results in DP literature ([BNSV15]) showing the necessity of such bounds for non-trivial DP algorithms. The good news is that we only require $\log\log$-dependence on these parameters. Namely, our sample complexity is now $\text{poly}(n, k, d, \frac{1}{\varepsilon}, \log(1/\delta), \log(1/\beta), \log\log((R + \sqrt{d}\sigma_{\max})/\sigma_{\min}))$. Secondly, because we replace the non-private Kruskal based algorithm with an algorithm that locates a terrific ball, we are forced to use one additional assumption — that the Gaussians are not "too flat". The reason for this requirement is that we are incapable of dealing with the case that the pairwise distances of points drawn from the *same* Gaussian have too much variation in them (see Property 2 below). Formally, we require the following interplay between $n$ (the number of datapoints), $\beta$ (probability of error), and the norms of each Gaussian variance:

$$\forall i, \quad \|\Sigma_i\|_F \sqrt{\log(nk/\beta)} \leq \tfrac{1}{8}\text{tr}(\Sigma_i), \quad \text{and} \quad \|\Sigma_i\|_2 \log(nk/\beta) \leq \tfrac{1}{8}\text{tr}(\Sigma_i). \tag{7}$$

Note that for a spherical Gaussian (where $\Sigma_i = \sigma_i^2 I_{d\times d}$) we have that $\text{tr}(\Sigma_i) = d\sigma_i^2$, while $\|\Sigma_i\|_F = \sigma_i^2 \sqrt{d}$ and $\|\Sigma_i\|_2 = \sigma_i^2$, thus, the above condition translates to requiring a sufficiently large dimension. This assumption was explicitly stated in the non-private work regarding learning spherical Gaussians [VW02] (also, we ourselves made such an assumption in the simplified version detailed in Section 4).

We now detail the main component of our GMM learner. Algorithm 3 takes $X$, the given collection of datapoints, and returns a $k$-partition of $X$ in the form of list of subsets. We thus focus on finding the correct partition. Note that intrinsically, the returned partition of indices cannot preserve privacy (it discloses the cluster of datapoint $i$), so once this $k$-partition is done, one must apply the existing $(\varepsilon, \delta)$-DP algorithms for finding the mean and variance of each cluster, as well as apply $(\varepsilon, 0)$-DP histogram in order to assess the cluster weights. This is the overall algorithm (PGME) given in Algorithm 4.

In our analysis, we prove that our algorithm is $(O(\varepsilon\sqrt{k \log(1/\delta)}), O(k\delta))$-DP, and accurate with probability at least $1 - O(k\beta)$. The desired $(\varepsilon, \delta)$-DP and error probability $1 - \beta$ guarantees are achieved by appropriately rescaling the values of $\varepsilon, \delta$, and $\beta$.

**Theorem 5.8.** *Algorithm 4 satisfies $\left(2\varepsilon + 8\varepsilon\sqrt{2k \log(1/\delta)}, (8k+1)\delta\right)$-differential privacy.*

*Proof.* Consider any two neighboring datasets $X$ and $X'$, which differ on at most one point. In each of the levels of the recursion in RPGMP we apply three $(\varepsilon, \delta)$-DP, one $(\frac{\varepsilon}{2}, \delta)$-DP, and one

**Algorithm 3:** Recursive Private Gaussian Mixture Partitioning

---

**Input:** Dataset $X \in \mathbb{R}^{n \times d}$ coming from a mixture of at most $k$ Gaussians. Upper bound on the number of components $k$. Bounds on the parameters of the GMM $w_{\min}, \sigma_{\min}, \sigma_{\max}$. Privacy parameters $\varepsilon, \delta > 0$.

**Output:** Partition of $X$ into clusters.

$\text{RPGMP}(X; k, R, w_{\min}, \sigma_{\min}, \sigma_{\max}, \varepsilon, \delta)$:

1. **If** *(k = 1)*
    Skip to last step (#8).

2. Find a small ball that contains $X$, and bound the range of points to within that ball:
    Set $n' \leftarrow |X| + \text{Lap}(2/\varepsilon) - \frac{n w_{\min}}{20}$.
    $B_{r''}(p) \leftarrow \text{PGLOC}(X, n'; \frac{\varepsilon}{2}, \delta, R, \sigma_{\min}, \sigma_{\max})$.
    Set $r \leftarrow 12 r''$.
    Set $X \leftarrow X \cap B_r(p)$.

3. Find 5-PTerrificBall in $X$ with $t = \frac{n w_{\min}}{2}$:
    $B_{r'}(p') \leftarrow \text{PTERRIFICBALL}(X, \frac{n w_{\min}}{2}, c = 5, largest = \texttt{FALSE}; \varepsilon, \delta, r, \frac{\sqrt{d} \sigma_{\min}}{2})$.

4. If the data is separable already, we recurse on each part:
    **If** *($B_{r'}(p') \neq \bot$)*
        Partition $X$ into $A = X \cap B_{r'}(p')$ and $B = X \setminus B_{5r'}(p')$.
        Set $C_A \leftarrow \text{RPGMP}(A; k-1, R, w_{\min}, \sigma_{\min}, \sigma_{\max}, \varepsilon, \delta)$.
        Set $C_B \leftarrow \text{RPGMP}(B; k-1, R, w_{\min}, \sigma_{\min}, \sigma_{\max}, \varepsilon, \delta)$.
        **Return** $C_A \cup C_B$.

5. Find a private $k$-PCA of $X$:
    Sample $N$, a symmetric matrix whose entries are taken from $\mathcal{N}(0, \frac{4 r^4 \ln(2/\delta)}{\varepsilon^2})$.
    $\Pi \in \mathbb{R}^{d \times d} \leftarrow k$-PCA projection of $X^T X + N$, where $\Pi$ is a rank $k$ matrix.

6. Find 5-PTerrificBall in $X\Pi$ with $t = \frac{n w_{\min}}{2}$:
    $B_{r'}(p') \leftarrow \text{PTERRIFICBALL}(X\Pi, \frac{n w_{\min}}{2}, c = 5, largest = \texttt{TRUE}; \varepsilon, \delta, r, \frac{\sqrt{k} \sigma_{\min}}{2})$.

7. If the projected data is separable, we recurse on each part:
    **If** *($B_{r'}(p') \neq \bot$)*
        Partition $X$ into $A = \{x_i \in X : \Pi x_i \in B_{r'}(p')\}$ and $B = \{x_i \in X : \Pi x_i \notin B_{5r'}(p')\}$.
        Set $C_A \leftarrow \text{RPGMP}(A; k-1, R, w_{\min}, \sigma_{\min}, \sigma_{\max}, \varepsilon, \delta)$.
        Set $C_B \leftarrow \text{RPGMP}(B; k-1, R, w_{\min}, \sigma_{\min}, \sigma_{\max}, \varepsilon, \delta)$.
        **Return** $C_A \cup C_B$.

8. Since the data isn't separable, we treat it as a single Gaussian:
    Set a single cluster: $C \leftarrow \{i : x_i \in X\}$.
    **Return** $\{C\}$.

---

$(\frac{\varepsilon}{2}, 0)$-DP procedures to the data, as well as fork into one of the two possible partitions and invoke recursive calls on the part of the data the contains the point that lies in at most one of $X$ and $X'$. Since the recursion can be applied at most $k$ times, it follows that the point plays a

---

**Algorithm 4:** Private Gaussian Mixture Estimation

---

**Input:** Dataset $X \in \mathbb{R}^{n \times d}$ coming from a $k$-Gaussian mixture model. Upper bound on the number of components $k$. Bounds on the parameters of the GMM $w_{\min}, \sigma_{\min}, \sigma_{\max}$. Privacy parameters $\varepsilon, \delta > 0$; failure probability $\beta > 0$.

**Output:** Model Parameters Estimation

$\text{PGME}(X; k, R, w_{\min}, \sigma_{\min}, \sigma_{\max}, \varepsilon, \delta, \beta)$:

1. Truncate the dataset so that for all points, $\|X_i\|_2 \le O(R + \sigma_{\max}\sqrt{d})$

2. $\{C_1, .., C_k\} \leftarrow \text{RPGMP}(X; k, R, w_{\min}, \sigma_{\min}, \sigma_{\max}, \varepsilon, \delta)$

3. **For** $j$ *from* $1$ *to* $k$
   $(\mu_j, \Sigma_j) \leftarrow \text{PGE}(\{x_i : i \in C_j\}; R, \sigma_{\min}, \sigma_{\max}, \varepsilon, \delta)$.
   $\tilde{n}_j \leftarrow |C_j| + \text{Lap}(1/\varepsilon)$.

4. Set weights such that for all $j$, $w_j \leftarrow \tilde{n}_j / (\sum_j \tilde{n}_j)$.

5. **Return** $\langle \mu_j, \Sigma_j, w_j \rangle_{j=1}^{k}$

---

role in at most $k$ rounds, each of which is $(4\varepsilon, 4\delta)$-DP. By advanced composition (Lemma 2.9), overall PGME is $(4\varepsilon \cdot \sqrt{8k\log(1/\delta)}, 8k\delta)$-DP. The additional calls for learning the mixture model parameters on each partition are $(2\varepsilon, \delta)$-DP. Summing the two parts together yields the overall privacy-loss parameters. $\qquad \square$

Since the latter steps of Algorithm 4 rely on existing algorithms with proven utility, our utility analysis boils down to proving the correctness of Algorithm 3. We argue that given $X$ taken from a Gaussian Mixture Model with sufficiently many points and sufficient center separation, then with high probabiliy, Algorithm 3 returns a $k$-partition of the data which is *laminar* with the data. In fact, the algorithm might omit a few points in each level of the recursion, however, our goal is to argue that the resulting $k$-subsets are pure — holding only datapoints from a single cluster.

**Definition 5.9.** Given that $X$ is drawn from a $k'$-GMM, we call two disjoint sets $A, B \subseteq X$ a *laminar partition* of $X$ if (i) there exists a partition of the set $\{1, 2, .., k'\}$ into $S$ and $T$ such that all datapoints in $A$ are drawn from the Gaussians indexed by some $i \in S$ and all datapoints in $B$ are drawn from the Gaussians indexed by some $i \in T$, and (ii) $|X \setminus (A \cup B)| \le \frac{n w_{\min} \alpha}{10 k' \log(1/\alpha)}$.

Towards the conclusion that the partition returned by Algorithm 3 is laminar, we require multiple claims regarding the correctness of the algorithm through each level of the recursion. These claims, in turn, rely on the following properties of Gaussian data.

1. The Hanson-Wright inequality (Lemma 2.15): $\forall i, \forall x \sim \mathcal{N}(\mu_i, \Sigma_i)$ we have that

$$\text{tr}(\Sigma_i) - 2\|\Sigma_i\|_F \sqrt{\log(n/\beta)} \le \|x - \mu_i\|_2^2 \le \text{tr}(\Sigma_i) + 2\|\Sigma_i\|_F \sqrt{\log(n/\beta)} + 2\|\Sigma_i\|_2 \log(n/\beta).$$

Using the assumption that $2\|\Sigma_i\|_F \sqrt{\log(n/\beta)} \le \frac{1}{4}\text{tr}(\Sigma_i)$ and that $2\|\Sigma_i\|_2 \log(n/\beta) \le \frac{1}{4}\text{tr}(\Sigma_i)$ (Equation 7) we get that $\frac{3}{4}\text{tr}(\Sigma_i) \le \|x - \mu_i\|^2 \le \frac{3}{2}\text{tr}(\Sigma_i)$.

2. Similarly, for every $i$ and for any $x, y \sim \mathcal{N}(\mu_i, \Sigma_i)$ we have that $x - y \sim \mathcal{N}(0, 2\Sigma_i)$ implying under the same concentration bound that $\frac{3}{2}\mathrm{tr}(\Sigma_i) \leq \|x - y\|_2^2 \leq 3\mathrm{tr}(\Sigma_i)$.

3. For every $i$, its empirical spectral norm is close to the expected spectral norm, assuming for all $i$ we have $w_i n = \Omega(d + \log(1/\beta))$ (see Lemma 2.16). Namely,

$$\frac{1}{n_i} \left\| \sum_{\{x \in X: \ x \sim \mathcal{N}(\mu_i, \Sigma_i)\}} (x - \mu_i)(x - \mu_i)^T \right\|_2 \in \left( \tfrac{1}{2}\|\Sigma_i\|_2, \tfrac{3}{2}\|\Sigma_i\|_2 \right).$$

4. Under center separation, we have that $\forall i, \forall x \sim \mathcal{N}(\mu_i, \Sigma_i)$ and for all $j \neq i$ we have that $\|x - \mu_i\|_2 \leq \|x - \mu_j\|_2$. This is a result of the fact that when projecting $x$ onto the line that connects $\mu_i$ and $\mu_j$ the separation between $\mu_i$ and $\mu_j$ is overwhelmingly larger than $(\sigma_i + \sigma_j)\sqrt{\ln(n/\beta)}$ (a consequence of the separation condition combined with (7)).

5. Lastly, we also require that for any $i$ the number of points drawn from the $i$th Gaussian component is roughly $w_i n$. In other words, we assume that for all $i$ we have $\{x \in X : \ x \text{ drawn from } i\}$ is in the range $(\frac{3w_i n}{4}, \frac{5w_i n}{4})$. Standard use of the multiplicative Chernoff bound suggests that when $n = \Omega(\log(1/\beta)/w_i)$ then for each cluster $i$ the required holds.

Thus, we have (informally) argued that, with probability $\geq 1 - 5\beta k$, the given dataset satisfies all of these properties. Next we argue a few structural propositions that follow.

**Proposition 5.10.** *Let $B_r(p)$ be a ball such that for some $i$, both $\mu_i$ and some $x \sim \mathcal{N}(\mu_i, \Sigma_i)$ lie in $B_r(p)$. Then $B_{4r}(p)$ holds all datapoints drawn from $\mathcal{N}(\mu_i, \Sigma_i)$.*

*Proof.* Based on Property 1 it follows that $B_{\sqrt{2}\|\mu_i - x\|_2}(\mu_i)$ holds all datapoints drawn from the $i$th Gaussian. As $\|\mu_i - x\|_2 \leq 2r$ we have that $B_{\sqrt{2}\|\mu_i - x\|_2}(\mu_i) \subset B_{4r}(p)$. $\quad\square$

**Proposition 5.11.** *Let $B_r(p)$ be a ball such that for some $i$, two points $x, y \sim \mathcal{N}(\mu_i, \Sigma_i)$ lie in $B_r(p)$. Then $B_{4r}(p)$ holds all datapoints drawn from $\mathcal{N}(\mu_i, \Sigma_i)$.*

*Proof.* Based on Property 2, we have that the ball $B_{\sqrt{2}\|x - y\|_2}(x)$ holds all datapoints drawn from the $i$th Gaussian. As $\|x - y\|_2 \leq 2r$ it follows that $B_{\sqrt{2}\|x - y\|_2}(x) \subset B_{4r}(p)$. $\quad\square$

The remainder of the utility proof also relies on the assumption that at all levels of the recursion, all subprocedures (which are probabilistic in nature) do not terminate per-chance or with a non-likely output. Note that this assumption relies in turn on a sample size assumption: we require that $\frac{w_{\min} n}{2}$ is large enough for us to find a terrific ball, namely, we require

$$n = \tilde{\Omega}\left( \left( \frac{\sqrt{d}\log(d/\beta)\log(1/\delta)\log\log((R + \sqrt{d}\sigma_{\max})/\sigma_{\min})}{w_{\min}\varepsilon} \right)^{\frac{1}{1-a}} \right)$$

for some small constant $a > 0$ (say $a = 0.1$). Recall that at any level of the recursion we deal with at most two such subprocedures. As we next argue, (a) at each level of the recursion we partition the data into two parts, which are each laminar with a mixture of $k' < k$ Gaussians, and (b) when

the given input contains solely points from a single Gaussian we reach the bottom level of the recursion. It follows that the recursion tree (a binary tree with $k$ leaves) has $2k$ nodes. Since each instantiation of the recusion has failure probability at most $6\beta$ (a consequence of Claim 5.12, Claim 5.13, Claim 5.14, Corollary 5.15, Claim 5.16, and Claim 5.17), this implies that that with probability $\geq 1 - 12k\beta$, all subroutines calls ever invoked by RPGMP are successful. Note that the overall success probability of Algorithm 4 is thus $1 - 19k\beta$ (with probability $\leq 5k\beta$, there exists some cluster, whose datapoints don't satisfy properties 1-5; with probability $\leq 12k\beta$, a call made through Step 2 of Algorithm 4 fails; and with probability $\leq 2k\beta$, a failure occurs at Steps 3 and 4 of Algorithm 4). We continue assuming no such failure happens.

Subject to successfully performing each subprocedure, the following claims yield the overall algorithm's correctness.

**Claim 5.12.** *Let $(X_1, \ldots, X_n)$ be samples from a mixture of $k$ Gaussians in PGME. Suppose $X$ (a subset of the samples) is the input dataset in RPGMP, such that (1) for each $1 \leq i \leq k$, the number of points in $X$ belonging to component $i$ is either equal to 0 or greater than $\Omega(n w_{\min})$; and (2) $X$ contains points from at least one component. If*

$$n \geq \left(\frac{\sqrt{d}k}{\varepsilon}\right)^{\frac{1}{1-a}} \cdot 9^{\log^*\left(\sqrt{d}\left(\frac{R\sigma_{\max}}{\sigma_{\min}}\right)^d\right)} \cdot \mathrm{polylog}\left(d, \frac{1}{\varepsilon}, \frac{1}{\delta}\frac{1}{\beta}, \frac{1}{\gamma}\right) + O\left(\frac{d + \log(k/\beta)}{w_{\min}} + \frac{\log(1/\beta)}{\varepsilon w_{\min}}\right),$$

*where $a > 0$ is an arbitrarily small constant, then with probability at least $1 - \beta$, the ball $B_p(r)$ found in Step 2 of RPGMP contains all points in $X$, and $r \leq c r_{opt}$, where $c > 0$ is a constant (which depends on $a$), and $r_{opt}$ is the radius of the smallest ball that contains all points in $X$.*

*Proof.* First, note that from Lemma D.1, we know that with probability at least $1 - \frac{\beta}{2}$, the magnitude of the Laplace noise added in Step 2 of RPGMP is at most $\frac{2}{\varepsilon}\log(2/\beta)$. Since for every component that has points in $X$, there are at least $\Omega(n w_{\min})$ points, we know from our bound on $n$ that this magnitude of Laplace noise is at most $\frac{n w_{\min}}{20}$. Therefore,

$$n' = n + \mathrm{Lap}(2/\varepsilon) - \frac{n w_{\min}}{20}$$
$$\geq n\left(1 - \frac{w_{\min}}{10}\right) \geq n/2,$$

which means that by asking PGLOC to search for $n'$ points, we are covering at least half of every component that has points in $X$.

Next, by Theorem 4.11, we know that with probability at least $1 - \frac{\beta}{2}$, if the radius of the smallest ball covering $n'$ points is $r'$, we will get a ball of radius $r'' = O(r')$ that covers at least $\frac{n'}{2}$ points. Let $B_{r''}(p)$ be the returned ball. There are two cases to consider: (1) the ball covers at least two points from all components in $X$; and (2) it covers at most one point from at least one component in $X$ (which we call "completely uncovered" for brevity). In the first case, because all points in $X$ satisfy Property 2, multiplying $r''$ by 4 would cover all the points in $X$. In the second case, since $n'$ is very large, there must be at least two points from every completely uncovered component in $X \setminus B_{r''}(p)$, that lie together in some optimal ball containing $n'$ points, and one point in $X \cap B_{r''}(p)$ that lies in the same optimal ball as those two points. Consider

any completely uncovered component, and let $y, z$ be two such points from it, with $x$ being a corresponding point in $X \cap B_{r''}(p)$. Then by triangle inequality,

$$
\begin{aligned}
\left\| p - y \right\|_2 &\le \left\| p - x \right\|_2 + \left\| x - y \right\|_2 \\
&\le r'' + 2r' \\
&\le 3r'',
\end{aligned}
$$

which also holds for $\| p - z \|_2$. Therefore, multiplying $r''$ by 3 would cover both $y$ and $z$. Since the choice of completely uncovered components was arbitrary, this holds for all completely uncovered components. Again, since all points in $X$ satisfy Property 2, multiplying $3r''$ by 4 (to get $r$) fully covers all points in completely uncovered components, and all other points in $X$ as well.

Finally, $r' \le r_{opt}$, which means that $r \in O(r_{opt})$. This completes our proof of the claim. $\qquad \square$

We now describe our strategy for proving the main theorem of this subsection (which will be Theorem 5.18 below). We will begin by establishing a series of claims (first in prose, then with formal statements), and show how they imply our main theorem statement. We then conclude the subsection by substantiating our claims with proofs.

First, we argue that if the mixture contains at least two components and Step 3 of the algorithm finds a terrific ball to split the data (in the original, non-projected space), then the partition that is induced by this ball will split the data in a laminar fashion. That is, it will call the algorithm recursively on two disjoint subsets of the data, such that for each component, all samples which were drawn from this component end up in only one of the two subsets. This will ensure that the two recursive calls operate on valid (sub)instances of the GMM learning problem.

**Claim 5.13.** *If the dataset $X$ contains samples from a mixture of $k \ge 2$ Gaussians, and Step 3 of RPGMP finds a terrific ball over the data, then with probability at least $1 - \beta$, the partition formed by the terrific ball is laminar.*

On the other hand, if we still have at least two components and the algorithm is unable to find a terrific ball, we argue that all points lie in a ball of limited radius. Since the sample complexity of our private PCA algorithm (Lemma 3.1 and Lemma 3.2) depends on the radius of this ball, this will allow us to bound the sample complexity of this step.

**Claim 5.14.** *If the dataset $X$ contains samples from a mixture of $k \ge 2$ Gaussians with $\sigma_{\max}^2$ denoting the largest directional variance of any component in the mixture, and Step 3 of RPGMP does not find a terrific ball over the data, then with probability at least $1 - \beta$, the radius $r$ of the entire instance (found in Step 2) is at most $k^{4.53}\sqrt{d}\,\sigma_{\max}$, that is, all points of $X$ lie in a ball of radius $k^{4.53}\sqrt{d}\,\sigma_{\max}$.*

In the same setting, after the PCA projection, we have that the projected means are close to the original means, and that the resulting data will have a terrific ball (due to the separation between components).

**Corollary 5.15.** *Under the same conditions as in Claim 5.14, we have that if*

$$
n \ge O\left( \frac{k^{9.06}d^{3/2}\sqrt{\log(2/\delta)}\log(1/\beta)}{w_{\min}\varepsilon} \right),
$$

*then with probability at least $1 - \beta$, we have that for each center $\mu_i$, the corresponding projected center $\hat{\mu}_i$ is within distance $\leq \frac{3\sigma_{\max}}{\sqrt{w_i}}$. As a result, under our center-separation condition, the projected data $X\Pi$ has a $21$-terrific ball.*

Similar to Claim 5.13, we conclude this case by arguing that the partition formed by the resulting terrific ball to split the data (in the projected space) is laminar, resulting in the two recursive calls being made to valid (sub)instances of the problem.

**Claim 5.16.** *Under the same conditions as in Claim 5.14, with probability at least $1 - \beta$, the partition formed by the terrific ball found on the projected data $X\Pi$ is laminar.*

Finally, if our dataset is generated from a single component, then we will not locate a terrific ball, and the recursion ceases.

**Claim 5.17.** *For any $i$, if the dataset $X$ is composed of at least*

$$\frac{400 w_i d \log(1/\beta w_{\min})}{w_{\min}} + \Omega\left(\frac{1}{\varepsilon}\left(\log\log\left((R + \sqrt{d}\sigma_{\max})/\sigma_{\min}\right) + \log(1/\beta)\right)\right)$$

*points drawn only from $\mathcal{N}(\mu_i, \Sigma_i)$, then with probability at least $1 - \beta$, neither Step 3 nor Step 6 will locate a terrific ball.*

Putting together all of these claims and the entire discussion, we have that following theorem.

**Theorem 5.18.** *Algorithm 3 satisfies $\left(8\varepsilon\sqrt{2k\log(1/\delta)}\right), 8k\delta\right)$-DP, and under the center-separation of (6), with probability $\geq 1 - 12k\beta$, it returns $k$-subsets which are laminar with the $k$ clusters, while omitting no more than $\frac{\alpha w_{\min} n}{20\log(1/\alpha)}$ of the datapoints, provided that*

$$n = \tilde{\Omega}\left(\left(\frac{\sqrt{d}\log(d/\beta)\log(1/\delta)\log\log((R + \sqrt{d}\sigma_{\max})/\sigma_{\min})}{w_{\min}\varepsilon}\right)^{\frac{1}{1-a}}\right) \text{ for an arbitrary constant } a > 0$$

$$n = \Omega\left(\frac{k^{9.06}d^{3/2}\sqrt{\log(2/\delta)}\log(1/\beta)}{w_{\min}\varepsilon}\right)$$

$$n = \Omega\left(\frac{d\log(1/\beta w_{\min})}{w_{\min}}\right)$$

$$n = \Omega\left(\frac{k\log(1/\alpha)\log(\log((R + \sqrt{d}\sigma_{\max})/\sigma_{\min})/\beta)}{\alpha\varepsilon w_{\min}}\right).$$

The proof of Theorem 5.18 follows from all the above mentioned properties of the data and the claims listed above. It argues that in each level of the recursion we are forced to make a laminar partition with probability at least $1 - 6\beta$ (conditioned on the success in the previous levels) until we reach a subset (of sufficient size) which is contained in a single cluster, then we halt. Since this implies that we ultimately return $k$ clusters, it means that there are at most $2k$ nodes in the recursion tree, so the failure probability adds up to $12k\beta$. The sample complexity bounds are the bounds required for all claims and properties 1-5, where the last bound guarantees that the total number of points omitted in the overall execution of the algorithm doesn't exceed $\frac{n w_{\min}\alpha}{20\log(1/\alpha)}$ (at most $O(k\Gamma) = O(\frac{k}{\varepsilon}\log(\log((R + \sqrt{d}\sigma_{\max})/\sigma_{\min})/\beta))$, since the recursion tree has at most $k$ non-leaf nodes).

*Proof of Claim 5.13.* Let $B_r(p)$ be the ball returned by Step 3 of the RPGMP algorithm. Let $x, y$ be two datapoints that lie inside the ball and belong to the same component $i$. It follows from Proposition 5.11 that all datapoints from cluster $i$ lie inside $B_{4r}(p)$, but since the annulus $B_{5r}(p) \setminus B_r(p)$ is effectively empty (contains significantly less than $\frac{nw_{\min}\alpha}{20k\log(1/\alpha)}$ points), it should be the case that (almost) all of these datapoints lie in $B_r(p)$ itself, and in particular, no point from component $i$ lies outside of $B_{5r}(p)$.

It follows that any component $i$ with at least two datapoints that fall inside $B_r(p)$ belongs to one side of the partition, and moreover, since the ball contains $> \frac{nw_{\min}}{4}$ datapoints, there exists at least one component $i$, such that all of its datapoints lie inside $B_{4r}(p)$.

Next, suppose that for some $j \neq i$, some datapoint $z$ drawn from the $j^{\text{th}}$ component lies inside $B_r(p)$. It follows that both $z$ and $\mu_i$ lie inside $B_r(p)$, and so

$$2r \geq \|z - \mu_i\|_2 \geq \|z - \mu_j\|_2 \geq \sqrt{\frac{3}{4}\operatorname{tr}(\Sigma_j)},$$

thus $r \geq \sqrt{\frac{3}{16}\operatorname{tr}(\Sigma_j)}$. Thus the ball of radius $4r$ centered at $z$, which is fully contained inside $B_{5r}(p)$, has radius $\geq \sqrt{3\operatorname{tr}(\Sigma_j)}$. This is large enough to include all datapoints from cluster $j$ as well. Again, the fact that the annulus $B_{5r}(p) \setminus B_r(p)$ is effectively empty implies that (effectively) all points from cluster $j$ also belong to $B_r(p)$.

Lastly, note that $\geq nw_{\min}/4$ datapoints are left outside $B_{5r}(p)$. This implies that at least some component is left outside of $B_{5r}(p)$. Therefore, the partition $(A, B)$ formed by the terrific ball is a laminar partition of the dataset $X$.

The failure probability of this claim is $\beta$ using Lemma 5.7 because the success of the event in this claim rests on PTERRIFICBALL functioning correctly. $\qquad\square$

*Proof of Claim 5.14.* Based on Property 1, for any datapoint in $X$ and its respective center we have that their distance is at most $s \overset{\text{def}}{=} \sqrt{1.5d}\sigma_{\max}$. Now, since we know that our procedure that looks for a 5-terrific ball failed, then by Claim 5.6 we know that the data holds no 21-terrific ball.

Consider the following function $f : \mathbb{N} \to [k]$,

$$f(i) = \min_{\substack{T \subseteq [k], \\ X \subseteq \bigcup_{j \in T} B_{is}(\mu_j)}} |T|,$$

that is, the minimum number of balls of radius $i \cdot s$ centered at some $\mu_j$ that are required to cover all datapoints in $X$. The above paragraph implies that $f(1) \leq k$, so it is enough to show that $f(k^{\log_2(23)}) \leq 1$, as it would imply that a ball of radius $O(k^{\log_2(23)}\sqrt{d}\sigma_{\max})$ covers the entire instance. We argue that if there exists no 21-terrific ball in $X$, then for all $i$ such that $f(i) > 1$, then $f(23i) \leq f(i)/2$, which by induction leads to $f(23^{\log_2(k)}) = f(k^{\log_2(23)}) \leq 1$.

Fix $i$. Assume $f(i) > 1$, otherwise we are done. By definition, there exists a subset $\{\mu_{j_1}, \mu_{j_2}, ..., \mu_{j_{f(i)}}\}$ of the $k$ centers, such that $X$ is contained in $\bigcup_t B_{is}(\mu_{j_t})$. Pick any center $\mu$ in this subset. We know that $B_{is}(\mu)$ is not a 21-terrific ball. Since it holds enough points (at least $nw_{\min}$) and leaves out enough points (since $f(i) > 1$), it must be the case that $B_{21is}(\mu) \setminus B_{is}(\mu)$ is non-empty, that is, there exists a point $x \in X$ that resides in $B_{21is}(\mu) \setminus B_{is}(\mu)$. This means that $B_{22is}(\mu)$ holds the

center $\mu'$ of the ball $B_{is}(\mu')$ that covers $x$, and therefore $B_{is}(\mu') \subset B_{23is}(\mu)$. Note that by symmetry it also holds that $\mu \in B_{22is}(\mu')$ and so $B_{is}(\mu) \subset B_{23is}(\mu')$.

Now, draw a graph containing $f(i)$ nodes (one node per center $\mu_{j_i}$), and connect two nodes $\mu$ and $\mu'$ if $B_{is}(\mu) \subset B_{23is}(\mu')$. This is a graph in which each node has degree $\geq 1$ because there is no 21-terrific ball centered at the corresponding mean, and therefore, has a dominating set of size $\leq f(i)/2$. Hence, $X$ is covered by balls of radius $23is$ centered at each node in this dominating set, implying that $f(23i) \leq f(i)/2$.

Finally, by Claim 5.12, we have that the radius $r$ found is Step 2 is in $O(k^{\log_2(23)}\sqrt{d}\sigma_{\max}) \in O(k^{4.53}\sqrt{d}\sigma_{\max})$.

As before, the failure probability of this claim is $\beta$ using Lemma 5.7. $\qquad\square$

*Proof of Corollary 5.15.* Under the same conditions as in Claim 5.14, it holds that the added noise is such that $\|N\|_2 \leq \frac{2r^2\sqrt{d\ln(2/\delta)}\log(1/\beta)}{\varepsilon}$ with probability $\geq 1-\beta$. Using the fact that $r = k^{4.53}\sqrt{d}\sigma_{\max}$, Lemmata 3.1 and 3.2, along with our bound on $n$ (hence, on $n_i$), imply that with probability at least $1 - \beta$, for any $i$,

$$
\begin{aligned}
\|\bar{\mu}_i - \Pi\bar{\mu}_i\| &\leq \sqrt{\frac{1}{n_i}\|A - C\|^2} + \sqrt{\frac{2r^2\sqrt{d\ln(2/\delta)}\log(1/\beta)}{\varepsilon n_i}} \\
&\leq \frac{1}{\sqrt{w_i}}\left(4\sqrt{2}\sigma_{\max} + \sigma_{\max}\right) \\
&\leq 7\frac{\sigma_{\max}}{\sqrt{w_i}} \\
&\leq 7\frac{\sigma_{\max}}{\sqrt{w_{\min}}}.
\end{aligned}
$$

Without loss of generality, assume cluster 1 is the Gaussian of largest variance. It follows that for all $i \neq 1$, we have

$$
\|\Pi\mu_1 - \Pi\mu_i\|_2 \geq 100(\sigma_1 + \sigma_i)\left(\sqrt{k\log(n)} + \frac{1}{\sqrt{w_1}} + \frac{1}{\sqrt{w_i}}\right) - 7\left(\frac{\sigma_1}{\sqrt{w_1}} + \frac{\sigma_1}{\sqrt{w_i}}\right) \geq 100\sigma_1\sqrt{k\log(n)}.
$$

Yet, similar to the analysis in [VW02, AM05], we have that $\|\Pi(x - \mu_i)\|_2 \leq \sqrt{k\sigma_i^2\log(n)}$ for each datapoint and its respective cluster $i$.[5] Roughly, the argument is that, if we draw a sample from a Gaussian in a $k$ dimensional space, its $\ell_2$ distance from its mean is $O(\sigma\sqrt{k\log n})$. The same argument holds if the sample is drawn in a $d$ dimensional space and then projected into $k$ dimensions, as long as the projection is independent of the sample. While the projection is dependent on the data, the fact that it was computed in a differentially private manner allows us to act as though it is independent.

This implies that all points that belong to cluster 1 fall inside the ball $B_{4\sqrt{k\sigma_{\max}^2\log(n)}}(\Pi\mu_1)$, whereas any point from a different cluster must fall outside the ball $B_{90\sqrt{k\sigma_{\max}^2\log(n)}}(\Pi\mu_1)$, with each side containing at least $\frac{nw_{\min}}{2}$ datapoints. $\qquad\square$

*Proof of Claim 5.16.* Again, as in the proof of Corollary 5.15, we leverage the fact that using the projection, we have that $\|\Pi(x - \mu_i)\|_2 \le \sqrt{k\sigma_i^2 \log(n)}$ for each datapoint $x$ belonging to cluster $i$. Note that we run the PTerrificBall procedure using the flag $largest = $ TRUE. As a result, based on Claim 5.6 and Corollary 5.15, we are guaranteed that the radius of the ball returned is at least as large as the radius of the terrific ball that holds all points from the cluster having the largest directional variance $\sigma_{\max}^2$ (without loss of generality, let that be cluster 1). In other words, we have that the radius of the terrific ball is at least $4\sqrt{k\sigma_{\max}^2 \log(n)}$.

Therefore, for any cluster $i$, the radius is large enough so that the ball of radius $4r$ holds all datapoints from cluster $i$. Again, the annulus $B_{5r}(p) \setminus B_r(p)$ holds very few points, and so almost all datapoints of cluster $i$ lie inside $B_r(p)$, and none of its points lie outside of $B_{5r}(p)$.

Again, the failure probability of this claim is $\beta$ using Lemma 5.7. $\qquad\square$

*Proof of Claim 5.17.* At a high level, the proof goes as follows. Suppose our dataset was generated according to a single Gaussian, and that Step 3 or 6 produces a terrific ball. Perform the following thought experiment: take the data, apply some univariate projection, and partition the line into 5 intervals (corresponding to the ball of radius $r$/diameter $2r$, the two surrounding intervals of diameter $4r$, and the two infinite intervals which extend past those points). For the ball to be terrific, there must be a projection such that the first interval has significantly many points, the second and third intervals have almost no points, and the last two intervals (collectively) have many points. Given the structure of a Gaussian, we know that no such projection and decomposition could exist when considering the *true* probability measure assigned by the Gaussian distribution, since the distribution is unimodal with quickly decaying tails. However, with some small probability, this could happen with respect to the empirical set of samples. For a given projection, we bound this probability by partitioning the line into intervals of appropriate width, and then applying a Chernoff and union bound style argument. We bound the overall probability by taking a union bound over a net of possible directions. A more formal argument follows.

First, due to the proof of Claim 5.13 we know that if a terrific ball $B_r(p)$ is found in Step 3 then its radius is large enough to hold all datapoints from the same cluster. Therefore, in the case where all datapoints are taken from the same Gaussian we have that no points lie outside of $B_{5r}(p)$.

Next, we argue something slightly stronger. Note that a 4-terrific ball, either on the data on over the projected data, implies that there's some direction (unit-length vector) $v$ — namely, a line going from the origin through the center of the ball — such that on projecting the data onto $v$ we have an interval of length $2r$ holding at least $\frac{nw_{\min}}{3}$ datapoints, surrounded by intervals of length $3r$ of both sides with very few points (quantified by the guarantees of Lemma 5.7), and the remainder of the line also holds $\frac{nw_{\min}}{3}$ datapoints. (The same holds for a ball for the projected points since this ball lies in some $k$-dimensional subspace.) However, since the datapoints are all drawn from the same Gaussian, its projection over $v$ also yields a (one-dimensional) Gaussian, with the property that for any three ordered intervals from left to right of the same length $I_1, I_2, I_3$, the probability mass held in $I_2$ is greater than the minimum between the probability held in $I_1$ and the probability mass held in $I_3$. We thus have that a necessary condition for the existence of such terrific ball is that there exists a direction $v$ and an interval $\ell$ which should have

a probability mass of at least $w_{\min}/10$ yet contains less than $w_{\min}/20$ fraction of the empirical mass.

We now apply classical reasoning: a Chernoff and union argument. Fix a cluster $i$. If we are dealing with a set of datapoints drawn only from the $i^{\text{th}}$ Gaussian, then this set has no more than $2w_i n$ points, and so the ratio $\lambda = \frac{w_{\min}}{20w_i}$ represents the fraction of points that ought to fall inside the above-mentioned interval. We thus partition the distribution projected onto direction $v$ into $2/\lambda$ intervals each holding $\lambda/2$ probability mass. The Chernoff bound guarantees that if the number of points from cluster $i$ is at least $\frac{200d \log(1/\beta\lambda)}{\lambda}$ then a given interval has empirical probability sufficiently close (up to a multiplicative factor of 2) to its actual probability mass. Applying the union-bound over all $\frac{2}{\lambda}$ intervals times all $2^{O(d)}$ unit-length vectors in a cover of the unit-sphere, we get that such an interval exists with probability $\leq \beta$. Note that the number of points of cluster $i$ is at least $\frac{nw_i}{2}$, so by re-plugging the value of $\lambda$ into the bound we get that it is enough to have $\frac{nw_i}{2} \geq \frac{200 w_i d \log(1/\beta w_{\min})}{w_{\min}}$, implying that $n \geq \frac{400d \log(1/\beta w_{\min})}{w_{\min}}$ is a sufficient condition to have that no such ill-represented interval exists, and as a result, no terrific ball exists. $\qquad\square$

## 5.3 Estimation

We want to show that once we have clusters $C_1, \ldots, C_k$ from RPGMP, such that each cluster contains points from exactly one Gaussian, no two clusters contain points from the same Gaussian, and that the fraction of points lost from the dataset is at most

$$\tau = O\left(\frac{w_{\min}\alpha}{\log(1/\alpha)}\right),$$

we can learn individual Gaussians and mixing weights accurately. We use the learner for $d$-dimensional Gaussians from [KLSU19] in the process.

**Theorem 5.19** (Learner from [KLSU19]). *For every $\alpha, \beta, \varepsilon, \delta, \sigma_{\min}, \sigma_{\max}, R > 0$, there exists an $(\varepsilon, \delta)$-differentially private algorithm $\mathrm{M}_{\mathrm{KLSU}}$, which if given $m$ samples from a $d$-dimensional Gaussian $\mathcal{N}(\mu, \Sigma)$, such that, $m \geq m_1 + m_2$, where,*

$$m_1 \geq O\left(\frac{d^2 + \log(\frac{1}{\beta})}{\alpha^2} + \frac{d^2 \mathrm{polylog}(\frac{d}{\alpha\beta\varepsilon\delta})}{\alpha\varepsilon} + \frac{d^{\frac{3}{2}} \log^{\frac{1}{2}}(\frac{\sigma_{\max}}{\sigma_{\min}}) \mathrm{polylog}(\frac{d \log(\sigma_{\max}/\sigma_{\min})}{\beta\varepsilon\delta})}{\varepsilon}\right)$$

$$m_2 \geq O\left(\frac{d \log(\frac{d}{\beta})}{\alpha^2} + \frac{d \log(\frac{d \log R \log(1/\delta)}{\alpha\beta\varepsilon}) \log^{\frac{1}{2}}(\frac{1}{\delta})}{\alpha\varepsilon} + \frac{\sqrt{d} \log(\frac{Rd}{\beta}) \log^{\frac{1}{2}}(\frac{1}{\delta})}{\varepsilon}\right)$$

*and $\sigma_{\min}^2 \leq \Sigma \leq \sigma_{\max}^2$ and $\|\mu\|_2 \leq R$, outputs $\widehat{\mu}, \widehat{\Sigma}$, such that with probability at least $1 - \beta$,*

$$\mathrm{d}_{\mathrm{TV}}(\mathcal{N}(\mu, \Sigma), \mathcal{N}(\widehat{\mu}, \widehat{\Sigma})) \leq \alpha.$$

Now, we are in a situation where at most $n\tau$ samples get lost from each component in the clustering process. So, we need a more robust version of $\mathrm{M}_{\mathrm{KLSU}}$ that works even when we lose a small fraction of the points. The following theorem guarantees the existence of one such robust learner.

**Theorem 5.20.** *For every $\alpha, \beta, \varepsilon, \delta, \sigma_{\min}, \sigma_{\max}, R > 0$, there exists an $(\varepsilon, \delta)$-differentially private algorithm PGE with the following guarantee. Let $(X_1, \ldots, X_n)$ be independent samples from a d-dimensional Gaussian $\mathcal{N}(\mu, \Sigma)$, where $\sigma_{\min}^2 \preceq \Sigma \preceq \sigma_{\max}^2$ and $\|\mu\|_2 \leq R$ and $n \geq \frac{1}{w_{\min}}(n_1 + n_2)$, for*

$$n_1 \geq O\left( \frac{(d^2 + \log(\frac{1}{\beta}))\log^2(1/\alpha)}{\alpha^2} + \frac{(d^2\mathrm{polylog}(\frac{d}{\alpha\beta\varepsilon\delta}))}{\alpha\varepsilon} + \frac{d^{\frac{3}{2}}\log^{\frac{1}{2}}(\frac{\sigma_{\max}}{\sigma_{\min}})\mathrm{polylog}(\frac{d\log(\sigma_{\max}/\sigma_{\min})}{\beta\varepsilon\delta})}{\varepsilon} \right)$$

$$n_2 \geq O\left( \frac{d\log(\frac{d}{\beta})\log^2(1/\alpha)}{\alpha^2} + \frac{d\log(\frac{d\log R\log(1/\delta)}{\beta\varepsilon})\log^{\frac{1}{2}}(\frac{1}{\delta})\log^2(1/\alpha)}{\alpha\varepsilon} + \frac{\sqrt{d}\log(\frac{Rd}{\beta})\log^{\frac{1}{2}}(\frac{1}{\delta})}{\varepsilon} \right).$$

*For every $S \subseteq [n]$ with $|S| \geq n(1 - O(\alpha/\log(1/\alpha)))$, when PGE is given $\{X_i\}_{i \in S}$ as input, with probability at least $1 - \beta$,*

$$d_{\mathrm{TV}}(\mathcal{N}(\mu, \Sigma), \mathcal{N}(\widehat{\mu}, \widehat{\Sigma})) \leq \alpha.$$

The proof of this theorem follows effectively the same structure as that of Theorem 5.19, the primary difference in the setting being that a miniscule fraction of points have been removed. However, fortunately, the proof of Theorem 5.19 uses the Gaussianity of the data essentially only to show concentration of the empirical mean and covariance in various norms, which are preserved even against an adversary who can delete a small fraction of the points (see, i.e., Lemma 4.3, 4.4, and Corollary 4.8 of [DKK$^+$16]). Substituting these statements into the proof, it follows essentially the same.

Now, we give a lemma that says that the mixing weights are accurately estimated.

**Lemma 5.21.** *Suppose $w_1, \ldots, w_k$ are the mixing weights of the Gaussians of the target distribution $\mathcal{D} \in \mathcal{G}(d, k)$. Let $\widehat{w}_1, \ldots, \widehat{w}_k$ be their respective estimations produced by the algorithm. If*

$$n \geq O\left( \frac{k^2}{\varepsilon\alpha}\ln(k/\beta) \right),$$

*then with probability at least $1 - O(\beta)$,*

$$\forall\ 1 \leq i \leq k, \quad |\widehat{w}_i - w_i| \leq \frac{\alpha}{3k}.$$

*Proof.* Let $w_1', \ldots, w_k'$ be the empirical weights of the Gaussians produced using the points in $X$. We have the following claim for them.

**Claim 5.22.** *Let $X$ be the dataset as in the algorithm. If*

$$n \geq O\left( \frac{k^2}{\varepsilon\alpha}\ln(k/\beta) \right),$$

*and $X$ satisfies Condition 2.5, then for $w_i \geq \frac{4\alpha}{9k}$,*

$$\left|w_i - w_i'\right| \leq \frac{\alpha}{9k},$$

*and for $w_i < \frac{4\alpha}{9k}$,*

$$\left|w_i - w_i'\right| \leq \frac{2\alpha}{9k},$$

*Proof.* There are two sources of error in this case: (1) adding Laplace noise; and (2) by having lost $\tau$ points from $X$. Let $\widetilde{n}_i$ be as defined in the algorithm, $\bar{n}_i = |C_i|$, and $n_i$ be the number of points of this component in $X$.

First, we want to show that the if the number of points is large enough, then the added noise does not perturb the empirical weights too much. Now, using Lemma D.1 with our bound on $n$, and applying the union bound over all calls to PCOUNT, we get that with probability at least $1 - O(\beta)$,

$$\forall\, i, \quad |\widetilde{n}_i - \bar{n}_i| \leq \frac{n\alpha}{40k^2}.$$

Also, we know that $\sum\limits_{i=1}^{k} \bar{n}_i \geq n(1 - \tau)$.

Now, let $e = \frac{\alpha}{40k^2}$. From the above, using triangle inequality, we get that for all $i$,

$$\left|\widehat{w}_i - w_i'\right| = \left| \frac{\widetilde{n}_i}{\sum\limits_{j=1}^{k} \widetilde{n}_j} - \frac{n_i}{n} \right|$$

$$\leq \left| \frac{\widetilde{n}_i}{\sum\limits_{i=1}^{k} \widetilde{n}_i} - \frac{\bar{n}_i}{\sum\limits_{i=1}^{k} \bar{n}_i} \right| + \left| \frac{\bar{n}_i}{\sum\limits_{i=1}^{k} \bar{n}_i} - \frac{n_i}{n} \right|$$

$$\leq \frac{e + ke}{1 - ke} + \frac{n_i}{n}\left| \frac{1}{1 - \tau} - 1 \right|$$

$$\leq \frac{\alpha}{18k} + \frac{\alpha}{18k}$$

$$\leq \frac{\alpha}{9k},$$

Where the second to last inequality holds because $\tau \leq \frac{\alpha}{20k}$. □

Because $\left| w_i - w_i' \right| \leq \frac{\alpha}{9k}$, using triangle inequality, we get the required result. □

Combining these statements is sufficient to conclude Theorem 5.1.

## 6 Sample and Aggregate

In this section, we detail methods based on sample and aggregate, and derive their sample complexity. This will serve as a baseline for comparison with our methods.

A similar sample and aggregate method was considered in [NRS07], but they focused on a restricted case (when all mixing weights are equal, and all components are spherical with a known variance), and did not explore certain considerations (i.e., how to minimize the impact of a large domain). We provide a more in-depth exploration and attempt to optimize the sample complexity.

The main advantage of the sample and aggregate method we describe here is that it is extremely flexible: given any non-private algorithm for learning mixtures of Gaussians, it can

immediately be converted to a private method. However, there are a few drawbacks, which our main algorithm avoids. First, by the nature of the approach, it will increase the sample complexity multiplicatively by $\Omega(\sqrt{d}/\varepsilon)$, thus losing any chance of the non-private sample complexity being the dominating term in any parameter regime. Second, it is not clear on how to adapt this method to non-spherical Gaussians. We rely on the methods of [NSV16, NS18], which find a small $\ell_2$-ball which contains many points. The drawback of these methods is that they depend on the $\ell_2$-metric, rather than the (unknown) Mahalanobis metric as required by non-spherical Gaussians. We consider aggregation methods which can handle settings where the required metric is unknown to be a very interesting direction for further study.

Our main sample-and-aggregate meta-theorem is the following.

**Theorem 6.1.** *Let* $m = \tilde{\Theta}\left(\frac{\sqrt{kd}+k^{1.5}}{\varepsilon}\log^2(1/\delta)\cdot 2^{O\left(\log^*\left(\frac{dR\sigma_{\max}}{\alpha\sigma_{\min}}\right)\right)}\right)$. *Suppose there exists a (non-private) algorithm with the following guarantees. The algorithm is given a set of samples* $X_1,\ldots,X_n$ *generated i.i.d. from some mixture of $k$ Gaussians* $\mathcal{D} = \sum_{i=1}^{k} w_i \mathcal{N}(\mu_i, \sigma_i^2 \mathbb{I}_{d\times d})$, *with the separation condition that* $\|\mu_i - \mu_j\|_2 \geq (\sigma_i + \sigma_j)\tau_{k,d}$, *where $\tau_{k,d}$ is some function of $k$ and $d$, and $\tau_{k,d} \geq c\alpha$, for some sufficiently large constant $c$. With probability at least $1 - m/100$, it outputs a set of points* $\{\hat{\mu}_1,\ldots,\hat{\mu}_k\}$ *and weights* $\{\hat{w}_1,\ldots,\hat{w}_k\}$ *such that* $\|\hat{\mu}_{\pi(i)} - \mu_i\|_2 \leq O\left(\frac{\alpha\sigma_i}{\sqrt{\log mk}}\right)$ *and* $|\hat{w}_{\pi(i)} - w_i| \leq O(\alpha/k)$ *for all $i \in [k]$, where* $\pi : [k] \to [k]$ *is some permutation.*

*Then there exists a $(\varepsilon,\delta)$-differentially private algorithm which takes $mn$ samples from the same mixture, and input parameters $R, \sigma_{\min}, \sigma_{\max}$ such that $\|\mu_i\|_2 \leq R$ and $\sigma_{\min} \leq \sigma_i \leq \sigma_{\max}$ for all $i \in [k]$. With probability at least $9/10$, it outputs a set of points* $\{\hat{\mu}_1,\ldots,\hat{\mu}_k\}$ *and weights* $\{\hat{w}_1,\ldots,\hat{w}_k\}$ *such that* $\|\hat{\mu}_{\pi(i)} - \mu_i\|_2 \leq O(\alpha\sigma_i)$ *and* $|\hat{w}_{\pi(i)} - w_i| \leq O(\alpha/k)$ *for all $i \in [k]$, for some permutation $\pi$.*

*Proof.* In short, the algorithm will repeat the non-private algorithm several times, and then aggregate the findings using the 1-cluster algorithm from [NSV16]. We will focus on how to generate the estimates of the means, and sketch the argument needed to conclude the accuracy guarantees for the mixing weights.

In more detail, we start by discretizing the space where all the points live, which is a set of diameter $\mathrm{poly}(R, \sigma_{\max}, d, \log n)$, at granularity $\mathrm{poly}\left(\frac{\alpha\sigma_{\min}}{d}\right)$, and every point we consider will first be rounded to the nearest point in this discretization. This will allow us to run algorithms which have a dependence on the size of the domain. Since this dependence will be proportional to the exponentiation of the iterated logarithm, we can take the granularity to be very fine by increasing the exponent of the polynomial, at a minimal cost in the asymptotic runtime. For clarity of presentation, in the sequel we will disregard accounting for error due to discretization.

Now, we use the following theorem of [NSV16].

**Theorem 6.2** ([NSV16]). *Suppose $X_1,\ldots,X_m$ are points from $S^d \subset \mathbb{R}^d$, where $S^d$ is finite. Let $m, t, \beta, \varepsilon, \delta$ be such that,*

$$t = \Omega\left(\frac{\sqrt{d}}{\varepsilon}\log\left(\frac{1}{\beta}\right)\log\left(\frac{md}{\beta\delta}\right)\sqrt{\log\left(\frac{1}{\beta\delta}\right)}\cdot 9^{\log^*(2|S|\sqrt{d})}\right).$$

*Let $r_{opt}$ be the radius of the smallest ball that contains at least $t$ points from the sample. There exists an $(\varepsilon,\delta)$-DP algorithm that returns a ball of radius at most $w \cdot r_{opt}$ such that it contains at least $t - \Delta$ points*

*from the sample with error probability $\beta$, where $w = O(\sqrt{\log m})$ and*

$$\Delta = O\left(\frac{1}{\varepsilon}\log\left(\frac{1}{\beta}\right)\log\left(\frac{m}{\delta}\right) \cdot 9^{\log^*(2|S|\sqrt{d})}\right).$$

Compare with the slightly different guarantees of Theorem B.1, which is the 1-cluster algorithm from [NS18]. We will use Theorem 6.2 $k$ times, with the following settings of parameters: their $\varepsilon$ is equal to our $O(\varepsilon/\sqrt{k\log(1/\delta)})$, their $\beta$ is equal to $O(1/k)$, their $|S|$ is our $\mathrm{poly}\left(\frac{dR\sigma_{\max}}{\sigma_{\min}\alpha}\right)$, their $m$ is equal to $mk$, and all other parameters are the same.

We start by taking the results of running the non-private algorithm $m$ times. We will run the algorithm of Theorem 6.2 with $t = m$ to obtain a ball (defined by a center and a radius). We remove all points within this ball, and repeat the above process $k$ times. Note that by advanced composition, the result will be $(\varepsilon, \delta)$-differentially private. We spend the rest of the proof arguing that we satisfy the conditions of Theorem 6.2, and that if we choose the $i$th mean to be an arbitrary point from the $i$th output ball, these will satisfy the desired guarantees.

First, we confirm that our choice of $t$ satisfies the conditions of the theorem statement. Since we set $t$ to be equal to $m = \tilde{\Theta}\left(\frac{\sqrt{kd}+k^{1.5}}{\varepsilon}\log^2(1/\delta) \cdot 2^{O\left(\log^*\left(\frac{dR\sigma_{\max}}{\alpha\sigma_{\min}}\right)\right)}\right)$, the "first term" (the one with the leading $\sqrt{kd}/\varepsilon$) is large enough so that $t$ will satisfy the necessary condition.

For the rest of the proof, we will reason about the state of the points output by the $m$ runs of the non-private algorithm. Note that the non-private algorithm will learn to a slightly better accuracy than our final private algorithm ($\alpha/\sqrt{\log mk}$, rather than $\alpha$). By a union bound, we know that all $m$ runs of the non-private algorithm will output mean estimates which are close to the true means with probability at least $99/100$, an event we will condition on for the rest of the proof. This implies that, around the mean of component $i$, there is a ball of radius $O\left(\frac{\alpha\sigma_i}{\sqrt{\log mk}}\right)$ which contains a set of $m$ points: we will call each of these point sets a *mean-set*. We will say that a mean-set is *unbroken* if all $m$ of its points still remain, i.e., none of them have been removed yet.

We claim that, during every run of the algorithm of Theorem 6.2, we will identify and remove points belonging solely to a single unbroken mean-set. First, we argue that the smallest ball containing $m$ points will consist of points solely from a single unbroken mean-set. There are two cases which could be to the contrary: if it contains points from one unbroken mean-set and another mean-set (either broken or unbroken), and if it contains points from only broken mean-sets. In the first case, the separation condition and triangle inequality imply that the ball consisting of points solely from the unbroken mean-set within this ball would be smaller. The second case is also impossible: this is because we require at least $m$ points in the ball, and during the $i$th iteration, there will be at most $(i-1)\Delta \le k\Delta$ points leftover from broken mean-sets (assuming that the claim at the start of this paragraph holds by strong induction). The "second term" of $m$ (the one with the leading $k^{1.5}/\varepsilon$) enforces that $m > k\Delta$, preventing this situation. Arguments similar to those for these two cases imply that any ball with radius equal to this minimal radius inflated by a factor of $w = O(\sqrt{\log mk})$ and containing $m - \Delta$ points must consist of points belonging solely to a single unbroken mean-set.

It remains to conclude that any point within a ball has the desired accuracy guarantee. Specifically, using any point within a ball as a candidate mean will approximate the true mean

of that component up to $O\left(\frac{\alpha\sigma_i}{\sqrt{\log mk}}\right)$. This is because the smallest ball containing an unbroken mean-set has radius at most $O\left(\frac{\alpha\sigma_i}{\sqrt{\log mk}}\right)$ (and we know that every point within this ball has the desired accuracy guarantee with respect to the true mean), and the algorithm will inflate this radius by a factor of at most $w = O(\sqrt{\log mk})$.

At this point, we sketch the straightforward argument to estimate the values of the mixing weights. The output of the non-private algorithm consists of pairs of mean and mixing weight estimates. By a union bound, all of the (non-private) mixing weights are sufficiently accurate with probability at least 99/100. In order to aggregate these non-private quantities into a single private estimate, we can use a stability-based histogram (see [KKMN09, BNSV15], and [Vad17] for a clean presentation). More precisely, for all the mean estimates contained in each ball, we run a stability-based histogram (with bins of width $O(\alpha/k)$) on the associated mixing weight estimates, and output the identity of the most populated bin.

We claim that the aggregated mixing weight estimates will all fall into a single bin with a large constant probability, simultaneously for each of the histograms. This is because all the mixing weight estimates are correct with probability 99/100, and the argument above (i.e., each run of the [NSV16] algorithm removes points belonging solely to a single unbroken mean-set) implies that we will aggregate mixing weight estimates belonging only to a single component. This guarantees the accuracy we desire. The cost in the sample complexity is dominated by the cost of the $k$ runs of the algorithm of [NSV16]. □

The following lemma can be derived from [VW02]. The first term of the sample complexity is the complexity of clustering from Theorem 4 of [VW02], the second and third terms are for learning the means and mixing weights, respectively.

**Lemma 6.3** (From Theorem 4 of [VW02]). *There exists a (non-private) algorithm with the following guarantees. The algorithm is given a set of samples $X_1,\ldots,X_n$ generated i.i.d. from some mixture of $k$ Gaussians $\mathcal{D} = \sum_{i=1}^k w_i \mathcal{N}(\mu_i,\sigma_i^2\mathbb{I}_{d\times d})$, with the separation condition that $\|\mu_i - \mu_j\|_2 \geq 14(\sigma_i + \sigma_j)(k\ln(4n/\beta))^{1/4}$. With probability at least $1 - \beta$, it outputs a set of points $\{\hat{\mu}_1,\ldots,\hat{\mu}_k\}$ and weights $\{\hat{w}_1,\ldots,\hat{w}_k\}$ such that $\|\hat{\mu}_{\pi(i)} - \mu_i\|_2 \leq O(\alpha\sigma_i)$ and $|\hat{w}_{\pi(i)} - w_i| \leq O(\alpha/k)$ for all $i \in [k]$, where $\pi : [k] \to [k]$ is some permutation. The number of samples it requires is $n = \tilde{O}\left(\frac{d^3}{w_{\min}^2}\log\left(\max_i \frac{|\mu_i|^2}{\sigma_i^2}\right) + \frac{d}{w_{\min}\alpha^2} + \frac{k^2}{\alpha^2}\right)$[6].*

From Theorem 6.1, this implies the following private learning algorithm.

**Theorem 6.4.** *There exists an $(\varepsilon,\delta)$-differentially private algorithm which takes $n$ samples from some mixture of $k$ Gaussians $\mathcal{D} = \sum_{i=1}^k w_i \mathcal{N}(\mu_i,\sigma_i^2\mathbb{I}_{d\times d})$, with the separation condition that $\|\mu_i - \mu_j\|_2 \geq (\sigma_i + \sigma_j)\tilde{\Omega}(k^{1/4} \cdot \mathrm{poly}\log(k,d,1/\varepsilon,\log(1/\delta),\log^*(\frac{R\sigma_{\max}}{\alpha\sigma_{\min}})))$, and input parameters $R,\sigma_{\min},\sigma_{\max}$ such that $\|\mu_i\|_2 \leq R$ and $\sigma_{\min} \leq \sigma_i \leq \sigma_{\max}$ for all $i \in [k]$. With probability at least 9/10, it outputs a set of points $\{\hat{\mu}_1,\ldots,\hat{\mu}_k\}$ and weights $\{\hat{w}_1,\ldots,\hat{w}_k\}$ such that $\|\hat{\mu}_{\pi(i)} - \mu_i\|_2 \leq O(\alpha\sigma_i)$ and $|\hat{w}_{\pi(i)} - w_i| \leq O(\alpha/k)$ for all $i \in [k]$, for some permutation $\pi$. The number of samples it requires is $n = \tilde{O}\left(\frac{\sqrt{kd}+k^{1.5}}{\varepsilon}\log^2(1/\delta) \cdot 2^{O\left(\log^*\left(\frac{dR\sigma_{\max}}{\alpha\sigma_{\min}}\right)\right)}\right)\left(\frac{d^3}{w_{\min}^2}\log\left(\max_i \frac{|\mu_i|^2}{\sigma_i^2}\right) + \frac{d}{w_{\min}\alpha^2} + \frac{k^2}{\alpha^2}\right).$*

We note that plugging more recent advances in learning mixtures of Gaussians [HL18a, KSS18, DKS18] into Theorem 6.1 allows us to derive computationally and sample efficient algorithms for separations which are $o(k^{1/4})$. However, we note that even non-privately, the specific sample and time complexities are significantly larger than what we achieve from our more direct construction.

## Acknowledgments

Part of this work was done while the authors were visiting the Simons Institute for Theoretical Computer Science. Parts of this work were done while GK was supported as a Microsoft Research Fellow, as part of the Simons-Berkeley Research Fellowship program, while visiting Microsoft Research, Redmond, and while supported by a University of Waterloo startup grant. This work was done while OS was affiliated with the University of Alberta. OS gratefully acknowledges the Natural Sciences and Engineering Research Council of Canada (NSERC) for its support through grant #2017-06701. JU and VS were supported by NSF grants CCF-1718088, CCF-1750640, and CNS-1816028.

## Footnotes

[1]We remark that there are also many popular *heurstics* for learning Gaussian mixtures, notably the EM algorithm [DLR77], but in this work we focus on algorithms with provable guarantees.

[2]To provide context, one might settle for a weaker goal of *proper learning* where the goal is merely to learn *some* Gaussian mixture, possibly with a different number of components, that is close to the true mixture, or *improper learning* where it suffices to learn *any* such distribution.

[3]These boundedness conditions are also provably necessary to learn even a single univariate Gaussian for pure differential privacy, concentrated differential privacy, or Rényi differential privacy, by the argument of [KV18]. One could only hope to avoid boundedness using the most general formulation of $(\varepsilon, \delta)$-differential privacy.

[4]Since [NSV16, NS18] call the ball found by their algorithm a *good* ball, we call ours a *terrific* ball.

[5] In fact, due to the fact that $\Pi$ was derived via in a differentially private manner, is is easier to argue this than in the [AM05] paper, see the following blog post.

[6]We note that [NRS07] states a similar version of this result, though their coverage omits dependences on the scale of the data.

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

# A    Proofs for Deterministic Regularity Conditions

**Lemma A.1.** *Suppose $X^L = ((X_1, \rho_1), \ldots, (X_n, \rho_n))$ are labelled samples from $\mathcal{D} \in \mathcal{G}(d, k, s, R, \kappa, w_{\min})$. If*

$$n \geq \frac{12}{w_{\min}} \ln(2k/\beta),$$

*then with probability at least $1 - \beta$, for every label $\rho_i$, for $1 \leq i \leq k$, the number of points having label $\rho_i$ is in*

$$\left[ \frac{nw_i}{2}, \frac{3nw_i}{2} \right].$$

*Proof.* It follows directly from Lemma D.3 by setting $p = w_i$, and taking the union bound over all mixture components.    □

**Lemma A.2.** *Suppose $X^L = ((X_1, \rho_1), \ldots, (X_n, \rho_n))$ are labelled samples from $\mathcal{D} \in \mathcal{G}(d, k, s, R, \kappa, w_{\min})$. If*

$$n \geq \frac{405k^2}{2\alpha^2} \ln(2k/\beta),$$

*then with probability at least $1 - \beta$, for every label $\rho_i$, for $1 \leq i \leq k$, such that $w_i \geq \frac{4\alpha}{9k}$, the number of points having label $\rho_i$ is in*

$$\left[ n\left( w_i - \frac{\alpha}{9k} \right), n\left( w_i + \frac{\alpha}{9k} \right) \right].$$

*Proof.* It follows directly from Lemma D.4 by setting $p = w_i$ and $\varepsilon = \frac{4\alpha}{9k}$, and taking the union bound over all mixture components.    □

**Lemma A.3.** *Let $X^L = ((X_1, \rho_1), \ldots, (X_n, \rho_n))$ be a labelled sample from a Gaussian mixture in $\mathcal{D} \in \mathcal{S}(\ell, k, \kappa, s)$, where $\ell \geq 512 \ln(nk/\beta)$ and $s > 0$. Then with probability at least $1 - \beta$, For every $1 \leq u \leq k$, the radius of the smallest ball containing the set of points with label $u$ (i.e. $\{X_i : \rho_i = u\}$) is in $[\sqrt{d}\sigma_u/2, \sqrt{3d}\sigma_u]$.*

*Proof.* For a given $u$, if $X_i, X_j$ are samples from $G_u$, then by Lemma D.6, we have with probability at least $1 - 4e^{-t^2/8}$,

$$2\ell\sigma_u^2 - 2t\sigma_u^2 \sqrt{\ell} \leq \left\| X_i - X_j \right\|_2^2 \leq 2\ell\sigma_u^2 + 2t\sigma_u^2 \sqrt{\ell}.$$

Setting $t = 8\sqrt{2\ln(mk/\beta)}$, and taking a union bound over all Gaussians and all pairs of samples from every Gaussian, we have that with probability at least $1 - \beta$, for any $u$ and all pairs of

points $X_i, X_j$ from $G_u$,

$$2\ell\sigma_u^2 - 16\sigma_u^2 \sqrt{2\ell\ln\left(\frac{mk}{\beta}\right)} \le \left\|X_i - X_j\right\|_2^2 \le 2\ell\sigma_u^2 + 16\sigma_u^2 \sqrt{2\ell\ln\left(\frac{mk}{\beta}\right)}.$$

By our assumption on $\ell$, we get

$$\ell\sigma_u^2 \le \left\|X_i - X_j\right\|_2^2 \le 3\ell\sigma_u^2.$$

Now, because the distance between any two points from the same Gaussian is at least $\sqrt{\ell}\sigma_u$, the smallest ball containing all the points must have radius at least $\sqrt{\ell}\sigma_u/2$.

For the upper bound, for any $u$, let the mean of all points, $P = \frac{1}{m_u} \sum\limits_{i=1}^{m_u} X_i$ be a candidate center for the required ball, where $X_i$ are samples from $G_u$, and $m_u$ is the number of samples from the Gaussian. Then for any point $X$ sampled from the Gaussian,

$$\|X - P\|_2 \le \left\|\frac{1}{m_u}\sum_{i=1}^{m_u} X_i - X\right\|_2 \le \frac{1}{m_u}\sum_{i=1}^{m_u}\|X_i - X\|_2 \le \frac{1}{m_u}\cdot m_u \sqrt{3\ell}\sigma_u = \sqrt{3\ell}\sigma_u.$$

This proves the lemma. $\qquad\square$

**Lemma A.4.** *Let $X^L = ((X_1, \rho_1), \ldots, (X_n, \rho_n))$ be a labelled sample from a Gaussian mixture in $\mathcal{D} \in \mathcal{S}(\ell, k, \kappa, C\sqrt{\ell})$, where $\ell \ge 512 \max\{\ln(nk/\beta), k\}$ and $C > 1$ is a universal constant. Then with probability at least $1 - \beta$, For every $\rho_i \ne \rho_j$,*

$$\left\|X_i - X_j\right\|_2 \ge \frac{C}{2}\sqrt{\ell}\max\{\sigma_{\rho_i}, \sigma_{\rho_j}\}.$$

*Proof.* Let $x, y$ be points as described in the statement of the Lemma. From Lemma D.6, we know that with probability at least $1 - 4e^{-t^2/8}$,

$$\left\|x - y\right\|_2^2 \ge \mathbb{E}\left[\left\|x - y\right\|_2^2\right] - t\left((\sigma_i^2 + \sigma_j^2)\sqrt{\ell} + 2\left\|\mu_i - \mu_j\right\|_2\sqrt{\sigma_i^2 + \sigma_j^2}\right)$$

$$= (\sigma_i^2 + \sigma_j^2)\ell + \left\|\mu_i - \mu_j\right\|_2^2 - t\left((\sigma_i^2 + \sigma_j^2)\sqrt{\ell} + 2\left\|\mu_i - \mu_j\right\|_2\sqrt{\sigma_i^2 + \sigma_j^2}\right).$$

Setting $t = 16\sqrt{\ln(mk/\beta)}$, $\sigma = \max\{\sigma_i, \sigma_j\}$, taking the union bound over all pairs of Gaussians, and all pairs of points from any two Gaussians, and using the assumption on $d$, we get the following with probability at least $1 - \beta$.

$$\left\|x - y\right\|_2^2 \ge (\sigma_i^2 + \sigma_j^2)\ell + \frac{\left\|\mu_i - \mu_j\right\|_2^2}{2} - 16(\sigma_i^2 + \sigma_j^2)\sqrt{\ell\ln\left(\frac{mk}{\beta}\right)}$$

$$\ge \left(1 + \frac{C^2}{2} - \frac{C}{2\sqrt{2}}\right)\sigma^2\ell > \frac{C^2}{4}\sigma^2\ell$$

This proves the lemma. $\qquad\square$

# B  Private Location for Mixtures of Gaussians

**Theorem B.1** ([NS18]). *Suppose $X_1, \ldots, X_m$ are points from $S^\ell \subset \mathbb{R}^\ell$, where $S^\ell$ is finite. Let $m, t, \beta, \varepsilon, \delta$ be such that,*

$$t \geq O\left(\frac{m^a \cdot \sqrt{\ell}}{\varepsilon} \log\left(\frac{1}{\beta}\right) \log\left(\frac{m\ell}{\beta\delta}\right) \sqrt{\log\left(\frac{1}{\beta\delta}\right)} \cdot 9^{\log^*(2|S|\sqrt{\ell})}\right),$$

*where $0 < a < 1$ is a constant that could be arbitrarily small. Let $r_{opt}$ be the radius of the smallest ball that contains at least $t$ points from the sample. There exists an $(\varepsilon, \delta)$-DP algorithm (called $\mathrm{PLOC}_{\varepsilon,\delta,\beta}$) that returns a ball of radius at most $w \cdot r_{opt}$ such that it contains at least $t - \Delta$ points from the sample with error probability $\beta$, where $w = O(1)$ (where the constant depends on the value of a), and*

$$\Delta = O\left(\frac{m^a}{\varepsilon} \log\left(\frac{1}{\beta}\right) \log\left(\frac{1}{\beta\delta}\right) \cdot 9^{\log^*(2|S|\sqrt{\ell})}\right).$$

---

**Algorithm 5:** Private Gaussians Location $\mathrm{PGLOC}(S, t; \varepsilon, \delta, R, \sigma_{\min}, \sigma_{\max})$

---

**Input:** Samples $X_1, \ldots, X_m \in \mathbb{R}^\ell$ from a mixture of Gaussians. Number of points in the target ball: $t$. Parameters $\varepsilon, \delta, \beta > 0$.

**Output:** Center $\vec{c}$ and radius $r$ such that $B_r(\vec{c})$ contains at least $t/2$ points from $S$.

Set parameters: $\lambda \leftarrow 0.1$

Let $\mathcal{X}$ be a grid in $[-R - 3\sqrt{\ell}\sigma_{\max}\kappa, R + 3\sqrt{\ell}\sigma_{\max}\kappa]^\ell$ of width $\lambda = \frac{\sigma_{\min}}{10}$.
Round points of $S$ to their nearest points in the grid to get dataset $S'$
$(\vec{c}, r') \leftarrow \mathrm{PLOC}_{\varepsilon,\delta,\beta}(S', t, \mathcal{X})$
Let $r \leftarrow r' + \lambda\sqrt{\ell}$

**Return** $(\vec{c}, r)$

---

*Proof of Theorem 4.11.* We show that Algorithm 5 satisfies the conditions in the theorem.

Privacy follows from Theorem B.1, and post-processing (Lemma 2.8).

The first part of the theorem follows directly from Theorem B.1 by noting that $\Delta \leq \frac{t}{2}$ for large enough $n, \ell, |S|$, and $t = \gamma n$, where $0 < \gamma \leq 1$. For all $x \in X$, with high probability, it holds that $\|x\|_2 \leq R + O(\sqrt{\ell}\sigma_{\max})$ by applying Lemma 2.16 after rescaling $x$ appropriately by its covariance, then applying the triangle inequality, and noting that the empirical mean of a set of points lies in their convex hull.

Now, we move to the second part of the lemma. Because of the discretization, we know that,

$$\left\|x - x'\right\|_2 \leq \lambda\sqrt{\ell}. \tag{8}$$

Therefore,

$$\left\|\vec{p} - x\right\|_2 \leq \left\|\vec{p} - x'\right\|_2 + \left\|x' - x\right\|_2$$
$$\leq r' + \lambda\sqrt{\ell}.$$

Let $x, y \in S$ and $x', y' \in S'$ be their corresponding rounded points. Then from Equation 8, we know that

$$\left\| x' - y' \right\|_2 \leq \left\| x' - x \right\|_2 + \left\| x - y \right\|_2 + \left\| y - y' \right\|_2$$
$$\leq \left\| x - y \right\|_2 + 2\lambda\sqrt{\ell}. \tag{9}$$

Let $r'_{opt}$ be the radius of the smallest ball containing at least $t$ points in $S'$. Because of Equation 9, we can say that

$$r'_{opt} \leq r_{opt} + 2\lambda\sqrt{\ell}.$$

From the correctness of PLOC, we can conclude the following,

$$r' \leq c r'_{opt}$$
$$\leq c\left( r_{opt} + 2\lambda\sqrt{\ell} \right),$$

where $c > 4$ is an absolute constant. This gives us,

$$r = r' + \lambda\sqrt{\ell}$$
$$\leq c\left( r_{opt} + \frac{9}{4}\lambda\sqrt{\ell} \right)$$
$$\leq c\left( r_{opt} + \frac{1}{4}\sqrt{\ell}\sigma_{\min} \right).$$

This completes the proof. $\qquad\square$

## C   Private Estimation of a Single Spherical Gaussian

*Proof of Theorem 4.16.* We show that Algorithm 6 satisfies the conditions in the theorem.

For the privacy argument for Algorithm 6, note that we truncate the dataset such that all points in $X$ lie within $B_r(\vec{c})$. Now, the following are the sensitivities of the computed functions.

1. $\sum_{i=1}^{|X'|} X_i$: $\ell_2$-sensitivity is $2r$.

2. $\sum_{i=1}^{m'} Y_i^2$: $\ell_1$-sensitivity is $2r^2$.

3. Number of points in $X'$: $\ell_1$-sensitivity is 1.

Therefore, by Lemmata 2.13, 2.11, and 2.9, the algorithm is $(\varepsilon, \delta)$-DP.

Firstly, either not all the points of $X$ lie within $B_r(\vec{c})$, that is, $|X'| < m$, in which case, we're done, or all the points do lie within that ball, that is, $|X'| = m$. Since we can only provide guarantees when we have a set of untampered random points from a Gaussian, we just deal with the second case. So, $m'_X = m + e$, where $e$ is the noise added by PCOUNT. Because $m \geq \frac{6}{\varepsilon}\ln(5/\beta)$, using Lemma D.1, we know that with probability at least $1 - \frac{\beta}{5}$, $|e| \leq \frac{3\ln(5/\beta)}{\varepsilon}$, which means that $m \geq 2|e|$. Because of this, the following holds.

$$\frac{1}{m}\left( 1 - \frac{|e|}{2m} \right) \leq \frac{1}{m} \leq \frac{1}{m}\left( 1 + \frac{2|e|}{m} \right) \quad \text{and} \quad \frac{1}{m}\left( 1 - \frac{|e|}{2m} \right) \leq \frac{1}{m+e} \leq \frac{1}{m}\left( 1 + \frac{2|e|}{m} \right) \tag{10}$$

---

**Algorithm 6:** Private Spherical Gaussian Estimator $\text{PSGE}(X; \vec{c}, r, \varepsilon, \delta)$

---

**Input:** Samples $X_1, \ldots, X_m \in \mathbb{R}^\ell$. Center $\vec{c} \in \mathbb{R}^\ell$ and radius $r > 0$ of target component.
Parameters $\varepsilon, \delta > 0$.

**Output:** Mean and variance of the Gaussian.

Set parameters: $\Delta_{\frac{\varepsilon}{3}, \sigma} \leftarrow \frac{6r^2}{\varepsilon} \quad \Delta_{\frac{\varepsilon}{3}, \delta, \mu} \leftarrow \frac{6r\sqrt{2\ln(1.25/\delta)}}{\varepsilon}$

Let $X' \leftarrow X \cap B_r(\vec{c})$

For each $i$ such that $X_{2i}, X_{2i-1} \in X'$, let $Y_i \leftarrow \frac{1}{\sqrt{2}}(X_{2i} - X_{2i-1})$, and let $Y \leftarrow Y_1, \ldots, Y_{m'}$

Let $m'_X = \text{PCOUNT}_{\frac{\varepsilon}{3}}(X, B_r(\vec{c}))$ and $m'_Y = \frac{m'_X}{2}$

```
// Private Covariance Estimation
```
$\widehat{\sigma}^2 \leftarrow \frac{1}{m'_Y \ell}\left(\sum_{i=1}^{m'} \|Y_i\|_2^2 + z_\sigma\right)$, where $z_\sigma \sim \text{Lap}\left(\Delta_{\frac{\varepsilon}{2}, \sigma}\right)$

```
// Private Mean Estimation
```
$\widehat{\mu} \leftarrow \frac{1}{m'_X}\left(\sum_{i=1}^{|X'|} X'_i + z_\mu\right)$, where $z_\mu \sim \mathcal{N}\left(0, \Delta^2_{\frac{\varepsilon}{3}, \delta, \mu} \mathbb{I}_{\ell \times \ell}\right)$

**Return** $(\widehat{\mu}, \widehat{\sigma}^2)$

---

We start by proving the first part of the claim about the estimated mean. Let $S_{X'} = \sum_{i=1}^{m} X'_i$ and $S_{X'}^{\vec{c}} = \sum_{i=1}^{m}(X'_i - \vec{c})$. Then,

$$\left\|\frac{1}{m} S_{X'}^{\vec{c}} - \frac{1}{m+e} S_{X'}^{\vec{c}}\right\|_2 = \frac{\left\|S_{X'}^{\vec{c}}\right\|_2 |e|}{m(m+e)}.$$

We want the above to be at most $\frac{\sigma \alpha_\mu}{4}$. This gives us the following.

$$m(m+e) \geq \frac{4\left\|S_{X'}^{\vec{c}}\right\|_2 |e|}{\alpha_\mu \sigma}$$

Because $|e| \leq \frac{m}{2}$, it is sufficient to have the following.

$$m^2 \geq \frac{8\left\|S_{X'}^{\vec{c}}\right\|_2 |e|}{\alpha_\mu \sigma}$$

But because $X_i \in B_r(\vec{c})$ for all $i$, $\left\|X_i - \vec{c}\right\|_2 \leq r$. So, due to our bound on $|e|$, it is sufficient to have the following.

$$m^2 \geq \frac{24mr\ln(5/\beta)}{\varepsilon \alpha_\mu \sigma} \iff m \geq \frac{24r\ln(5/\beta)}{\varepsilon \alpha_\mu \sigma}$$

This gives us,

$$\left\| \frac{1}{m} S_{X'} - \frac{1}{m+e} S_{X'} \right\|_2 \le \frac{\sigma \alpha_\mu}{4}.$$

Now, we want to bound the distance between $\frac{1}{m+e} S_{X'}$ and $\frac{1}{m+e}(S_{X'} + z_\mu)$ (by $\frac{\sigma \alpha_\mu}{4}$).

$$\left\| \frac{1}{m+e} S_{X'} - \frac{1}{m+e}(S_{X'} + z_\mu) \right\|_2 \le \frac{\sigma \alpha_\mu}{4} \iff \left\| \frac{z_\mu}{m+e} \right\|_2 \le \frac{\sigma \alpha_\mu}{4} \iff m + e \ge \frac{4 \|z_\mu\|_2}{\sigma \alpha_\mu}$$

Because $m \ge 2|e|$, it is sufficient to have the following.

$$\frac{m}{2} \ge \frac{4 \|z_\mu\|_2}{\sigma \alpha_\mu} \iff m \ge \frac{8 \|z_\mu\|_2}{\sigma \alpha_\mu}$$

Using Lemma D.5, we and noting that $\ell \ge 8 \ln(10/\beta)$, we know that with probability at least $1 - \frac{\beta}{5}$,

$$\|z_\mu\|_2 \le \frac{6r\sqrt{2\ln(1.25/\delta)}\sqrt{2\ell}}{\varepsilon} = \frac{12r\sqrt{\ell \ln(1.25/\delta)}}{\varepsilon}$$

Therefore, it is sufficient to have the following.

$$m \ge \frac{96r\sqrt{\ell \ln(1.25/\delta)}}{\sigma \alpha_\mu \varepsilon}$$

To complete the proof about the accuracy of the estimated mean, we need to bound the distance between $\frac{1}{m} S_{X'}$ and $\mu$ (by $\frac{\sigma \alpha_\mu}{2}$). Let $\widetilde{\mu} = \frac{1}{m} S_{X'}$. Using Lemma 2.16, and the fact that $m \ge \frac{c_1 \ell + c_2 \log(1/\beta)}{\alpha_\mu^2}$ for universal constants $c_1, c_2$, we get that with probability at least $1 - \frac{\beta}{5}$,

$$\|\mu - \widetilde{\mu}\|_2 \le \frac{\alpha_\mu \sigma}{2}.$$

We finally apply the triangle inequality to get,

$$\|\mu - \widehat{\mu}\|_2 \le \frac{\sigma \alpha_\mu}{2} + \frac{\sigma \alpha_\mu}{4} + \frac{\sigma \alpha_\mu}{4} = \sigma \alpha_\mu.$$

We now prove the lemma about the estimated covariance. Note that $m' = \frac{m}{2}$. Let $\Sigma_Y = \sum_{i=1}^{m'} \|Y_i\|_2^2$. We want to show the following.

$$(1 - \alpha_\sigma)^{1/3} \sigma^2 \le \frac{1}{m'\ell} \Sigma_Y \le (1 + \alpha_\sigma)^{1/3} \sigma^2$$

$$(1 - \alpha_\sigma)^{1/3} \frac{1}{m'\ell} \Sigma_Y \le \frac{1}{m_Y'\ell} \Sigma_Y \le (1 + \alpha_\sigma)^{1/3} \frac{1}{m'\ell} \Sigma_Y$$

$$(1 - \alpha_\sigma)^{1/3} \frac{1}{m_Y'\ell} \Sigma_Y \le \frac{1}{m_Y'\ell}(\Sigma_Y + z_\sigma) \le (1 + \alpha_\sigma)^{1/3} \frac{1}{m_Y'\ell} \Sigma_Y$$

The claim would then follow by substitution. We use the fact that for any $t \in [0,1]$,

$$(1 - t)^{1/3} \le 1 - \frac{t}{6} \le 1 + \frac{t}{6} \le (1 + t)^{1/3}. \tag{11}$$

We start by proving the first inequality. Note that for each $i, j$, $Y_i^j \sim \mathcal{N}(0, \sigma^2)$ is i.i.d. Using Lemma D.5 and the fact that $m \geq \frac{576}{\alpha_\sigma^2 \ell} \ln(10/\beta)$, we know that with probability at least $1 - \frac{\beta}{5}$,

$$\left(1 - \frac{\alpha_\sigma}{6}\right)\sigma^2 \leq \frac{1}{m'\ell}\Sigma_Y \leq \left(1 + \frac{\alpha_\sigma}{6}\right)\sigma^2.$$

Combining the above with Inequality 11, we get the first result.

To prove the second result, since $m' = \frac{m}{2}$, using Inequality 10, it is enough to show that,

$$\frac{\Sigma_Y}{m'\ell}\left(1 + \frac{|e|}{m'}\right) \leq \frac{\Sigma_Y}{m'\ell}\left(1 + \frac{\alpha_\sigma}{6}\right) \quad \text{and} \quad \frac{\Sigma_Y}{m'\ell}\left(1 - \frac{\alpha_\sigma}{6}\right) \leq \frac{\Sigma_Y}{m'\ell}\left(1 - \frac{|e|}{4m'}\right).$$

Since with high probability, $|e| \leq \frac{3}{\varepsilon}\ln(5/\beta)$, having $m \geq \frac{36}{\varepsilon\alpha_\sigma}\ln(5/\beta)$ satisfies the two conditions. This gives us,

$$\left(1 - \frac{\alpha_\sigma}{6}\right)\frac{1}{m'\ell}\Sigma_Y \leq \frac{1}{m'_Y\ell}\Sigma_Y \leq \left(1 + \frac{\alpha_\sigma}{6}\right)\frac{1}{m'\ell}\Sigma_Y,$$

which gives us the required result after combining with Inequality 11.

To prove the third result, it is sufficient to show the following.

$$\frac{1}{m'_Y\ell}(\Sigma_Y + |z_\sigma|) \leq \frac{\Sigma_Y}{m'_Y\ell}\left(1 + \frac{\alpha_\sigma}{6}\right) \quad \text{and} \quad \frac{\Sigma_Y}{m'_Y\ell}\left(1 - \frac{\alpha_\sigma}{6}\right) \leq \frac{1}{m'_Y\ell}(\Sigma_Y - |z_\sigma|)$$

Note that from Lemma D.1, with probability at least $1 - \frac{\beta}{10}$,

$$|z_\sigma| \leq \frac{6r^2\ln(10/\beta)}{\varepsilon}.$$

From Lemma A.3, we know that for any $i, j$, with probability at least $1 - \frac{\beta}{10}$,

$$\left\|Y_i - Y_j\right\|_2 \geq \frac{\sqrt{\ell}\sigma}{2}.$$

This means that at least half of the points of $Y$ must have $L_2$ norms at least $\frac{\sqrt{\ell}\sigma}{4}$, which implies that $\Sigma_Y \geq \frac{m'\ell\sigma^2}{32} = \frac{m\ell\sigma^2}{64}$. Then the two conditions above will be satisfied if,

$$\frac{m\ell\sigma^2}{64} \geq \frac{6}{\alpha_\sigma} \cdot \frac{6r^2\ln(10/\beta)}{\varepsilon} \iff m \geq \frac{2304r^2\ln(10/\beta)}{\alpha_\sigma\varepsilon\sigma^2\ell}.$$

This gives us,

$$\left(1 - \frac{\alpha_\sigma}{6}\right)\frac{1}{m'_Y\ell}\Sigma_Y \leq \frac{1}{m'_Y\ell}(\Sigma_Y + z_\sigma) \leq \left(1 + \frac{\alpha_\sigma}{6}\right)\frac{1}{m'_Y\ell}\Sigma_Y,$$

which when combined with Inequality 11, gives us the third result. Combining the three results via substitution, we complete the proof for the accuracy of the estimated variance. $\square$

# D  Additional Useful Concentration Inequalities

Throughout we will make use of a number of concentration results, which we collect here for convenience. We start with standard tail bounds for the univariate Laplace and Gaussian distributions.

**Lemma D.1** (Laplace Tail Bound). *Let $Z \sim \mathrm{Lap}(t)$. Then $\mathbb{P}\left[|Z| > t \cdot \ln(1/\beta)\right] \le \beta$.*

**Lemma D.2** (Gaussian Tail Bound). *Let $X \sim \mathcal{N}(\mu, \sigma^2)$. Then $\mathbb{P}\left[\left|X - \mu\right| > \sigma\sqrt{2\ln(2/\beta)}\right] \le \beta$.*

We also recall standard bounds on the sums of well behaved random variables.

**Lemma D.3** (Multiplicative Chernoff). *Let $X_1, \dots, X_m$ be independent Bernoulli random variables taking values in $\{0, 1\}$. Let $X$ denote their sum and let $p = \mathbb{E}[X_i]$. Then for $m \ge \frac{12}{p}\ln(2/\beta)$,*

$$\mathbb{P}\left[X \notin \left[\frac{mp}{2}, \frac{3mp}{2}\right]\right] \le 2e^{-mp/12} \le \beta.$$

**Lemma D.4** (Bernstein's Inequality). *Let $X_1, \dots, X_m$ be independent Bernoulli random variables taking values in $\{0, 1\}$. Let $p = \mathbb{E}[X_i]$. Then for $m \ge \frac{5p}{2\varepsilon^2}\ln(2/\beta)$ and $\varepsilon \le p/4$,*

$$\mathbb{P}\left[\left|\frac{1}{m}\sum X_i - p\right| \ge \varepsilon\right] \le 2e^{-\varepsilon^2 m/2(p+\varepsilon)} \le \beta.$$

**Lemma D.5** (Concentration of Empirical Variance). *Let $X_1, \dots, X_m \sim \mathcal{N}(0, \sigma^2)$ be independent. If $m \ge \frac{8}{\varepsilon^2}\ln\left(\frac{2}{\beta}\right)$ and $\varepsilon \in (0, 1)$, then*

$$\mathbb{P}\left[\left|\frac{1}{m}\sum_{i=1}^{m} X_i^2 - \sigma^2\right| > \varepsilon\sigma^2\right] \le \beta.$$

Finally, we have a concentration lemma from [VW02] for the distance between two points drawn from not-necessarily identical spherical Gaussians.

**Lemma D.6.** *Let $X \sim \mathcal{N}(\mu_1, \sigma_1^2 \mathbb{I}_{d \times d})$ and $y \sim \mathcal{N}(\mu_2, \sigma_2^2 \mathbb{I}_{d \times d})$. For $t > 0$,*

$$\mathbb{P}\left[\left|\left\|x - y\right\|_2^2 - \mathbb{E}\left[\left\|x - y\right\|_2^2\right]\right| > t\left((\sigma_1^2 + \sigma_2^2)\sqrt{d} + 2\left\|\mu_1 - \mu_2\right\|_2\sqrt{\sigma_1^2 + \sigma_2^2}\right)\right] \le 4e^{-t^2/8}.$$