[Reviews · NeurIPS 2019]

Reviewer 1



The point about strong bounds on parameters not being required is somewhat subtle (and perhaps overstated in the abstract and intro) because bounds on these parameters are required to be known, but the sample complexity depends only polynomially and polylogarithmically on these bounds, so an analyst can have very loose bounds on these terms and it will not affect the sample complexity significantly. From a first read of the abstract/intro, it seems like these parameters are not required to known at all. I would encourage the authors to re-phrase the informal statements of these claims to more accurately represent the results of the paper. The algorithm works similarly to that of Achlioptas and McSherry by using PCA to project the data into a low-dimensional space and then recursively clustering data points. It differs from this previous work by using differentially private versions of PCA and clustering. The authors develop a new algorithm for private clustering that is based on private clustering algorithms of Nissim, Stemmer, and Vadhan, but satisfies additional properties needed for this particular problem. The authors also derive sample complexity results for solving the problem using subsample-and-aggregate, and show their algorithm has asymptotically better sample complexity. Overall, this paper provides novel technical results on a relevant problem in private machine learning, and would be a good fit for NeurIPS. The writing and presentation is also very clear. Minor comment: pg 4, line 132: "an secluded cluster"

Reviewer 2



The paper considers the fundamental problem of learning a mixture of Gaussians under differential privacy. It builds on an algorithm by Achlioptas and McSherry and the robustness of PCA-projection to noise. The resulting sample complexity (for achieving a given level of accuracy) matches that of the non-private version of the algorithm. In particular, it shows a provable advantage compared with the well-known sample-and-aggregate framework. The paper is well written in general and the contributions are clearly stated.

Reviewer 3



Post rebuttal: the author feedback addresses my concerns, though I still think the readability of the paper needs improvement. I increased the overall score. --------- My main concern is that the main algorithm (Sec 4) and the privacy analysis are not clearly stated, making it hard to check the quality of the paper. More specifically, In terms of clarity, - Algorithm 1 and 2 needs to be better explained in the text. The pseudo code can also be made more readable, for example, Line 4 and 7 are actually if-then-else statements but the else parts are implicit (the full version is better). - PTERRIFICBALL should be explained more. It is unclear how much noise is added. - It seems to me Algorithm 1 would find k clusters. Yet the recursive part calls it twice with k-1. Doesn’t that mean we would have 2(k-1) clusters? - Line 3 in Algorithm 2 mentioned a subroutine PGE which is not defined. PGE seems to be important as it is the main component in Algorithm 2 that guarantees Differential Privacy. - Line 251, is TerrificRadius the same as TerrificBall? In terms of privacy analysis: - In Algorithm 2, why do we add Lap(1/epsilon) noise to |Cj|? Is the sensitivity 1?

[Author Response · NeurIPS 2019]

We first would like to thank all the reviewers for their careful reading and constructive comments. We especially would like to thank Reviewer 6 for their detailed suggestions. Indeed, given the length of the full version (over 50 pages in the supplementary material), compressing our results to an 8 page version proved to be a significant challenge, and we appreciate a reading by a fresh pair of eyes and specific recommendations on ways to improve it. We will certainly take all of these into account.

We are encouraged by all the reviewers' positive comments on the strength of our work's contributions, and we will work to incorporate their comments. We spend the remainder of our response addressing specific technical questions and suggestions on the presentation.

**Reviewer 3:** Regarding the point about strong bounts on parameters: to avoid confusion, we will be more precise in the introduction by saying that we require only weak bounds on the range parameters.

**Reviewer 6:** We address the reviewer's points in the order they are stated:

- Regarding the explanation of Algorithms 1 and 2: We will try to expand on our analyses and algorithm description. The full version does contain a precise desription of the algorithm, which might help with the overall understanding of the algorithm.

- Regarding PTerrificBall: The magnitude of the noise added in PTerrificBall is actually the $\Gamma$ parameter in Lemma 4.2 that guarantees the return of a $(c, 2\Gamma)$-terrific ball. We will specify what it is, and say that it stems from the noise added by the invocation of AboveThreshold in the TerrificRadius procedure.

- Regarding recursive calls in Algorithm 1: In Algorithm 1, the parameter $k$ is an upper bound on the number of Gaussian components that can have points in the dataset. So, the dataset $X$ supplied to it actually has points from $k' \leq k$ Gaussian components. When it makes the recursive calls with $k - 1$ as the corresponding argument, it actually means that the dataset it supplies to each call can have points from at most $k - 1$ Gaussian components. So, for example, if in a recursive call, if the dataset it supplies has points from exactly $k' - 2$ Gaussians, then the recursive call will return $k' - 2$ clusters, while the other recursive call will return exactly 2 clusters, even though the upper bound on the number of components supplied to each call is $k - 1$.

- Regarding the PGE procedure: PGE is actually the differentially private learner for high-dimensional Gaussians from [KLSU19], with a small caveat. [KLSU19] is guaranteed to estimate a Gaussian in total variation distance when it gets independent samples from the Gaussian, but in our case, a small fraction of points can be removed in the clustering process. We can prove that with a small multiplicative overhead in sample complexity, it can still learn the Gaussian accurately. We do make this more formal in the supplement.

- Regarding TerrificRadius versus TerrificBall: TerrificRadius is the modified version of GoodRadius of [NSV16]. As discussed within the same paragraph, TerrificBall is an application of two private algorithms: TerrificRadius, followed by the modified GoodCenter algorithm of [NS18].

- Regarding noise addition in Algorithm 2: Yes, in Algorithm 2, the sensitivity is 1. An alternative view of Algorithm 1 could be that it takes sets of points and their respective indices in the original dataset as input, and returns subsets of indices, so that no two subsets index points corresponding to the same Gaussian component. So, in line 3 of Algorithm 2, we would compute the size of each indexing subset privately, and use its corresponding points to learn its Gaussian component. Given a dataset $X$, and a set of indices $S$, the function that computes the number of points in $X$ indexed by $S$, has sensitivity 1. This is because by changing a point in the dataset, the new point may or may not lie in the subset indexed by S. We will modify our presentation to reflect this better.

# References

[KLSU19] Gautam Kamath, Jerry Li, Vikrant Singhal, and Jonathan Ullman. Privately learning high-dimensional distributions. In *Proceedings of the 32nd Annual Conference on Learning Theory*, COLT '19, 2019.

[NS18] Kobbi Nissim and Uri Stemmer. Clustering algorithms for the centralized and local models. In *Algorithmic Learning Theory*, ALT '18, pages 619–653. JMLR, Inc., 2018.

[NSV16] Kobbi Nissim, Uri Stemmer, and Salil Vadhan. Locating a small cluster privately. In *Proceedings of the 35th ACM SIGMOD-SIGACT-SIGART Symposium on Principles of Database Systems*, PODS '16, pages 413–427, New York, NY, USA, 2016. ACM.


[Meta-Review · NeurIPS 2019]

The paper studies the problem of learning a mixture of well-separated Gaussians under a DP constraint. This is a basic ML task and studying DP versions of it is natural. The current work gives a Differentially Private version of a simple spectral algorithm of Achlioptas and McSherry. This authors show that their algorithm has a very small sample complexity overhead for a large range of parameters. This improves on previous work on this problem, and the reviewers found the techniques to be interesting.